# Selective conversion of syngas to C_{4+} long-chain alcohols

Yihui Li[1,5], Ziang Zhao[1,5], Miao Jiang[1], Guoqing Wang[1], Zheng Li[1,2], Wei Lu[1], Wenhao Cui[1], Rong Liu[3], Ronghe Lin [4]✉, Yu Meng[3]✉, Yuan Lyu[1], Li Yan [1]✉, Hejun Zhu [1]✉ & Yunjie Ding [1,4]✉

The selective conversion of syngas to C_{4+} long-chain alcohols holds significant industrial and scientific interest, but challenges in product selectivity and process efficiency remain. Here, we report a precisely catalytic strategy for C_{4+} alcohol synthesis with a selectivity of 80% at 17% CO conversion. The reaction channel involves: (i) the development $Cs_2O$-$Co_2C$-Co catalysts, capable of catalyzing CO hydrogenation to long-chain oxygenates/olefins; and (ii) complete conversion to C_{4+} alcohols is subsequently achieved on the single-Rh-site and Cu-$ZrO_2$ interfaces by integrated cooperative catalysis. A comprehensive catalyst design and compatibility assessment of each catalytic module ensures optimal combinations, meanwhile effectively eliminates costly separation steps, and reduces $CO_2$ selectivity down to 1%. The developed process achieves ultra-high carbon-efficiency (>95%) and improves oxygen-efficiency, effectively overcoming the key limitations of current syngas conversion technologies and thus representing a competitive and sustainable solution for producing high-value long-chain alcohols with a minimal carbon footprint.

C_{4+} long-chain alcohols are critical chemicals with diverse applications across the food, textile, pharmaceutical, and energy sectors[1,2]. These alcohols serve as crucial raw materials for a variety of fine chemicals, with specific chain lengths determining their purposes. For instance, butanol, a key precursor in plastics (e.g., butyl acrylate and butyl acetate) and gasoline additive, The global markets for n-butanol are projected to grow at a compound annual growth rates (CAGR) of 6.3% during 2021–2032[3,4], reflecting their economic significance. Beyond these, C_5-C_9 alcohols and C_{10}-C_{20} fatty alcohols are essential in manufacturing plasticizers, surfactants, and detergents, with global market values projected at 8.14 and 14.98 billion USD, respectively[5,6].

Current industrial production techniques for C_{4+} alcohols, which mainly include carbonylation (OXO-synthesis) and Ziegler process, face significant limitations in terms of scalability, complexity and sustainability. The carbonyl synthesis process is complicated, and the

separation and recovery of the catalyst from the product are difficult, hindering its large-scale application. For the Ziegler process, ethylene oligomerization relies on highly reactive triethyl aluminum, which restricts the production of odd-chain primary alcohols[7,8]. As global demand for C_{4+} alcohols grows, alternative synthetic routes are urgently needed. Syngas-based synthesis offers a more sustainable alternative, as it enables the utilization of diverse feedstocks such as natural gas, biomass, or even $CO_2$. Moreover, syngas-to-C_{4+} alcohols can more reasonably and efficiently transform oxygen-atoms from CO into organic products, which reduces $CO_2$ emissions or waste $H_2O$ as by-products. Aldol condensation (i.e., upgrading ethanol to long-chain alcohols) is an attractive route[9,10], considering the industrial maturity of ethanol synthesis from syngas-derived methanol, either via direct homogeneous carbonylation to oxygenates or heterogeneous dimethyl oxalate synthesis followed by subsequently hydrogenation

---

[1]State Key Laboratory of Catalysis, Dalian Institute of Chemical Physics, Chinese Academy of Sciences, Dalian, China. [2]University of Chinese Academy of Sciences, Beijing, China. [3]Shaanxi Key Laboratory of Low Metamorphic Coal Clean Utilization, School of Chemistry and Chemical Engineering, Yulin University, Yulin, China. [4]Key Laboratory of the Ministry of Education for Advanced Catalysis Materials, Zhejiang Key Laboratory of Advanced Catalysis and Adsorption Materials, Hangzhou Institute of Advanced Studies, Zhejiang Normal University, Hangzhou, China. [5]These authors contributed equally: Yihui Li, Ziang Zhao. ✉e-mail: catalysis.lin@zjnu.edu.cn; mengyu@yulinu.edu.cn; yanli@dicp.ac.cn; zhuhj@dicp.ac.cn; dyj@dicp.ac.cn

(Fig. S1). Despite its promise, technological immaturity and systemic complexity prevent its industrial implementation.

Direct synthesis of higher alcohol (HA) from syngas using single catalyst is conceptually attractive but faces serious challenges, particularly in product selectivity[11,12]. The reaction network involves complex steps, including CO and $H_2$ adsorption, $H_2$ dissociation, CO dissociation and non-dissociation activation, C-C coupling between various intermediates, and product desorption[13], often resulting in undesirable by-products, including alkanes, alkenes, and $CO_2$[8]. As demonstrated by Wasim et al., precise control over CO activation and C-C coupling is crucial for the selective synthesis of long-chain alcohols[14], which requires rational design of multifunctional catalysts to steer the reaction toward desired products. Despite advances in bi- or multi-component catalysts ($MoS_2$[15], Cu-Co[16-19], Cu-Fe[20], Co-Fe[21], Co-Mn[22,23], Cs-Cu-Zn[24]), selectivity for $C_{4+}$ alcohols remains low (ca. 20%, Fig. S1 and Table S5). Tandem catalysis approaches have achieved improved results, with notable results including CoMn|CuZnAlZr (46% oxygenates selectivity)[25], CuCoAl|t-$ZrO_2$ (33% $C_{2+}OH$ selectivity)[26], CuCoAl|ZnO/$ZrO_2$ (34% $C_{2+}OH$ selectivity)[27], $ZnCrAlO_x$|KNiMoS-MMO-5 (60% total alcohol selectivity)[28], CoMn/MAC(P)|Rh/3v-$PPh_3$@POPs (60.8% total oxygenates selectivity)[29] and NaPr-CoRu/AomM| $Co_2CO_8$ + $PCy_3$ (54% total alcohol)[30]. These novel catalytic processes have stimulated a new pulse in the integrated syngas conversion chemistry, but most efforts are focused on $C_{2+}$ alcohols rather than $C_{4+}$ products. In addition to the moderate selectivity of $C_{4+}$ alcohols (30%), the process is hampered by the complex separation and distillation steps and the limited availability of olefins. In China, a syngas-to-$C_{4+}$ alcohols process has been industrialized[31], utilizing by-product olefins from Fe-based Fischer-Tropsch synthesis (FTS) followed by hydroformylation and hydrogenation using aqueous-organic two-phase homogeneous hydroformylation catalysis. Nonetheless, additives are indispensable in order to enhance the solubility of $C_{4+}$ olefins in an aqueous phase, where the hydroformylation reaction was carried out, causing additional difficulties in catalyst-product separation.

Herein, we firstly report a precisely designed integrated catalytic system for selective conversion of syngas-to-$C_{4+}$ alcohols (Fig. S1). Appropriate design of $Cs_2O$-$Co_2C$-Co key catalysts for selectively catalyzing syngas to long-chain oxygenates/olefins, and then the complete transformation of the cascaded hydroformylation and hydrogenation to produce $C_{4+}$ alcohols was deeply explored. Key challenges, including achieving high selectivity toward oxygenates/ olefins and ensuring compatibility among different catalytic modules, were thoroughly studied through rational catalyst design and comprehensive process optimization. Density functional theory (DFT) calculations elucidated the high selectivity toward $C_{4+}$ alcohols/$C_{3+}$ olefins of a novel syngas-to-oxygenates/olefins catalyst, while compatibility assessments of integrated catalytic modules for target products (syngas-to-oxygenates/olefins plus olefins hydroformylation, syngas-to-oxygenates/olefins plus hydrogenation of oxygenates and olefins, and syngas-to-oxygenates/olefins plus olefins hydroformylation plus hydrogenation of oxygenates and olefins). Our optimized system combines $Cs_2O$-$Co_2C$-Co catalysts for the highly selective syngas-to-oxygenates/olefins conversion in the first reactor, the single-Rh-site catalyst for olefins hydroformylation and the $CuZrO_2$/$SiO_2$ catalyst for aldehydes hydrogenations in the second reactor. This integrated approach achieved an unprecedented 87% total alcohol selectivity, including 80% $C_{4+}$ alcohols and 50% $C_{6+}$ alcohols, respectively, establishing a promising route for sustainable long-chain alcohol production.

## Results

### Selective conversion of syngas to oxygenates/olefins

Syngas-to-oxygenates/olefins is the key step to produce $C_{4+}$ long-chain alcohols, as it not only directly determines CO conversion and product distribution, but also the composition of the intermediates will strongly influence the catalytic performances in subsequent hydroformylation and hydrogenation. Therefore, it is crucial to identify appropriate catalysts for the selective conversion of syngas into oxygenates/olefins at this stage. Carbon-supported cobalt (Co/C) was selected as the benchmark system for its demonstrated high performance in higher alcohol synthesis, and a series of modified analogs with different metal promoters (Mn, Cr, Ni, Ca, Ba, and Cs) was explored (Fig. 1a and Tables S1, 2). When 0.5 wt% Mn was doped in Co/ C, the selectivity of $C_{4+}$ oxygenates/$C_{3+}$ olefins was markedly increased from 37% to 50% without compromising CO conversion (Fig. 1b) and Mn was assumed to promote the C-C coupling and chain growth process[32], reducing methane selectivity from 22.0% to 10.2% (Table S1, S2 and Fig. S2). The CoMn/C with the fixed compositions was then chosen for further optimization with a second promoter. While the additions of Cr or Ni improved CO conversion but inhibited the formations of oxygenates/olefins, Ca and Ba additives promoted $CH_4$ formation, and great promotional effect was observed with the addition of Cs (Fig. 1a). The regulation of different metal additives has little effect on $CO_2$ selectivity (all less than 5%, Fig. S2). In particular, $Co_{0.5}Mn_{0.1}Cs$/C exhibited relatively low alkanes/olefins ratios (0.7-0.9) and the highest total selectivity of oxygenates/olefins, with the total selectivity of $C_{4+}$ oxygenates/$C_{3+}$ olefins reaching 73% (Table S1-2). Anderson-Schulz-Flory (ASF) plots have been constructed and are compiled in Fig. S3a-d. Accordingly, a linear ASF behavior with chain-lengthening probability $\alpha = 0.65$ is obtained, and as the pressure increased to 8 MPa, $\alpha$ increased to 0.72. Control experiments showed that additions of individual Mn or Cs could also improve the total selectivity of oxygenates/olefins and reduce $CH_4$ selectivity, with $\alpha$ values of 0.67 and 0.66, respectively, but both led to decreased activity (Table S1). Under the same reaction conditions, the CO conversion on $Co_{0.5}Mn_{0.1}Cs$/C (29%) was higher than those on $Co_{0.5}Mn$/C (20%) and $Co_{0.1}Cs$/C (11%, Fig. 1b and Table S1). The above results strongly point to the synergy between Mn/Cs additives and the active Co species in steering the formation of desirable products.

To understand the promotional effect of Mn and Cs co-addition, key catalysts were characterized by multiple techniques. Powder X-ray diffraction (PXRD) patterns revealed the typical diffraction lines corresponding to face-centered cubic (fcc) Co on $Co_{0.5}Mn$/C, $Co_{0.1}Cs$/C, and $Co_{0.5}Mn_{0.1}Cs$/C (Fig. S4). The noticeable peaks at 41.3°, 42.5°, and 45.7° $2\theta$ corresponding to $Co_2C$ was more evident on $Co_{0.5}Mn_{0.1}Cs$/C. This result was corroborated by deconvolution of the Co $2p$ X-ray photoelectron spectra (XPS) for $Co_{0.5}Mn_{0.1}Cs$/C, the Co $2p_{3/2}$ binding energy peaks at 780.9 and 778.3 eV were assigned to $Co^{2+}$ and $Co^0$ species, respectively (Fig. 1c). These observations strongly suggested the co-existence of both $Co^0$ and in situ formed $Co_2C$ on the catalyst surface, which was further confirmed by detailed electron microscopic observations. Scanning transmission electron microscopy (STEM) coupled with elemental color mapping image of the spent CoMnCs/C catalyst unambiguously pointed to the surface enriching of carbon species around Co nanoparticles (Fig. S5). This was further supported by line-scanning of an individual nanoparticle, showing relatively even distributions of Cs and Mn across the nanoparticles but much higher C signal at the periphery (Fig. 1d). As carburization gradually accumulated, Co particles and the carbon layer underwent further reaction under reaction conditions and were gradually and partially converted into $Co_2C$. To validate this hypothesis, HRTEM image of the spent catalyst was acquired (Fig. 1e). Lattice fringe analysis revealed the structure of fcc-Co(111) in an individual particle decorated by small $Co_2C$ ensembles (Fig. 1e). Noted that $Co_2C$ was present as small patches on the Co surface, forming $Co_2C$-Co interphases, but did not fully cover the latter. To understand the individual roles of Mn and Cs in the catalyst structure, the STEM with line-scanning images of Co/C, CoMn/ C, and CoCs/C catalysts, together with their corresponding HRTEM images, were acquired (Fig. S6). These results showed that the addition of Mn alone also promoted the clear carbon enrichment on Co

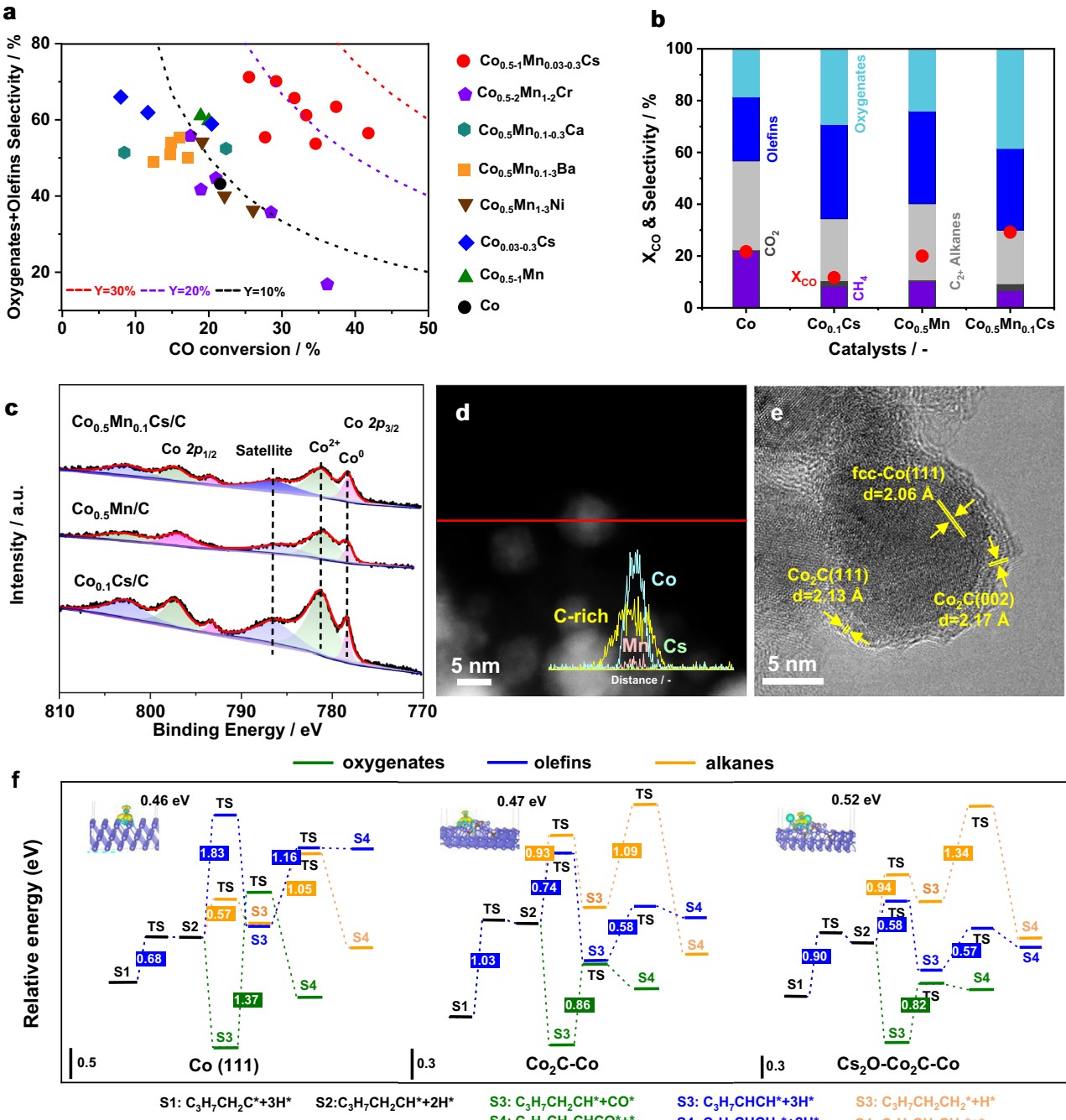

**Fig. 1 | Design of efficient syngas-to-oxygenates/olefins catalysts. a** CO conversion vs. total selectivity of alcohols and olefins on carbon-supported Co catalysts with different promoters. ($Co_xM_yN$: M and N represent metal additives; x and y, respectively, represent the loading amounts of additives M and N, catalysts preparation for more details) **b** Comparison of the syngas-to-oxygenates/olefins performance on representative Co-based catalysts. Reaction conditions: 210 °C, 3.0 MPa, $H_2/CO$ = 2.0, $F$ = 65 ml $min^{-1}$ (See Table S1 for details). **c** Co $2p$ XPS spectra of spent $Co_xMn_yCs/C$ catalysts. **d**, **e** STEM and HRTEM images of $Co_{0.5}Mn_{0.1}Cs/C$ catalyst. **d** Line scanning profiles (inset). **f** The reaction coordinate of $C_3H_7CH_2C^*$ desorption, hydrogenation, and CO insertion pathways on Co(111), $Co_2C$-Co, and $Cs_2O$-$Co_2C$-Co models. Bader charge analysis was shown in the inset.

particles and in situ formation of $Co_2C$-Co structure during FTS reaction, which were not observed on Co/C and CoCs/C. Considering that Cs was mainly enriched on Co nanoparticles for the best-performing CoMnCs/C catalyst (potentially existing as $Cs^+$ species due to the nature of electronic promoter, also supported by DFT calculations in Table S3), the $Cs_2O$-$Co_2C$-Co model was constructed for further simulation. In addition, statistic counting of TEM images revealed that the particle size of CoMnCs/C was significantly reduced as compared with the other two individually promoted ones ($D$ = 6.9 vs. 8.0 and

8.6 nm, Fig. S7), thus the increased dispersion might thereby increase the exposed Co and $Co_2C$ active sites and improve the catalytic activity. The interfacial sites of $Co_2C$ and Co nanoparticles, denoted hereafter as $Co_2C$-Co, are crucial for the preferential formation of alcohols[33,34], and the addition of Cs can further enhance alcohol formation. Based on these observations and the catalyst characterization data, a new catalyst model of $Cs_2O$-$Co_2C$-Co, plus Co(111) and $Co_2C$-Co, was constructed to further elucidate their effects on the syngas-to-olefins/oxygenates activity and product distribution through DFT

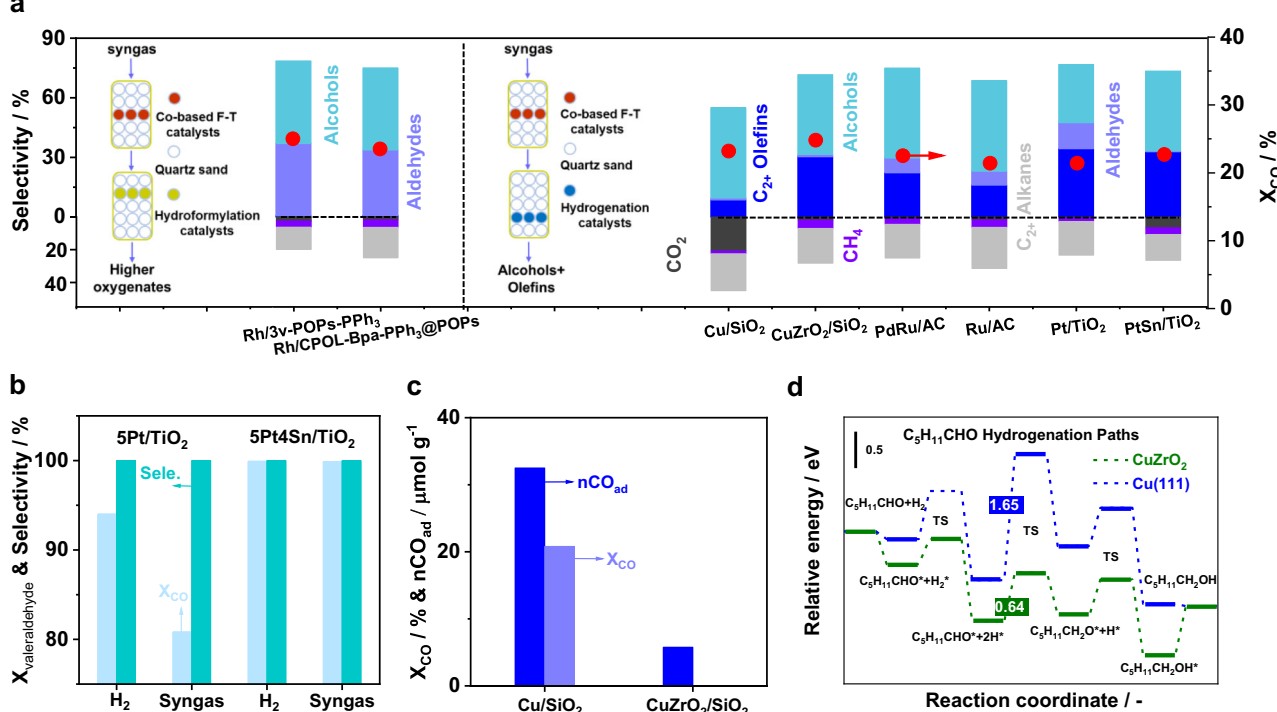

**Fig. 2 | Compatibility assessment between syngas-to-oxygenates/olefins and the downstream hydroformylation or hydrogenation processes. a** Catalyst performances between syngas-to-oxygenates/olefins plus hydroformylation of olefins (left) and syngas-to-oxygenates/olefins plus hydrogenation (right). Reaction conditions: 210 °C (syngas-to-oxygenates/olefins) and 140 °C (hydroformylation/hydrogenation), $H_2/CO = 1.5$, 3.0 MPa, $F = 65$ ml min$^{-1}$. **b** Catalytic performances of 5Pt/TiO$_2$ and 5Pt4Sn/TiO$_2$ for valeraldehyde hydrogenation in H$_2$ and syngas streams.

Reaction conditions: 5Pt/TiO$_2$ or 5Pt4Sn/TiO$_2$, 1.5 g; $T = 140$ °C; $P = 3.0$ MPa; $F$ (valeraldehyde) $= 0.01$ ml min$^{-1}$; $F$ (H$_2$) or $F$ (H$_2$/CO = 2) $= 30$ ml min$^{-1}$. **c** WGS performances and CO uptakes of CuZnAl, Cu/SiO$_2$, and CuZrO$_2$/SiO$_2$. Reaction conditions: catalysts 1.5 g; $T = 140$ °C; $P = 3.0$ MPa; $F$ (H$_2$O) $= 0.01$ ml min$^{-1}$; $F$ (CO) $= 30$ ml min$^{-1}$. **d** Valeraldehyde hydrogenation reaction pathways on Cu(111) and CuZrO$_2$ model system.

simulations (Fig. S8, Table S3). To explain the different product distributions, three key fundamental steps, including the formation of alkanes, olefins, and oxygenates from alkylidyne species ($C_3H_7CH_2C^*$), the presumed key reaction intermediates, were simulated (Fig. 1f, S8–S11, and Table S3). For Co(111), the energy barrier for the formation of alkanes from $C_5H_9^*$ is much lower than those for olefins and oxygenates (1.05 vs. 1.83, 1.37 eV), explaining the high propensity of deep hydrogenation on pure metallic cobalt. In contrast, the $C_5H_9^*$ intermediates are more prone to forming $C_3H_7CHCH_2^*$ or undergoing CO insertion instead of hydrogenation to alkanes on $Co_2C$-Co and $Cs_2O$-$Co_2C$-Co, suggesting that $Co_2C$-Co interphase can generally enhance the formation of olefins and oxygenates. To further reveal the promotional role of Cs, the energy barrier differences between $C_5H_9^*$ hydrogenation to alkanes and olefins, CO insertion were calculated and used as a selectivity indicator. A larger difference was observed on $Cs_2O$-$Co_2C$-Co than on $Co_2C$-Co (0.44 vs. 0.06 eV), explaining the improved selectivity towards oxygenates/olefins on the former. To further understand the promotional effects of Cs addition on syngas-to-oxygenates/olefins activity, we calculated the projected density of states (PDOS) for the different models (Fig. S12). It was found that the $3d$ occupied and unoccupied electronic states of Co in Co(111) are significantly localized near the Fermi level. This indicated that the $3d$ electrons of Co are not easily engaged in strong interactions with the $2p$ orbital electrons of the adsorbed CO, due to a mismatch in their energy levels. In comparison to the $Co_2C$-Co model, the incorporation of Cs markedly broadens the distribution of Co's $3d$ orbitals, particularly promoting the extension of the $3d$ orbitals toward lower energy levels. This effectively enhances the interaction between the C $2p$ and Co $3d$ orbitals, facilitating CO activation as also supported by the Bader

charge analysis (Fig. 1f). Therefore, the combined experimental and theoretical studies provide a molecular-level understanding of the roles of Cs and Mn in regulating the electronic properties of the active sites in Co-based catalysts, achieving directional modulation over the syngas-to-oxygenates/olefins reaction steps through precise engineering of the active $Cs_2O$-$Co_2C$-Co architectures that efficiently suppress excessive hydrogenation and facilitates direct CO insertion, thus enhancing the selectivity of oxygenates/olefins.

**Interplay between different catalytic modules and compatibility**

Since substantial amounts of olefins (25–40%) and aldehydes (10–20%) were produced during the syngas-to-oxygenates/olefins (Table S1), it is advantageous to convert them into $C_{4+}$ alcohols by employing additional heterogeneous hydroformylation and hydrogenation steps. To this end, the compatibility of the combined process between syngas-to-oxygenates/olefins and olefins hydroformylation/aldehydes hydrogenation was investigated. Two porous organic polymers (POPs)-supported single-Rh-site catalysts (Rh$_1$/3v-POPs-PPh$_3$ and Rh$_1$/POPs-BP&PPh$_3$, PXRD, ac-HAADF-STEM, Figs. S13, S14) with strong coordination bonds between Rh and exposed P embedded into the framework of POPs (XPS, Fig. S15) were selected as the heterogeneous hydroformylation candidates owing to their documented hydroformylation performance[35,36]. The performance of the catalysts in syngas-to-oxygenates/olefins plus hydroformylation of olefins showed comparable CO conversion and product distribution (Fig. 2a). Both alcohols and aldehydes were the major products, while selectivity of olefins was below 1%, suggesting successful transformation of the formed olefins into aldehydes through hydroformylation chemistry. In addition, different hydroformylation catalysts can effectively regulate

the normal and isomeric ($n/i$) ratio of alcohols and aldehydes (Table S4).

Selective hydrogenation process is particularly crucial for the catalysis system of syngas-to-oxygenates/olefins plus hydrogenation of oxygenates and olefins, as the presence of CO and the water generated from syngas-to-oxygenates/olefins could significantly impact hydrogenation performance[37]. In addition to the poisoning effect of CO on hydrogenation catalysts, other side reactions could occur, including generations of $CH_4$ and methanol, and water–gas shift reaction[38,39]. To identify potential candidates for the selective hydrogenation of aldehydes, a broad screening of Cu-, Ru-, Pd-, and Pt-based hydrogenation catalysts was performed (Fig. 2a). Several trends emerged among the different catalyst families. Pd- and Ru-based catalysts displayed lower activity in the hydrogenation of aldehydes compared to previous studies[40,41], likely due to CO poisoning. Similarly, the pristine $Pt/TiO_2$ showed minimal activity, while the Sn-modified $Pt/TiO_2$ demonstrated good performance in the selective hydrogenation of aldehydes to alcohols. The excellent performance could be originated from the electronic effect of Sn, leading to markedly increased electron density around Pt species (Fig. S16, XPS)[42]. Lastly, Cu-based catalysts displayed high hydrogenation activity in the CO-rich feeds, although product selectivity varied markedly. Olefin hydrogenation was significantly enhanced, accompanied by boosted formation of alkanes and $CO_2$ on $Cu/SiO_2$, pointing to vigorous hydrogenations of both aldehydes and olefins. In stark contrast, $CuZrO_2/SiO_2$ exhibited excellent selective hydrogenation performance in the complex feeds, fully consuming aldehydes while inhibiting hydrogenation of olefins.

The comprehensive screening identifies $5Pt4Sn/TiO_2$ and $CuZrO_2/SiO_2$ as promising hydrogenation candidates of aldehydes for the integrated system. The compatibility between syngas-to-oxygenates/olefins plus hydrogenation processes of aldehydes over the above identified catalysts was further understood by different probe reactions. Firstly, valeraldehyde hydrogenation with pure $H_2$ and syngas was applied as model reactions (Fig. 2b). It was found that the addition of Sn could avoid CO poisoning and improve the activity of $5Pt/TiO_2$, while comparable performances were observed on the Cu-based catalysts (Fig. S17). The $Cu/SiO_2$ showed high reactivity, but $CuZrO_2/SiO_2$ was basically inactive, likely due to the different electronic states of the active Cu species. This hypothesis was corroborated by CO pulse experiments, showing significantly decreased CO adsorption on $CuZrO_2/SiO_2$ as compared with those on the other catalysts (Fig. 2c). The interfacial structures of small Cu nanoparticles on $ZrO_2(111)$, as indicated by combined PXRD, XPS, and HRTEM analyses (Figs. S18–S20), were adopted as the catalyst model for the DFT investigations. Comparison of the reaction coordinate of aldehyde hydrogenation pathways between $Cu(111)$ and $CuZrO_2$ models revealed the much lower energy barriers of 0.64 eV on $CuZrO_2$ instead of 1.65 eV on $Cu(111)$ (Fig. 2d and S21, S22), agreeing well with the favorable formation of alcohols on $CuZrO_2$. These thorough studies thus lay a solid foundation for the design of more complicatedly integrated processes.

## Process design for an integrated catalysis system

Having successfully converted syngas to olefins and oxygenates by using $Cs_2O$-$Co_2C$-Co catalysts, a more advanced integrated technology was developed, aiming to efficient and precise regulating over the reaction pathways for direct syngas-to-$C_{4+}$ alcohol synthesis. This innovative approach employs cascaded heterogeneous hydroformylation and hydrogenation catalysis within a single fixed-bed reactor in the downstream of the syngas-to-oxygenates/olefins reactor. Our strategic combination of different catalytic modules allows the elimination of complicated separations from syngas to the end products. To validate this concept, two chain reactors packing with three well-designed catalysts of specific functions were designed to perform the selective conversion of syngas into $C_{4+}$ alcohols under conditions of $H_2/CO = 1.5$, $F = 130$ ml min$^{-1}$, 210 °C (syngas-to-oxygenates/olefins), 140 °C (hydroformylation and hydrogenation), and 3 MPa (Fig. 3a and Tables S5, S6). Notably, the cascade catalysis featured by $Co_{0.5}Mn_{0.1}Cs/C|Rh_1/3v$-POPs-PPh$_3$|$CuZrO_2/SiO_2$ displayed the highest alcohol selectivity of 81% at 15% CO conversion. Alcohols accounted for 95% of the liquid products, with the shares of $C_{4+}$ and $C_{6+}$ alcohols reaching 90% and 44%, respectively. Replacing $Co_{0.5}Mn_{0.1}Cs/C$ with a carrier-modified catalyst CoMnCs/MC further improved the performance. The CoMnCs/MC|Rh$_1$/POPs-BP-PPh$_3$|$CuZrO_2/SiO_2$ integrated system displayed 73% alcohol selectivity with $C_{4+}$ and $C_{6+}$ alcohols reaching 69% and 50%, respectively, at 25% CO conversion, corresponding to an unprecedented space-time yield (STY based on syngas-to-oxygenates/olefins) of $C_{4+}$ alcohols of 112 g·kg$_{cat}^{-1}$ h$^{-1}$.

Given the promising results on $Co_{0.5}Mn_{0.1}Cs/C|Rh_1/3v$-POPs-PPh$_3$|$CuZrO_2/SiO_2$, a systematic examination of the key process parameters including $H_2/CO$ ratio, gas hourly space velocity (GHSV), bed temperature, and pressure was further conducted (Tables S7–S10). The $H_2/CO$ ratios significantly influenced CO conversion and the selectivity of by-products (Table S7). Higher CO conversion was achieved with the increase of $H_2/CO$ ratios, resulting in a slight reduction in $CO_2$ selectivity and an increase in alkane selectivity. In contrast, $C_{4+}$ alcohol selectivity remains largely unaffected (68–75%) across a wide range of $H_2/CO$ ratios. This stability in alcohol selectivity is particularly advantageous for potential applications involving the conversion of biomass- or coal-derived syngas. The impact of temperature on the integrated catalysis was examined at an $H_2/CO$ ratio of 1.5 (Table S8). CO conversion significantly increased from 15% to 72% when the temperature was varied within 210–240 °C, accompanied by a notable decrease in alcohol selectivity from 81% to 11%, primarily due to over-hydrogenation as evidenced by the substantial formation of alkanes. Nevertheless, several promising results were observed at lower temperatures. For example, a high $C_{4+}$ alcohol selectivity of 73% was achieved at 210 °C, while the STY of $C_{4+}$ alcohols and total alcohols attained 119 g·kg$_{cat}^{-1}$ h$^{-1}$ and 143 g·kg$_{cat}^{-1}$ h$^{-1}$ at 230 °C, respectively. The impact of GHSV was further examined (Table S9). While CO conversion consistently decreased, the alcohol selectivity remained relatively stable and witnessed an apparent drop by 6% at 325 ml min$^{-1}$. The pressure significantly influenced catalytic performance (Table S10). CO conversion increased markedly from 1 to 3 MPa, leveled off at 8 MPa. Concurrently, $C_{4+}$ alcohol selectivity increased substantially from 60% to 80%, while olefin selectivity decreased from 11 % to negligible levels. Additionally, the ratio of normal to isomeric ($n/i$) alcohols decreased from 7 to 3 as pressure increased from 2 to 8 MPa. These results suggested that higher pressure favored the formation of branched aldehydes, leading to their subsequent hydrogenation into the corresponding isomeric alcohols. Finally, the long-term stability of the developed $Co_{0.5}Mn_{0.1}Cs/C\,|\,Rh_1/3v$-POPs-PPh$_3\,|\,20CuZnO_2/SiO_2$ integrated catalytic system was evaluated under the reaction conditions of $H_2/CO = 1.5$, $F = 130$ ml min$^{-1}$, 210 °C (syngas-to-oxygenates/olefins), 140 °C (hydroformylation and hydrogenation), and 3 MPa, and a firmly stable performance for more than 130 h was attained (Fig. 3b). ASF distribution clearly pointed to the favorable formation of $C_{4+}$ alcohols with an overall $n/i$ ratio of 6 (Fig. 3c, d, Fig. S3). In the integrated system, the $C_1$–$C_3$ products strongly deviate from the ASF linear distribution, and the chain growth probability $\alpha$ increases from 0.65 to 0.68, indicating that the hydroformylation process effectively modulates product distribution (Fig. S3). As the reaction pressure increased from 3.0 to 8.0 MPa, the deviation of $C_1$-$C_3$ products from the ASF linear distribution became increasingly pronounced (Fig. S3e, f), while the chain growth probability $\alpha$ rose from $0.68 \pm 0.01$ to $0.72 \pm 0.01$, demonstrating that elevated pressure enhances carbon chain elongation and suppresses the formation of low-carbon products. Key performance descriptors of this integrated process were

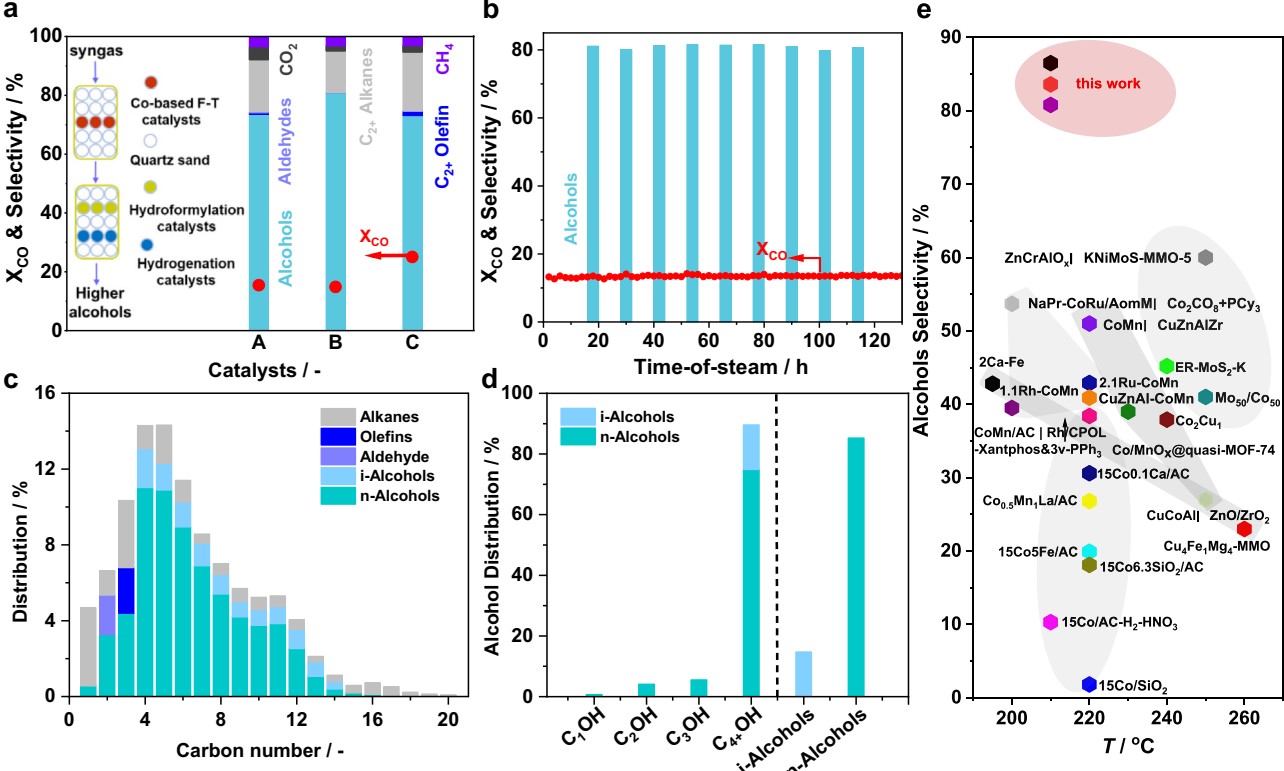

**Fig. 3 | Design of an integrated strategy for the conversion of syngas to $C_{4+}$ long-chain alcohols. a** Catalytic performances of (A) CoMnCs/C|Rh$_1$/3v-POPs-PPh$_3$| 5Pt4Sn/TiO$_2$, (B) CoMnCs/MC|Rh$_1$/POPs-BP&PPh$_3$|CuZrO$_2$/SiO$_2$ and (C) CoMnCs/C| Rh$_1$/3v-POPs-PPh$_3$|CuZrO$_2$/SiO$_2$ (see Supplementary Table S6 for more details). **b** Stability test of CoMnCs/C|3v-POPs-PPh$_3$|CuZrO$_2$/SiO$_2$. Reaction conditions: 210 °C (syngas-to-oxygenates/olefins) and 140 °C (hydroformylation/hydrogenation), 3.0 MPa, H$_2$/CO = 1.5, F = 130 ml min$^{-1}$. **c** Carbon number distributions of the products. **d** Alcohol distributions and the ratios between normal and isomeric alcohols. **e** Comparison on alcohol selectivity between this work and the previous literature results (see Supplementary Table S5 for more details).

compared with the previously reported syngas-to-$C_{4+}$ alcohol conversion systems (Fig. 3e and Table S5), demonstrating the superiority in selectivity of $C_{4+}$ (80% *vs.* 45%) and $C_{6+}$ alcohols (50% *vs.* 33%), and productivity (*STY* of total, $C_{4+}$ and $C_{6+}$ alcohols reaching 143, 132, and 119 g·kg$_{cat}^{-1}$·h$^{-1}$, respectively). These excellent results can be well reproduced with minor variations (CO conversion within 14 ± 1% and alcohol selectivity within 78 ± 3%, Table S11), thus demonstrating the great potential for industrial applications.

### Comparative analysis of syngas-to-$C_{4+}$ alcohol synthesis pathways

Currently, the production of $C_{4+}$ alcohols from syngas primarily relies on two routes: the heterogeneous hydroformylation and hydrogenation of $C_{3+}$ olefins obtained downstream of Fe-based Fischer-Tropsch synthesis, and the higher alcohol synthesis directly from syngas (Fig. 4a). Our proposed process employs two fixed-bed reactors to achieve an integrated, multi-stage conversion of syngas into $C_{4+}$ alcohols with unprecedented selectivity (~80%), alongside 10% $C_1$-$C_3$ short-chain alcohols and 8% $C_{5+}$ alkanes. Compared to the existing routes, this approach significantly reduces the need for product separation, distillation, and reactor units (Fig. S23). Furthermore, it can optimize waste heat utilization in industrial applications, resulting in substantial energy savings and promoting the sustainable development of novel chemical processes. The diversification and value-added nature of the products further stabilize market value, offering substantial competitive advantages.

Figure 4b compares $CO_2$ selectivity across the three routes. In Route 1, Fe-based FTS generates 20–40% $CO_2$ during the synthesis of $C_{3+}$ olefins, which are subsequently converted into $C_{4+}$ alcohols via heterogeneous hydroformylation and hydrogenation. Route 2

employs Rh-based or modified methanol synthesis catalysts for direct HA synthesis, producing less $CO_2$ (1-10%) but achieving limited carbon chain growth, with only 5−10% $C_{4+}$ alcohols. Modified Mo-based catalysts produce even higher $CO_2$ (~30%) while predominantly generating $C_1$-$C_3$ alcohols. In contrast, our approach achieves exceptional $C_{4+}$ alcohol selectivity (~80%) with $CO_2$ selectivity as low as ~1%, which has ultra-high carbon-efficiency (>95%, carbon-atom utilization efficiency from CO to organic products other than $CH_4$) by minimizing $CH_4$ and $CO_2$ formation and dramatically reduces industrial production costs. By coupling residual $CO_2$ with green hydrogen via the reverse water-gas shift reaction to recycle syngas, the process might approach net-zero $CO_2$ emissions, significantly reducing greenhouse gas outputs. Additionally, the catalyst demonstrates an oxygen-efficiency (oxygen-atom utilization efficiency from CO to organic products) of 18%, a substantial improvement over the HA synthesis and the industrial FTS paraffin production process (Fig. S24), where oxygen efficiency is just 6% and 1%, respectively (Fig. 4c). This enhanced efficiency reduces CO costs and industrial wastewater generation, mitigating environmental hazards.

## Discussion

The catalytic system presented in this study demonstrates a meaningful advance in the conversion of syngas to higher alcohols. The results show that careful catalyst design and process integration can substantially shift the product distribution toward $C_{4+}$ long-chain alcohols by precise control of the reaction channel. Under optimized conditions of (H$_2$/CO = 1.5, F = 130 ml min$^{-1}$, 210 °C for syngas-to-oxygenates/olefins, 140 °C for hydroformylation/hydrogenation), the system achieved up to 80% selectivity to $C_4^+$ alcohols with only 1% $CO_2$ selectivity, 17% CO conversion, and 97% overall carbon efficiency. The

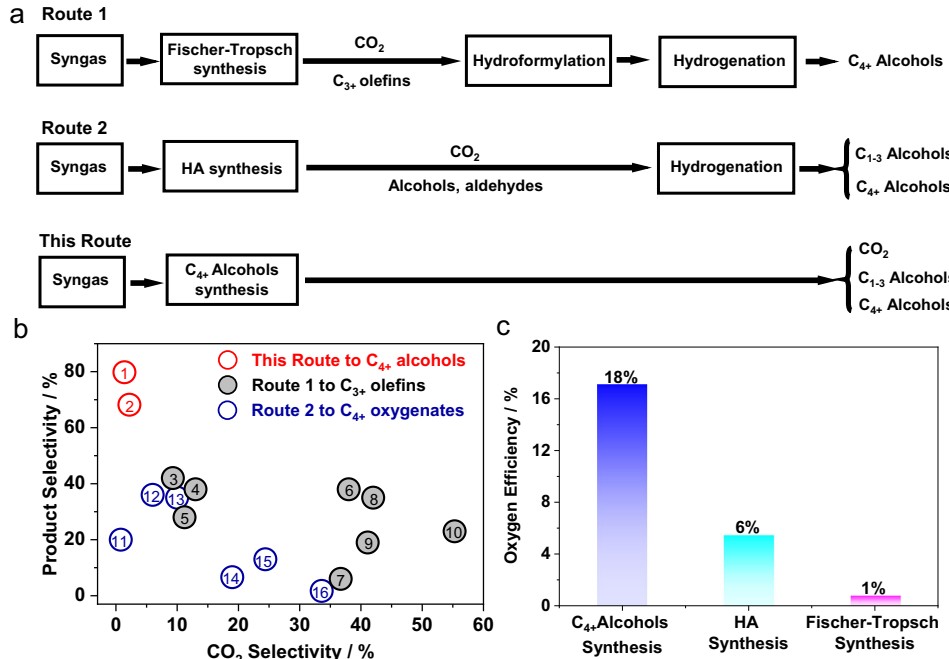

**Fig. 4 | Comparison of the typical processes of $C_{4+}$ alcohol synthesis from syngas. a** Different reaction routes. **b** Comparison on the $C_{4+}$ alcohol and $CO_2$ selectivity from different technologies. (red, $C_{4+}$ alcohols synthesis Route: $C_{4+}$ alcohols selectivity, 1.CoMnCs/C|Rh$_1$/3v-POPs-PPh$_3$|CuZrO$_2$/SiO$_2$, 2. CoMnCs/MC| Rh$_1$/POPs-BP&PPh$_3$|CuZrO$_2$/SiO$_2$; black, Route 1: $C_{3+}$ **olefins selectivity**, 3.Mn-χ-Fe$_5$C$_2$[44], 4.FeMn@Si[45], 5.χ-Fe$_5$C$_2$[44], 6.1.5Na-Fe$_1$Zn$_1$[46], 7.Fe-Al$_2$O$_3$ (SCS350)[47], 8.Na$_2$S-Fe-CNF[44], 9.Fe-Al$_2$O$_3$(SCP)[47], 10.Fe$_3$O$_4$/MAG[48]; blue, Route 2: **$C_{4+}$ oxygenates selectivity**, 11.Co/MnO$_x$@quasi-MOF-74[49], 12.CoMn|CuZnAlZr[25], 13.Co$_4$Mn$_1$K$_{0.1}$[50], 14.ZnCrAlO$_x$|KNiMoS-MMO-5[28], 15.Cu$_4$Fe$_1$Mg$_4$-MMO[20], 16.ER-MoS$_2$-K[51]. See Supplementary Table S5 for more details). **c** Comparison on the oxygen-efficiency between this process and the Fischer-Tropsch synthetic paraffin process.

in situ reconstructed Cs$_2$O-Co$_2$C-Co active sites play a crucial role in directing the steps and intermediates toward oxygenates/olefins. Cs modification enhances Co dispersion, favoring olefin desorption and CO insertion. In the subsequent stages, by precisely regulating the reaction channels, single-site Rh catalysts enable efficient olefin-to-aldehyde conversion while suppressing excessive hydrogenation, and CuZrO$_2$/SiO$_2$ catalysts effectively hydrogenate aldehydes under syngas conditions, mitigating CO poisoning and minimizing side reactions. The strategic cooperative catalysis illustrates how rationally matching three distinct catalytic modules under temperature-optimized conditions can achieve both high alcohol selectivity and operational stability. Rather than representing a complete departure from ASF-type behavior, this work provides a practical pathway to enhanced higher-alcohol selectivity through precise active-site engineering, staged reactor operation, and catalyst compatibility. We believe these findings offer valuable guidance for designing integrated catalytic systems for the sustainable and scalable production of long-chain alcohols from syngas.

## Methods
### Catalysts preparation
**Co-based syngas-to-oxygenates/olefins catalysts.** The modified Co/C catalysts were prepared by incipient wetness impregnation method. According to the contents of active component of Co and additives of Mn, Cs, Ca, Ba, Cr and Ni, series of mixed solution of Co(NO$_3$)$_2$·6H$_2$O (Aladdin, 99.99%) and Mn(CH$_3$COO)$_2$·6H$_2$O (Aladdin, 99.99%), Ca(NO$_3$)$_2$·6H$_2$O (Aladdin, 99.99%), Ba(NO$_3$)$_2$·6H$_2$O (Aladdin, 99.99%), Cs(NO$_3$)$_2$·6H$_2$O (Aladdin, 99.99%), Cr(NO$_3$)$_2$·6H$_2$O (Aladdin, 99.99%) and Ni(NO$_3$)$_2$·6H$_2$O (Aladdin, 99.99%), respectively, were prepared to add into the carbon (Brilliant Tech Co. Ltd.). Loading of Co and the other metal additives was 15 wt% and 0.03–3 wt%, respectively (Co$_x$M$_y$N: M and N represent metal additives; x and y respectively represent the loading amounts of additives M and N Catalysts include: 15Co, 15Co$_{0.03}$Cs, 15Co$_{0.1}$Cs, 15Co$_{0.3}$Cs, 15Co$_{0.5}$Mn, 15Co$_1$Mn,

15Co$_{0.5}$Mn$_{0.1}$Cs, 15Co$_1$Mn$_{0.1}$Cs, 15Co$_1$Mn$_{0.3}$Cs, 15Co$_{0.5}$Mn$_1$Ni, 15Co$_{0.5}$Mn$_2$Ni, 15Co$_{0.5}$Mn$_3$Ni, 15Co$_{0.5}$Mn$_{0.1}$Ca, 15Co$_{0.5}$Mn$_{0.3}$Ca, 15Co$_{0.5}$Mn$_2$Cr, 15Co$_{0.5}$Mn$_1$Cr, 15Co$_1$Mn$_1$Cr, 15Co$_1$Mn$_2$Cr, 15Co$_2$Mn$_2$Cr, 15Co$_{0.5}$Mn$_{0.1}$Ba, 15Co$_{0.5}$Mn$_{0.3}$Ba, 15Co$_{0.5}$Mn$_{0.5}$Ba, 15Co$_{0.5}$Mn$_1$Ba, 15Co$_{0.5}$Mn$_2$Ba, 15Co$_{0.5}$Mn$_3$Ba). Then, the catalysts were dried at room temperature and transferred to the oven at 50 °C overnight. The catalyst precursors were further calcined in a flow of argon at 350 °C for 2 h. In addition, modified carbon carrier (MC) with enriched concentration of surface oxygen groups was prepared as follows. The conventional carbon carrier was immersed with 10% HNO$_3$ in a round-bottom flask and stirred at 60 °C for 6 h. After cooling to room temperature, the solids were filtered, dried, and used for the loading of active metals as stated previously. The Co-based syngas-to-oxygenates/olefins catalysts were reduced in situ in fixed bed reactor for 10 h at 430 °C, 2000 h$^{-1}$, 0.1 MPa, under an H$_2$ atmosphere before reaction. The catalyst composition for ICP characterization analysis is shown in Table S12.

**Rh-based heterogeneous hydroformylation catalysts.** The Rh-based heterogeneous hydroformylation catalysts supported on porous organic polymers (POPs) were prepared by the post-loading method. 3v-POPs-PPh$_3$ and POPs-BP&PPh$_3$ were previously reported to have been synthesized by the solvothermal polymerization method (see references for preparation method of polymer carriers[35,43]). For example, 0.35 g of 2v-BPa, 0.025 g of AIBN as an initiator, 10 mL of THF, and 0.65 g of 3V-PPh$_3$ were introduced to a 30 mL autoclave under an argon atmosphere. Following sealing, the solution was stirred for 1 h at room temperature and then left in an oil bath at 100 °C for 24 h. After the insoluble was filtered, washed with THF and dried under vacuum at 60 °C, the POPs-BP&PPh$_3$ could be obtained. The synthesis of 3v-POPs-PPh$_3$ involved adding 0.025 g of AIBN, 10 mL of THF, and 1 g of 3v-PPh$_3$ to a 30 mL autoclave under an argon atmosphere, and following the same steps.

1.2 g 3v-POPs-PPh$_3$ or POPs-BP&PPh$_3$ polymer carrier was added to 100 mL tetrahydrofuran (THF, 99%) containing 3.8 mg Rh(CO)$_2$(acac) under N$_2$ atmosphere. The obtained solution was stirred in and N$_2$ atmosphere at room temperature for 24 h. The catalysts were centrifugally separated, washed with excess THF, and dried under vacuum at 60 °C to obtain the Rh$_1$/3v-POPs-PPh$_3$ and Rh$_1$/POPs-BP&PPh$_3$ catalysts. Rh-based heterogeneous hydroformylation catalysts had no reduction treatment process. The catalyst composition for ICP characterization analysis is shown in Table S12.

**Hydrogenation catalysts.** The Sn-modified Pt/TiO$_2$ catalyst was prepared by the ultrasonic-assisted impregnation technique. Ethylene glycol (Aladdin, 99%), solution of stannous chloride (Aladdin, 99%) and aqueous solution of chloroplatinic acid (Aladdin, 99%) were mixed. Subsequently, a certain amount of TiO$_2$ (Aladdin, 99%) carrier was weighed and dispersed in the above mixed solution, and ultrasound was performed for 30 min. It was dried overnight in an oven at 80 °C and then calcined in a muffle furnace at 450 °C for 4 h (ramping 10 °C min$^{-1}$). The Pt-based hydrogenation catalysts were reduced in situ in fixed bed for 4 h at 450 °C and 2000 h$^{-1}$ under H$_2$ atmosphere before reaction.

Cu/SiO$_2$ and CuZrO$_2$/SiO$_2$ catalysts were prepared by the ammonia evaporation method. The 0.3125 M Cu(NO$_3$)$_2$·3H$_2$O (Aladdin, 99%) solution was evenly stirred with SiO$_2$ (Aladdin, 99%) or ZrO$_2$ (Aladdin, 99%), and then precipitated with 25% ammonia solution (Aladdin, 99%) until the pH of the solution was 9–10, then heated at 80 °C and washed the precipitate with deionized water. The prepared precipitate was dried at 120 °C for 10 h, and finally calcined in a muffle furnace at 450 °C for 4 h (ramping 2 °C min$^{-1}$). Cu-based hydrogenation catalysts were reduced in situ in a fixed bed for 2 h at 260 °C, 2000 h$^{-1}$, 0.1 MPa under H$_2$ atmosphere before reaction. The catalyst composition for XRF characterization analysis is shown in Table S12.

RuPd/C and Ru/C catalysts were prepared by the incipient wetness impregnation method. According to the calculated content of active components Ru and Pd (RuCl$_3$·3H$_2$O, Aladdin, 99%; PdCl$_2$, Aladdin, 99%) a glycol solution (Aladdin, 99%) was prepared to impregnate the activated carbon carrier. After drying at room temperature, the catalyst was transferred to the oven at 50 °C for overnight drying, and then roasted at 350 °C in a flow of argon for 6 h to obtain the catalyst. The Ru-based hydrogenation catalysts were reduced in situ in fixed bed for 4 h at 450 °C and 2000 h$^{-1}$ under H$_2$ atmosphere before reaction.

## Catalyst evaluation

**Syngas to C$_{4+}$ long-chain alcohols.** The catalyst was evaluated in a self-made integrated system composed of two consecutive stainless steel fixed-bed reactors with a reactor length of 450 mm and an inner diameter of 9 mm (Fig. S25). In a typical test, 2 ml syngas-to-oxygenates/olefins catalyst was loaded into the first reactor, and 1 ml heterogeneous hydroformylation catalyst or/and 2 ml hydrogenation catalyst were placed to the second reactor, separated by quartz sand and quartz cotton. The pressure of the reactors was controlled by a back pressure valve located at the downstream of a cold trap after the second reactor, while the bed temperatures were tuned by individual temperature controllers. After stabilized for 24 h of reaction, the gaseous, liquid and solid products were periodically sampled for analysis. The CO conversion was calculated from the inlet and outlet gas flow rates measured by the mass flowmeter, and the product selectivity was calculated according to the composition of the gas, liquid and solid products. Agilent 7890A gas chromatograph with a Plot Q packed column and a HP-5 capillary column was used to analyze the product compositions. Exhaust gas was analyzed online using a thermal conductivity detector (TCD) and organic liquid products and solid were analyzed offline using a hydrogen flame ionization detector (FID). The secondary butanol as the internal standard was used to analyze

and calculate the components of the aqueous phase products. The organic phase products were calculated using the normalization method with corrected peak area. The typical GC spectra of different products were shown in Fig. S26. A comprehensive analysis, including liquid phases (organic and aqueous), solid wax, and gaseous components, leads to excellent carbon balances (generally within 100 ± 5%).

## Probe experiments

**Valeraldehyde hydrogenation.** Valeraldehyde hydrogenation was evaluated in a steel fixed-bed reactor with an inner diameter of 9 mm and a length of 380 mm. 2 ml hydrogenation catalyst was packed in the middle of the reactor and pre-reduced by H$_2$ under 260 °C (ramping rate 10 °C min$^{-1}$) at 0.1 MPa for 2 h. After cooling to 80 °C, H$_2$ or syngas (H$_2$/CO = 2/1) was introduced and raised to 140 °C and 3.0 MPa. Valeraldehyde (Aladdin, 99%) was then admitted with a syringe pump (flow rate of 0.01 mL min$^{-1}$) for the hydrogenation reaction. An Agilent 7890N GC with Plot Q packed column and HP-5 capillary column was used to analyze the products. Exhaust gas was analyzed online with a TCD detector, and organic liquid products were analyzed offline with an FID detector.

**Water-gas shift.** The water-gas shift (WGS) reaction was evaluated in the same setup as valeraldehyde hydrogenation. A similar prereduction catalyst was applied. Then WGS reaction was conducted by introducing N$_2$/CO mixture gas and water with a mass flow controller and syringe pump, respectively, under 140 °C and 3.0 MPa. The reaction products were analyzed on-line using an Agilent 7890N GC with a TDX-01 packed column and a TCD detector.

**Catalyst characterization.** PXRD measurements were carried out on a PANalytical X'Pert[3] powder X-ray diffractometer. All tests were performed using Cu K$\alpha$ radiation with operating conditions of 40 kV and 40 mA. The samples were first ground to a fine powder in a mortar and then tested on glass slides.

Elemental content testing of samples by the Inductively Coupled Plasma (ICP) technique on the Perkin Elmer ICP-OES 7300DV equipment. The metal content in the Cu-based catalysts was determined by X-ray fluorescence spectroscopy (XRF).

The HRTEM images of the catalysts were collected on the Tecnai G2 F30 S-Twin transmission electron microscope of FEI Company. The acceleration voltage was 300 kV, the line resolution was 0.1 nm, and the image point resolution was 0.2 nm.

The ac-HAADF-STEM and STEM images determination of catalyst samples were acquired on the JEM-ARM200F STEM/TEM (resolution of 0.08 nm) and JEM-F200 instrument of JEOL, respectively. The sample was dispersed in EtOH and placed onto Cu grids.

The CO pulse chemisorption experiment was carried out on the Zeton Altramira AMI-300 chemisorption instrument. About 100 mg samples was packed into a U-shaped tube using He as carrier gas at a flow rate of 30 mL min$^{-1}$. The catalyst sample was first heated to 120 °C at a heating rate of 10 °C min$^{-1}$ in a flow of He, purged for 1 h, and cooled to 50 °C. Then, 10% CO/He mixture was quantitatively pulsed injected until adsorption saturation, and the released CO was detected by a TCD detector.

The XPS tests were carried out on the VG Thermo Escalab 250Xi X-ray photoelectron spectrometer, using Al K$\alpha$ as excitation source, accelerating voltage of 15 kV and transmitting current of 10.8 mA.

**DFT calculation methods.** The theoretical calculation was completed by Vienna Ab Initio Simulation Package (VASP.5.4.1) within the density functional theory method. The standard PBE pseudopotential of each element in projected augment plane-wave (PAW) method was chosen. After an energy test convergence, the Cut-off energy was 450 eV. In order to obtain the much more accurate energy and electronic structure, the spin polarization was employed in all calculation. Gaussian

method combined with the electron broadening of 0.05 eV was applied. For all structural optimization and property calculations, the convergence thresholds of energy and force were $10^{-5}$ eV and $0.02$ eV Å$^{-1}$, respectively. Each transition state was found by the climbing image nudged elastic band (CI-NEB) method and further confirmed by the frequency analysis calculation. K-points sampling scheme in Brillouin zone adopted the Monkhorst-Pack automatic sampling method and the sampling density of K-point was 0.04 2*π/ Angstrom for each structure model, which met the energy convergence with K-points test.

## Data availability
The data presented in the figures of this paper are publicly available. Other supporting data are available from the corresponding authors upon request. Source data are provided with this paper.

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

## Acknowledgements

We greatly appreciate the financial support by the National Key Research and Development Program of China (No. 2023YFB4103100, Z.H.), the Strategic Priority Research Program of the Chinese Academy of Sciences (Grant No. XDA 29050300, Z. Ziang), Young Elite Scientists Sponsorship Program by CAST (No. 2023QNRC001, Z. Ziang), the State Key Laboratory of Catalysis (Grant No. 2024SKL-B-005, Z. Ziang), Liaoning Binhai Laboratory (Grant No. LBLD-2025-08, Z. Ziang), Zhejiang Provincial Natural Science Foundation of China (No. LQ21B030009, L.R.), Key Laboratory of the Ministry of Education for Advanced Catalysis Materials and Zhejiang Key Laboratory of Advanced Catalysis and Adsorption Materials (2025ZY01096, L.R.), and the Young Star of Science and Technology in Shaanxi Province (No. 2024ZC-KJXX-088, M.Y.), Industry-University-Research Project of Yulin Science and Technology Plan (2024-CXY-077, M.Y.). We also appreciate Oushisheng Company for the help with the catalyst evaluation apparatus.

## Author contributions

Yihui Li: Investigation, Data curation, Conceptualization, Formal analysis, Methodology, Visualization, Writing-original draft. Ziang Zhao: Supervision, Conceptualization, Writing-review & editing. Miao Jiang and Guoqing Wang: Hydroformylation catalyst provided. Zheng Li: Data curation. Wei Lu: Formal analysis. Wenhao Cui: STEM characterization. Rong Liu: Density functional theory calculations. Ronghe Lin: Writing-review & editing. Yu Meng: Density functional theory calculations. Yuan Lyu: Formal analysis, Investigation. Li Yan: Project administration, Resources, Design of the experimental hydroformylation system. Hejun Zhu: Project administration, Supervision, Validation, Writing-review & editing. Yunjie Ding: Supervision, Project administration, Writing-review & editing.

## Competing interests

The authors declare no competing interests.
