## [Transparent Peer Review file · Nature Communications]

Selective conversion of syngas to C4+ long-chain alcohols

Corresponding Author: Professor Yunjie Ding

Version 0:

Reviewer comments:

Reviewer #1

(Remarks to the Author)

This work reported the novel catalysis system for the conversion of syngas to C4+ long-chain alcohols. The system achieves a high selectivity of 80% C4+ long-chain alcohols with low CO₂ and CH₄ selectivities. Some interesting and important results were reported and discussed. However, the series connection of multiple catalysts causes an exceptionally complex reaction process, which may affect the industrial application prospects of the catalytic system. In addition, some conclusions have not received strong support and are not convincing. Therefore, there are some concerns that the authors should address well before further considering publication. The detailed comments are as follows:

1. The catalytic performance should be discussed with reaction conditions in the abstract and conclusion.
2. For a series of modified Co/C catalysts with different metal promoters (Mn, Cr, Ni, Ca, Ba, and Cs), the effect of promoters on selectivities of CO₂ and CH₄ should be discussed.
3. The author suggested that interfacial sites of Co@Co₂C structure were crucial for the preferential formation of alcohols. How to confirm that the catalyst forms a Co@Co₂C core-shell structure during the reaction process?
4. For DFT calculations, the detailed E, H, S, ZPE, G values for each IS, TS, and FS states should be listed in table. Also the coordinate files for optimized structures should be given in the supporting information.
5. Does the lifespan of three catalysts, which are coupled in this work, differ? Will the mismatched lifespan of different catalysts limit the further industrial application of this catalytic system?
6. I suggest the author also compare the STY of C4+alcohols with other catalytic systems reported in the literature.

Reviewer #2

(Remarks to the Author)

The authors report a nice study where they combine an optimized promoted Co-FT catalyst with a typical heterogeneous hydroformylation catalyst and a selective hydrogenation catalyst to convert syngas to C4+ alcohols with a remarkable selectivity at low CO conversions (below 15%) in a two-step process without intermediate separation. The low product yield is likely one of the barriers to implement this two-step process. Separation of the alcohols from water is costly –separation of water after the FT step is likely more efficient and indeed done industrially.

This is a nice experimental study and worth publishing. The experiments are supported by standard characterizations (XRD for some of the spent Co catalysts, XPS for some of the spent Co catalysts, TEM for the selected catalyst). BET, H₂ chemisorption and TPR could be included for the optimized Co catalysts.

Some DFT calculations are included for selected reactions on model surfaces (Co(111), Co island on Co₂C, and a Cu₁₃ cluster on ZrO₂). The DFT calculations add little scientific value to the paper since they do not consider the relevant active sites, surface coverages or the relevant reaction steps. Also the introduction of a Cs atom to describe the effect of the Cs promoter is incorrect. Cs is likely present as Cs₂O or Cs₂CO₃, and not all reactions will occur next to the promoter as proposed by the calculations. The quality of the paper would improve if the DFT calculations are not included.

On line 38, the authors mention iso-butanol as a valuable product, but the proposed process does not seem to produce significant amounts of isobutanol since the n/i selectivity in the hydroformylation is not controlled.

It would be interesting to include a Table with the state-of-the-art in C4+ alcohol synthesis in the SI to support the statement of ca. 20% on line 73.

The authors use subscripts to describe the catalysts (Co_{0.5}Mn_{0.1}Cs), but do not explain the subscripts. In the synthesis section, it is stated that all catalysts have 15 wt% Co loading. More details about the catalyst synthesis and composition (ICP?) could be included.

Evidence for interfacial sites on Co@Co₂C is not provided (line 147). Very little is known about the structure of the CoMnCs/C catalyst.

Reviewer #3

(Remarks to the Author)

The authors report an integration catalytic strategy for long-chain alcohol synthesis from syngas conversion using two separate reactors and a three-stage packed catalysts process, which realizes the tandem connection of syngas to long-chain oxygenates/olefins and the subsequent hydroformylation/hydrogenation to produce C₄₊ alcohols. The experiment achieves awfully excellent catalytic performance with 80% C₄₊ alcohols selectivity and 1% CO₂ selectivity that surpasses the previous reports of the related syngas conversion to alcohols. Generally, the present work is featured by process integration and the optimized combination of several different types of functional catalysts, which is not significantly innovative with respect to the design concept because similar tandem patterns have been commonly reported. Besides, the author's discussion about the cognition of the cooperative catalysis between different modules is somewhat general. The specific comments or queries are listed as follows.

(1)The authors presume that the CoMnCs/C catalysts used for syngas conversion to oxygenates/olefins possess the interfacial structure of CoCs@Co₂C, plus Co(111) and Co@Co₂C, which lacks characterization evidence. Does CoCs @ Co₂C refer a core-shell structure? However, it was not obviously observed in the TEM image (Fig S4). In addition, this model also ignores the role of manganese promoters.

(2)Personally, I think it is not of great significance to discuss the ecological benefits of the integrated catalytic strategy for syngas conversion in this work because the results are only obtained based on the small laboratory experiments without pilot or benchmark test.

(3)In the text (Line 128 ~ 132), the authors mentioned that the single Mn and Cs modified Co-based catalysts would increase the selectivity of ROH and Olefins, but would reduce CO conversion. For example, the CO conversion over Co_{0.5}Mn / C and Co_{0.1}Cs / C were 20 % and 11 %, respectively, but that was increased to 29 % on Mn and Cs co-modified Co-based catalysts. Why?

(4)The space-time yield of C₄₊ alcohols was calculated based on the single CoMnCs/C catalyst. This calculation method might be not reasonable because the other two modular catalysts also synergistically promote the formation of C₄₊ alcohols products.

(5)The specific composition of the hydrogenation catalysts should be provided for deep understanding of the catalyst.

(6)In the section of Catalyst evaluation, the pretreatment of the catalyst (e.g. reduction method) should be described in detail, because catalyst pretreatment has a great influence on the reaction performance. Moreover, in the characterization measurements, the authors should make clear how the samples were treated? What is the state of these samples (oxidic or reduced state, fresh or spent)?

(7)In Fig 2b, it demonstrated CO (not valeraldehyde) conversion. This might be written error because the image is related the probe test of valeraldehyde hydrogenation.

(8)From Table S6, it can be seen that the CO₂ selectivity of Co_{0.5}Mn_{0.1}Cs / C | Rh1 / 3v-POPs-PPh₃ | CuZrO₂ / SiO₂ should show an upward trend with the increase of reaction temperature. Why is the CO₂ selectivity at 230°C lower than 220°C?

Version 1:

Reviewer comments:

Reviewer #2

(Remarks to the Author)

The authors have included extra information to support their results. The HRTEM images are particularly helpful.

The DFT calculations remain problematic.

The HRTEM image (Figure 1e) suggests Co₂C islands are formed on metallic Co. The DFT model is the inverse: Co island on Co₂C.

Recent studies have conclusively shown that K is present as K₂O during FT, and the O atoms play a crucial role in the promotion. The model with a single Cs atom (not Cs⁺) and the active site next to it is not representative for the actual sites. Experimental and modeling studies have shown that alkylidyne species are the key intermediates, not olefins.

If metallic Co sites are available, olefin hydrogenation will still dominate over CO insertion and olefin desorption, because of the much lower barrier.

The selectivity for Co/C is unusual. Experimental studies have shown that olefins are the primary products of Co-FT, the high selectivity to methane and oxygenates, and the high alkane/olefin ratio in Fig 1b should be compared to literature.

The grammar should be improved. Words are missing in some of the new sentences in the revised manuscript.

Reviewer #3

(Remarks to the Author)

In the revised manuscript, the authors well replied to the queries and made careful modification. So it may meet the publication requirements.

Reviewer #4

(Remarks to the Author)

I have reviewed the revised manuscript submitted by Li et al., along with the authors' responses to the comments raised in the original round of peer review. Since I have joined the peer review process at this stage, I have taken into account both the content of the revised submission and the earlier reviewer remarks and authors' replies.

The authors present a dual-reactor, single-pass system for the conversion of syngas to higher alcohols. While the emphasis on C₄₊ alcohols rather than the more commonly targeted C₂₊ or C₃₊ alcohols is noted, I do not consider this shift in focus to represent a significant conceptual advance. Selectivity toward higher alcohols—except for ethanol, which cannot form via olefin hydroformylation—remains fundamentally constrained by the statistical nature of the Anderson–Schulz–Flory (ASF) product distribution. This is expected given that all alcohol products, either directly or indirectly, originate from FTS hydrocarbons formed on the surface of the first (Co-based) catalyst.

In several ways, this study appears as an extension of the authors' earlier work (Li et al., ACS Catal. 2021, 11, 14791–14802; <https://doi.org/10.1021/acscatal.1c04442>), where a similar catalytic pairing, i.e., CoMn/AC for FTS and a Rh-based polymer catalyst for hydroformylation, was implemented within a single reactor using two packed-beds. In my opinion, the above article should have been cited in the current manuscript.

The primary distinctions in the present manuscript, relative to the earlier work are: (1) the addition of a third catalyst to hydrogenate aldehydes into alcohols, shifting the major oxygenate product class toward alcohols exclusively, and (2) the segmentation of the process into two reactors, allowing independent temperature control (210 °C for FTS and 140 °C for hydroformylation/hydrogenation), but clearly deviating from the concept of tandem operation of catalysts and reactions.

The experimental work in the current submission is thorough, optimizes there different catalysts (although with promoters which have been well described for the actual role they are meant for here) and the alcohol selectivities achieved are indeed notable, primarily due to the inhibition of C–O cleavage at the stage of hydroformylation and oxygenate hydroformylation thanks to splitting these conversion steps in another reactor operated at a milder temperature. Beyond the high selectivity, the claim of novelty appears somewhat overstated, especially with respect to the emphasis on C₄₊ products. This emphasis, though presented as a point of differentiation, seems more arbitrary than chemically justified given the nature of the ASF product distribution. Thus, although the manuscript is compelling, and it deserves publication (following further revision), I do not perceive that the conceptual threshold typically required for a journal such as Nature Communications is reached. The system described extends earlier work while increasing process complexity through the serial arrangement of two separate reactors. Referring to this as a “tandem” process is, in my view, inaccurate: chaining reactors without intermediate workup is standard industrial practice, particularly when distinct reaction conditions (e.g., temperature) are needed for sequential transformations.

Regarding the authors' responses to previous reviewer comments, all revision points have been addressed, but not all have been satisfactorily resolved in my opinion.

In response to requests for greater clarity in the DFT section, the authors have now provided full energetics along the calculated reaction pathways in tabulated form, which is appreciated. However, they have not included the coordinate files for the optimized models, as requested by the reviewers. This is important, particularly because concerns have been raised about the physical realism of the modeled surfaces. For instance, Cs on low work function metal surfaces like Co would be expected to adsorb as a neutral species (e.g., oxide or hydroxide) under FT conditions. The authors model Cs as a cation (Cs⁺), but no counterion is presented. I concur with other reviewers that this representation lacks chemical plausibility. Also, direct comparisons of energy profiles for fundamentally different reactions, such as the water–gas shift (WGS) and CO insertion into C_xH_y species, are overinterpretative without accompanying microkinetic modeling.

The reported methane selectivity (1.9–5.1 C%) appears unrealistically low to me, certainly falling below the ASF-predicted value. This is surprising, as the ASF value would be expected to be the theoretically lowest, even if hydrogenation activity, and hence overall conversion, has been heavily inhibited on a Co FTS catalyst. The authors should carefully re-examine their product quantification, include linearized ASF plots for all major product families, and assess the extent of any deviations from the expected statistical distribution. Selectivity trends across product classes are critically dependent on this analysis.

Authors should note that the comparison established in Figure 4b is very much dependent particularly on the CO/syngas conversion level at which selectivities are considered.

Other, minor, comments are:

The authors attribute paraffin formation to excessive hydrogenation of oxygenates. However, complete hydrogenation of oxygenates should yield alcohols, not paraffins. If paraffins are indeed forming, C–O bond cleavage must be occurring,

implying additional chemistries that have not been considered in the current discussion.

There is some inconsistency in the terminology used for catalyst preparation. The synthesis method is described at times as “incipient wetness impregnation” and at other times as “wetness impregnation.” These are distinct techniques, and the usage should be clarified to avoid confusion.

The mis correspondence between the composition and the notation for the catalysts has not been solved upon revision, which makes it very hard for a reader to actually understand what the composition of each material discussed in the text is.

Finally, the term “net CO₂ emissions” is typically associated with life-cycle analysis. Its use here to describe a scenario where minor CO₂ side-products are recycled upstream is, in my opinion, imprecise and should be revised.

Version 2:

Reviewer comments:

Reviewer #2

(Remarks to the Author)

As suggested, the authors have improved the DFT model of the catalyst.

Over the last decade, it has been established that alkylidyne species (CH₃CH₂C) are the key reaction intermediates that undergo hydrogenation, dehydrogenation or CO insertion. Direct olefin hydrogenation is not a relevant reaction in Fischer-Tropsch synthesis.

It has also been established that olefins are the primary products of cobalt Fischer-Tropsch synthesis, but thermodynamically, hydrogenation to paraffins is of course favorable and will dominate at higher conversions.

Reviewer #4

(Remarks to the Author)

The authors have further revised their manuscript. Important flaws in the earlier version have been addressed, for example the construction of more relevant DFT models and the inclusion of Cs as a promoter in a more sensible speciation therein. Other technical aspects have also been improved, making the manuscript technically more sound.

For publication in Nature Communications, I still consider that splitting a previously published process (by the same authors, ACS Catalysis 2021) into two reactors in series, to gain a more independent temperature control, does not constitute a sufficiently solid innovation. In addition, several of the reported performance indicators remain unconvincing. As an example, the authors claim to achieve a selectivity to C₄₊ alcohols above 80%, following an ASF distribution with a chain growth probability of approximately 0.67 (Figure S3). This is presented as a major achievement of this study. However, for said chain growth probability, the mathematically derived absolute maximum carbon selectivity to all products with four or more carbon atoms (C₄₊) is approximately 60%. How, then, can the authors report selectivities to C₄₊ alcohols as high as 80%? In my view, this strongly suggests issues in the product analysis and/or data treatment. It is similarly surprising that alcohols as short as C₁ and C₂ (methanol, ethanol) follow the same ASF distribution, despite the fact that neither of those could be produced by hydroformylation of olefins. All in all, I believe this study still requires substantial technical revision, including a critical assessment of product analysis and carbon balances, before it can be considered for publication.

Version 3:

Reviewer comments:

Reviewer #2

(Remarks to the Author)

The authors have replaced the original DFT simulations with more relevant simulations to support the experimental results. I have no additional comments.

Reviewer #4

(Remarks to the Author)

The current revised version of the manuscript can be recommended for publication.

Journal: Nature Communcations

Manuscript ID: NCOMMS-25-14503A

Title: Selective conversion of syngas to C₄₊ long-chain alcohols

Author(s): Yihui Li, Ziang Zhao, Miao Jiang, Guoqing Wang, Zheng Li, Wei Lu, Wenhao Cui, Rong Liu, Ronghe Lin, Yu Meng, Yuan Lyu, Li Yan, Hejun Zhu, Yunjie Ding

Response to Reviewers

Reviewer #1:

Comment:

This work reported the novel catalysis system for the conversion of syngas to C₄₊ long-chain alcohols. The system achieves a high selectivity of 80% C₄₊ long-chain alcohols with low CO₂ and CH₄ selectivities. Some interesting and important results were reported and discussed. However, the series connection of multiple catalysts causes an exceptionally complex reaction process, which may affect the industrial application prospects of the catalytic system. In addition, some conclusions have not received strong support and are not convincing. Therefore, there are some concerns that the authors should address well before further considering publication. The detailed comments are as follows:

Reply:

We warmly thanked the Reviewer #1 for recognizing the merits of this work and his/her careful assessment. We also sincerely thank you for providing the critical and inspiring comments, based on which we have now performed additional experiments and theoretical calculations to fully address your concerns. A detailed point-to-point has now been listed as follows:

- 1) The catalytic performance should be discussed with reaction conditions in the abstract and conclusion.

Reply:

Thank you for the nice suggestion. Following your advice, we have provided an expanded description of the catalytic performance by accommodating the relevant

reaction conditions. Still, we keep the Abstract untouched because of the word's limit of the journal.

“The system achieves, i) an unprecedented selectivity of 80% C₄₊ long-chain alcohols with a remarkably low CO₂ selectivity of 1% at 17% CO conversion (typical conditions of H₂/CO = 1.5, *F* = 130 ml min⁻¹, 210 °C (syngas-to-oxygenates/olefins) and 140 °C (hydroformylation and hydrogenation)), ii) ultra-high 97% carbon-efficiency, and iii) improved oxygen-efficiency, effectively addressing a key limitation of current syngas conversion technologies.”

2) For a series of modified Co/C catalysts with different metal promoters (Mn, Cr, Ni, Ca, Ba, and Cs), the effect of promoters on selectivities of CO₂ and CH₄ should be discussed.

Reply:

Thank you for the insightful instruction, following which we have provided additional a figure to show the impacts of different promoters on the CO₂ and CH₄ selectivity (**Fig. S2**). As one can see, Mn additives could significantly reduce the methane selectivity from 25.3% to 10.2%, and CO₂ selectivity is basically unchanged. After adding Cs additives, methane selectivity is significantly reduced from 25.3% to 8.4%, and CO₂ selectivity is increased from 0.9% to 2.1%. When Cs and Mn were added together, the methane selectivity decreased to 6.6%, but the CO₂ selectivity increased slightly to 2.7%. The addition of Ca and Ba additives significantly increased the methane selectivity. In addition, excessive Ni additives will increase the methane selectivity, but the addition of Ca, Ba and Ni additives has no obvious effect for CO₂ selectivity.

Fig. S2. CO₂ selectivity vs. CH₄ selectivity on carbon-supported Co catalysts with different promoters.

The corresponding discussion has been briefly supplemented in the revision:

“When 0.5 wt.% Mn was doped in Co/C, the selectivity of C₄₊ oxygenates/C₃₊ olefins was markedly increased from 37% to 50% without compromising CO conversion (Fig. 1b), which Mn was assumed to promote the C-C coupling and chain growth process, reducing CH₄ selectivity from 22.0% to 10.2% (Table S1).”

“While the additions of Cr or Ni improved CO conversion but inhibited the formations of oxygenates/olefins, Ca and Ba additives promoted CH₄ formation (Fig. S2), and great promotional effect was observed with the addition of Cs (Fig. 1a). The regulation of different metal additives scarcely impacted CO₂ selectivity (all less than 5%).”

- 3) The author suggested that interfacial sites of Co@Co₂C structure were crucial for the preferential formation of alcohols. How to confirm that the catalyst forms a Co@Co₂C core-shell structure during the reaction process?

Reply:

We thank the Reviewer for the insightful instruction. We sincerely apologize for any confusion caused by our previous message. We should stress that here Co@Co₂C was used to indicate the formation of interphases between Co and Co₂C entities, but not core-shell structures.

In order to clarify the catalyst structure, we have supplemented additional experiments (in situ PXRD, STEM and HRTEM) for a more detailed description (Fig. S5 and Fig.1).

The in situ PXRD tests of CoMnCs/C under different atmospheres revealed marked differences on the catalyst compositions. Specifically, only fcc-Co was formed subject to H₂ reduction even up to 430 °C, but both fcc-Co and Co₂C were generated after syngas treatment at 210 °C at prolonged time. The latter phenomenon evidenced the coexistence of both Co species, and indicated that Co₂C was likely in situ formed only under the syngas conversion conditions.

Fig. S5. In situ PXRD patterns of the CoMnCs/C catalysts under different conditions: (a) in H₂ atmosphere (430 °C, 0.1 MPa); (b) syngas ambience (210 °C, 0.1 MPa).

To further visualize the more precise catalyst structures, we performed additional electron microscopic analyses of the spent CoMnCs/C catalysts. First, we acquired STEM with elemental color mapping image of the spent CoMnCs/C, showing the apparent enrichment of carbon species around the Co nanoparticle (Fig. 1c). Then we also performed a line scanning of an individual particle as shown in Fig. 1d. One can see the relatively even distributions of Cs and Mn along the Co nanoparticle, possibly because of their lower contents. In contrast, the C signals were still quite strong beyond the Co nanoparticles, further substantiating the color-mapping result. As carburization gradually accumulates, Co particles and the carbon layer undergo further reaction under reaction conditions and are gradually and partially converted into Co₂C. To validate this hypothesis, we performed HRTEM test and analyzed the lattice fringes of an individual Co particle and the surroundings (Fig. 1e). We found that the Co(111) crystals and the periphery was decorated by smaller Co₂C ensembles. Notably, the Co particle was only partially decorated by several Co₂C ensembles, instead of fully covered by the latter. Therefore, we trust it is more appropriate to use interphases rather than core-shell

structure to depict the key structure of the CoMnCs/C catalyst.

Fig. 1. Design of efficient syngas-to-oxygenates/olefins catalysts. (a) CO conversion vs. total selectivity of alcohols and olefins on carbon-supported Co catalysts with different promoters. (b) Comparison on the syngas-to-oxygenates/olefins performance on representative Co-based catalysts. Reaction conditions: 210 °C, 3.0 MPa, H₂/CO = 2.0, $F = 65 \text{ ml min}^{-1}$ (See Table S1 for details). (c-e) STEM and HRTEM images of Co_{0.5}Mn_{0.1}Cs/C catalyst. (c) EDX mapping images. (d) Line scanning profiles (inset). (f) The reaction coordinate of C₃H₁₀* desorption, hydrogenation, and CO insertion pathways on Co(111), Co@Co₂C, and CoCs@Co₂C models. Bader charge analysis were shown in the inset.

We have now accommodated these new results and discussions into the revised manuscript.

“These observations strongly suggested the co-existence of both Co⁰ and *in situ* formed Co₂C on the catalyst surface, which was further confirmed by detailed high-resolution transmission in situ PXRD and electron microscopy observations. The in situ PXRD experiments verified the formation of both fcc-Co and Co₂C on CoMnCs/C under syngas treatment conditions but only fcc-Co after H₂ reduction (Fig. S5), hinting the

critical role of reaction atmospheres on the catalyst structure. Scanning transmission electron microscopy (STEM) coupled with elemental color mapping image of the spent catalyst unambiguously pointed to the surface enriching of carbon species around Co nanoparticle of CoMnCs/C catalyst (Fig. 1c). This was further supported by line-scanning of an individual nanoparticle, showing relatively even distributions of Cs and Mn across the nanoparticles but much higher C signal than that of Co at the periphery (Fig. 1d). As carburization gradually accumulated, Co particles and the carbon layer underwent further reaction under reaction conditions and were gradually and partially converted into Co₂C. To validate this hypothesis, HRTEM image of the spent catalyst was acquired (Fig. 1e). Lattice fringe analysis revealed the structure of fcc-Co(111) in an individual particle decorated by small Co₂C ensembles (Fig. 1e). Noted that Co₂C was present as small patches on the Co surface, forming Co@Co₂C interphases, but did not fully cover the latter.”

4) For DFT calculations, the detailed E, H, S, ZPE, G values for each IS, TS, and FS states should be listed in table. Also the coordinate files for optimized structures should be given in the supporting information.

Reply:

We thank the reviewers for their insightful and valuable suggestions. The detailed E, H, S, ZPE, G values for each IS, TS, and FS states in different calculation models have now been listed in Table S3, and the coordinate files for all optimized structures were given in the Table S9-10.

Table S3 The detailed E, H, S, ZPE, G values for each IS, TS, and FS states of the reaction pathway studied in different calculation models

Model	Adsorbate	E	ZPE	H	S	G
Co	*C ₅ H ₁₀ +*H	-476.401	3.79037	4.29604	1.01725	3.27879
	TS	-474.539	3.7998	4.32034	1.089303	3.23104
	C ₅ H ₁₁ +	-478.182	3.90762	4.423559	1.096589	3.32697
	*C ₅ H ₁₁ +*H	-481.493	4.09588	4.657957	1.214	3.443957
	TS	-481.256	4.17643	4.68669	1.107665	3.57902
	C ₅ H ₁₂ +	-482.153	4.243895	4.72761	1.05287	3.67474
	*C ₅ H ₁₀ +*CO	-491.074	3.814761	4.439282	1.310257	3.129025
	TS	-489.132	3.78738	4.3921	1.29069	3.1015
	*C ₅ H ₁₀ CO	-489.794	3.85222	4.44694	1.19525	3.25168

Co@Co ₂ C	*C ₅ H ₁₀ +*H	-1172.37	3.239188	3.414707	0.258252	3.156455
	TS	-1171.65	3.754593	4.239493	1.00823	3.231262
	C ₅ H ₁₁ +	-1171.74	3.935586	4.452394	1.114208	3.338186
	*C ₅ H ₁₁ +*H	-1175.41	4.07887	4.593925	1.050358	3.543567
	TS	-1175.06	4.10611	4.64439	1.158693	3.485698
	C ₅ H ₁₂ +	-1175.75	4.244185	4.765384	1.139558	3.625826
	*C ₅ H ₁₀ +*CO	-1191.02	3.836483	4.4599	1.276726	3.183174
	TS	-1190.02	3.81345	4.411023	1.252404	3.158618
CoC ₅ @Co ₂ C	*C ₅ H ₁₀ +*H	-1166.95	3.807738	4.322573	1.043739	3.278834
	TS	-1164.89	3.676979	4.168255	1.00371	3.164542
	C ₅ H ₁₁ +	-1166.61	3.93566	4.440723	1.045434	3.395289
	*C ₅ H ₁₁ +*H	-1170.35	4.110334	4.643229	1.085104	3.558125
	TS	-1169.4	4.077055	4.614339	1.120726	3.493614
	C ₅ H ₁₂ +	-1170.1	4.249638	4.76587	1.118895	3.646974
	*C ₅ H ₁₀ +*CO	-1179.76	3.769668	4.365136	1.203293	3.161843
	TS	-1178.7	3.791014	4.356055	1.105943	3.250112
Cu	*C ₅ H ₁₀ CO	-1190.54	3.894902	4.475153	1.174496	3.300656
	*C ₅ H ₁₀ +*H	-1166.95	3.807738	4.322573	1.043739	3.278834
	TS	-1164.89	3.676979	4.168255	1.00371	3.164542
	C ₅ H ₁₁ +	-1166.61	3.93566	4.440723	1.045434	3.395289
	*C ₅ H ₁₁ +*H	-1170.35	4.110334	4.643229	1.085104	3.558125
	TS	-1169.4	4.077055	4.614339	1.120726	3.493614
	C ₅ H ₁₂ +	-1170.1	4.249638	4.76587	1.118895	3.646974
	*C ₅ H ₁₀ +*CO	-1179.76	3.769668	4.365136	1.203293	3.161843
	TS	-1178.7	3.791014	4.356055	1.105943	3.250112
	*C ₅ H ₁₀ CO	-1178.79	3.830185	4.42587	1.216015	3.209856
	*H ₂ O	-434.871	0.622854	0.744382	0.269046	0.475336
	TS	-433.461	0.41278	0.506652	0.19411	0.312538
	*OH+*H	-434.788	0.521215	0.624136	0.172818	0.451318
	*CO+*OH+*H	-450.507	0.698897	0.922948	0.420045	0.502903
	TS	-449.488	0.711526	0.900765	0.396387	0.504378
	*COOH+*H	-450.148	0.772586	0.968776	0.421367	0.547409
TS	-449.075	0.587511	0.734066	0.294878	0.439187	
*CO ₂ +*H ₂	-449.383	0.591737	0.761651	0.35426	0.407389	
*H ₂	-427.223	0.294778	0.410038	0.280709	0.129328	
TS	-426.593	0.369225	0.438501	0.126612	0.311889	
*2H	-427.769	0.350252	0.37815	0.036417	0.341733	
*C ₅ H ₁₁ CHO+*H	-529.36	4.661801	5.155494	1.065767	4.089727	
TS	-527.71	4.618257	5.08528	0.955278	4.130002	
C ₅ H ₁₁ CH ₂ O+	-528.93	4.764797	5.170733	0.800561	4.370172	
*C ₅ H ₁₁ CH ₂ O+*H	-532.276	4.9056	5.351431	0.91401	4.43742	
TS	-532.106	4.87405	5.357289	0.99015	4.367139	
C ₅ H ₁₁ CH ₂ OH+	-533.395	5.117547	5.586337	0.96158	4.624758	
Cu@ZrO ₂	*H ₂ O	-1744.2	0.634097	0.773856	0.318171	0.455685
	TS	-1743.44	0.419979	0.506379	0.162394	0.343985
	*OH+*H	-1744.29	0.479353	0.58429	0.2266	0.35769
	*CO+*OH+*H	-1757.15	0.53155	0.71298	0.357041	0.35594
	TS	-1756.09	0.568742	0.728394	0.31864	0.409753
	*COOH+*H	-1759.48	0.718562	0.860856	0.292235	0.568621
	TS	-1758.98	0.627312	0.768796	0.289707	0.479089
	*CO ₂ +*H ₂	-1759.39	0.659887	0.813417	0.284827	0.52859
	*H ₂	-1736.3	0.305213	0.373511	0.129036	0.244476

TS	-1735.97	0.321603	0.392989	0.13511	0.25788
*2H	-1737.26	0.337568	0.379741	0.064128	0.315613
*C ₅ H ₁₁ CHO+*H	-1839.46	4.650485	5.047186	0.777926	4.26926
TS	-1838.82	4.618898	5.04363	0.872195	4.171435
C ₅ H ₁₁ CH ₂ O+	-1839.38	4.786809	5.199093	0.85693	4.342163
*C ₅ H ₁₁ CH ₂ O+*H	-1842.54	4.970997	5.416286	0.913088	4.503198
TS	-1842.08	4.931647	5.352619	0.848944	4.503675
C ₅ H ₁₁ CH ₂ OH+	-1843.15	5.147832	5.560021	0.828798	4.731223

	*C ₅ H ₁₀ + *CO	TS	*C ₅ H ₁₀ CO
Co(111)			Co@Co ₂ C			CoCs@Co ₂ C			
Fig. S9. Illustrations of different configurations of *CO insertion into *C₅H₁₀ species on the Co(111), Co@Co₂C and CoCs@Co₂C.

	*C ₅ H ₁₀ + *H	TS	*C ₅ H ₁₁	*C ₅ H ₁₁ + *H	TS	*C ₅ H ₁₂
Co(111)						Co@Co ₂ C						CoCs@Co ₂ C						
Fig. S10. Illustrations of different configurations of *C₅H₁₀ hydrogenation on the Co(111), Co@Co₂C and CoCs@Co₂C.

5) Does the lifespan of three catalysts, which are coupled in this work, differ? Will the mismatched lifespan of different catalysts limit the further industrial application of this catalytic system?

Reply:

We thank the reviewers for the insightful and valuable suggestions. Stability test of 130 h for CoMnCs/C|3v-POPs-PPh₃|CuZrO₂/SiO₂ tandem catalytic system was shown in

Fig. 3b in the manuscript. To address the Reviewer's concern, we have supplemented additional 300 h stability experiments for both syngas-to-oxygenates/olefins and hydrogenation processes, showing relatively stable performances (Fig. R1-2). Based on our previous works (Chin. J. Catal., 2025, 73, 16-38; *Reac. Kinet. Mech. Cat.*, 2015, 116, 223-234; *J. Mol. Catal.*, 2015, 404, 211-217.)¹⁻³, the hydroformylation catalyst could run stably for 1000 and 500 h for hydroformylation of ethylene and propylene, respectively. These preliminary results demonstrated the great potential of the proposed tandem processes for C₄₊ alcohol synthesis from syngas, although we acknowledge that longer durability tests in scale-up utilities are indispensable for realizing industrial application in the future.

Fig. 3b. Stability test of CoMnCs/C[3v-POPs-PPh₃]CuZrO₂/SiO₂. Reaction conditions: 210 °C (syngas-to-oxygenates/olefins) and 140 °C (hydroformylation/hydrogenation), 3.0 MPa, H₂/CO = 1.5, *F* = 130 ml min⁻¹.

Fig. R1. Stability test of the Co_{0.5}Mn_{0.1}Cs/C catalysts Fischer-Tropsch synthetic (FTS). Reaction conditions: *P* = 3.0 MPa, *T* = 210 °C, GHSV = 2000 h⁻¹, H₂/CO = 2.

Fig. R2. Stability test of the $\text{CuZrO}_2/\text{SiO}_2$ catalyst for hydrogenation of Fischer-Tropsch synthetic products. Reaction conditions: $P = 3.0$ MPa, $T = 140$ °C, $\text{H}_2/\text{CO} = 2$, $F (\text{H}_2/\text{CO} = 2) = 30$ ml min^{-1} , F (FTS products) = 0.02 ml min^{-1} .

6) I suggest the author also compare the STY of C_{4+} alcohols with other catalytic systems reported in the literature.

Reply:

Thank you for your valuable comments. We have now accommodated the STY of C_{4+} alcohols in Table S5.

Table S5. Catalytic performances of various catalysts for syngas conversion to oxygenates in the literatures.

Catalysts	T / °C	P / MPa	H_2/CO / mol mol ⁻¹	GHSV/ ml g ⁻¹ h ⁻¹	$\text{STY}_{\text{C}_4+\text{OH}}$ / g kg _{cat} ⁻¹ h ⁻¹	refs.
CoMnCs/C Rh/3v-POPs- PPh ₃ CuZrO ₂ /SiO ₂	210/140 ^a	3.0	1.5	1600 ^b	70.8	this work
	210/140 ^a	8.0	1.5	1600 ^b	89.7	this work
	230/140 ^a	3.0	1.5	1600 ^b	118.7	this work
CoMnCs/MC Rh/POPs- BP&PPh ₃ CuZrO ₂ /SiO ₂	210/140	3.0	1.5	1600 ^b	111.9	this work
Cu ₄ Fe ₁ Mg ₄ -MMO	260	3.0	2.0	2400 ^b	101	4
15Co5Fe/AC	220	3.0	2.0	2800	-	5
ER-MoS ₂ -K	240	5.0	2.0	3000 ^b	-	6
50Mo/50Co	250	2.0	2.0	15 ^c	-	7
Cs ₂ O-Cu/ZnO/Al ₂ O ₃	280	5.4	3.0	3750	-	8
Co/CuZnO	250	0.2	2.0	45 ^c	-	9
15Co/SiO ₂	220	3.0	2.0	5600	-	10
15Co/AC	220	3.0	2.0	2800	28	5
15Co6.3SiO ₂ /AC	220	3.0	2.0	500	-	10
15Co1.9Al ₂ O ₃ /AC	220	3.0	2.0	500	-	10
15Co/AC-H ₂ -HNO ₃	210	3.0	2.0	2000	-	11

15Co0.5Mn/AC	220	3.0	2.0	2000	50	12
15Co0.5Mn1La/AC	220	3.0	2.0	2000	25	13
15Co2Cr/AC	220	3.0	2.0	2000	-	14
15Co0.1Ca/AC	220	3.0	2.0	900	-	15
Co/MnO _x @quasi-MOF-74	230	3.0	2.0	4500 ^b	18	16
Co₁Cu₁Mn₁	240	6.0	2.0	2000	-	17
Co ₂ Cu ₁	240	4.0	1.5	2000	-	18
Co ₁ Cu ₂ Nb _{0.2}	200	6.0	1.5	40 ^c	-	19
Co ₄ Mn ₁ K _{0.1}	220	4.0	1.5	40 ^c	44	20
Co ₁ Cu ₁ Mn ₁	240	6.0	2.0	2000	78	17
2.1Ru-CoMn	220	6.0	1.0	2000 ^b	26	21
1.1Rh-CoMn	220	6.0	1.0	2000 ^b	40	21
CuCoAl ZnO/ZrO ₂	250	5.0	2.0	4000 ^b	11	22
CuZnAl-CoMn	220	6.0	2.0	2000 ^b	-	23
CoMn CuZnAlZr	220	6.0	2.0	2000 ^b	23	24
NaPr-CoRu/AomM Co ₂ CO ₈ +PCy ₃	200	1.2	2.0	-	-	25
ZnCrAlO _x KNiMoS-MMO-5	250	5.0	1.0	3000 ^b	-	26
CuCoAl t-ZrO ₂	-	-	-	-	-	27

a 210 °C (syngas-to-oxygenates/olefins), 140 °C (hydroformylation/hydrogenation).

b h⁻¹.

c scem total flow.

d estimated value

e Alcohol and aldehyde.

Reviewer #2:**Comment:**

The authors report a nice study where they combine an optimized promoted Co-FT catalyst with a typical heterogeneous hydroformylation catalyst and a selective hydrogenation catalyst to convert syngas to C₄+ alcohols with a remarkable selectivity at low CO conversions (below 15%) in a two-step process without intermediate separation. The low product yield is likely one of the barriers to implement this two-step process. Separation of the alcohols from water is costly separation of water after the FT step is likely more efficient and indeed done industrially.

This is a nice experimental study and worth publishing. The experiments are supported by standard characterizations (XRD for some of the spent Co catalysts, XPS for some of the spent Co catalysts, TEM for the selected catalyst). BET, H₂ chemisorption and TPR could be included for the optimized Co catalysts.

Reply:

We sincerely thanked the Reviewer # 2 for praising our work and reaccommodating the publication of this piece. We also thank you for your insightful comments and suggestions, which are valuable for improving the quality of this work.

- 1) Some DFT calculations are included for selected reactions on model surfaces (Co(111), Co island on Co₂C, and a Cu₁₃ cluster on ZrO₂). The DFT calculations add little scientific value to the paper since they do not consider the relevant active sites, surface coverages or the relevant reaction steps. Also, the introduction of a Cs atom to describe the effect of the Cs promotor is incorrect. Cs is likely present as Cs₂O or Cs₂CO₃, and not all reactions will occur next to the promotor as proposed by the calculations. The quality of the paper would improve if the DFT calculations are not included.

Reply:

We agree with the Reviewer that the selection of models should be conducted with great care. We also trust the rational combinations of experiments with computational studies valuable for providing insights into the reaction mechanism. To strengthen the basis for

the DFT calculations, the following actions have been taken in the revision:

i) We have conducted detailed structure analysis of CoMnCs/C catalyst by HRTEM and STEM, based on which the rationale of Co@Co₂C was established. The Reviewer is kindly asked to refer the detailed reply of Question 5.

ii) Secondly, for the key intermediates such as C₅H₁₀ in the studied reaction, we systematically calculated multiple possible active sites, and then selected the most stable adsorption sites to conduct the relevant key elementary reaction calculations. Because the calculation of coverage is complex and involves a large amount of work, we did not consider the influence of coverage.

iii) We have simulated Cs^{δ+} as the corresponding site for the model construction. For the additive Cs, although we considered using Cs instead of Cs₂O, as many studies have shown that it is mainly used as an electronic additive, we adopted the substitution of Cs⁺ cations for Cs₂O in order to reduce the excessive complexity of the model here. Table S1 shows the Bader charge results of Cs in CoCs@Co₂C, which indicates that Cs is in the +1-valence state. In addition, we supplemented a Cs₂O model (Fig. R5) and calculated the adsorption of the key intermediate C₅H₁₀. We found that the results of this model for the most stable adsorption of C₅H₁₀ were relatively close to those of the Cs cation model shown as in Table S2. Therefore, we believe that the results of the Cs cation model adopted in this work are also convincing.

iv) Additionally, the detailed E, H, S, ZPE, G values for each IS, TS, and FS states in different calculation models have now been listed in Table S3, and the coordinate files for all optimized structures were given in the Supporting Information.

Table S2. Bader charge of Cs in CoCs@Co₂C model

Model	The number of electrons in Cs	The number of electrons gained or lost by Cs
CoCs@Co ₂ C	8.228	-0.772

Fig. R5. Illustrations of different configurations of $*C_5H_{10}$ adsorption on the Co-Cs₂O@Co₂C.

Table R1 adsorption energy of $*C_5H_{10}$ at different sites in three Cs doping model

Models	Adsorption energy (eV)
CoCs@Co ₂ C	-0.64
	-0.91
Co-Cs ₂ O@Co ₂ C	-0.52
	-0.95

Table S3 The detailed E, H, S, ZPE, G values for each IS, TS, and FS states of the reaction pathway studied in different calculation models

Model	Adsorbate	E	ZPE	H	S	G	
Co	$*C_5H_{10}+*H$	-476.401	3.79037	4.29604	1.01725	3.27879	
	TS	-474.539	3.7998	4.32034	1.089303	3.23104	
	$*C_5H_{11}+*$	-478.182	3.90762	4.423559	1.096589	3.32697	
	$*C_5H_{11}+*H$	-481.493	4.09588	4.657957	1.214	3.443957	
	TS	-481.256	4.17643	4.68669	1.107665	3.57902	
	$*C_5H_{12}+*$	-482.153	4.243895	4.72761	1.05287	3.67474	
	$*C_5H_{10}+*CO$	-491.074	3.814761	4.439282	1.310257	3.129025	
	TS	-489.132	3.78738	4.3921	1.29069	3.1015	
	$*C_5H_{10}CO$	-489.794	3.85222	4.44694	1.19525	3.25168	
	Co@Co ₂ C	$*C_5H_{10}+*H$	-1172.37	3.239188	3.414707	0.258252	3.156455
		TS	-1171.65	3.754593	4.239493	1.00823	3.231262
		$*C_5H_{11}+*$	-1171.74	3.935586	4.452394	1.114208	3.338186
		$*C_5H_{11}+*H$	-1175.41	4.07887	4.593925	1.050358	3.543567
		TS	-1175.06	4.10611	4.64439	1.158693	3.485698
$*C_5H_{12}+*$		-1175.75	4.244185	4.765384	1.139558	3.625826	
$*C_5H_{10}+*CO$		-1191.02	3.836483	4.4599	1.276726	3.183174	
TS		-1190.02	3.81345	4.411023	1.252404	3.158618	

	*C ₅ H ₁₀ CO	-1190.54	3.894902	4.475153	1.174496	3.300656
	*C ₅ H ₁₀ +*H	-1166.95	3.807738	4.322573	1.043739	3.278834
	TS	-1164.89	3.676979	4.168255	1.00371	3.164542
	C ₅ H ₁₁ +	-1166.61	3.93566	4.440723	1.045434	3.395289
	*C ₅ H ₁₁ +*H	-1170.35	4.110334	4.643229	1.085104	3.558125
CoCs@Co ₂ C	TS	-1169.4	4.077055	4.614339	1.120726	3.493614
	C ₅ H ₁₂ +	-1170.1	4.249638	4.76587	1.118895	3.646974
	*C ₅ H ₁₀ +*CO	-1179.76	3.769668	4.365136	1.203293	3.161843
	TS	-1178.7	3.791014	4.356055	1.105943	3.250112
	*C ₅ H ₁₀ CO	-1178.79	3.830185	4.42587	1.216015	3.209856
	*H ₂ O	-434.871	0.622854	0.744382	0.269046	0.475336
	TS	-433.461	0.41278	0.506652	0.19411	0.312538
	*OH+*H	-434.788	0.521215	0.624136	0.172818	0.451318
	*CO+*OH+*H	-450.507	0.698897	0.922948	0.420045	0.502903
	TS	-449.488	0.711526	0.900765	0.396387	0.504378
	*COOH+*H	-450.148	0.772586	0.968776	0.421367	0.547409
	TS	-449.075	0.587511	0.734066	0.294878	0.439187
	*CO ₂ +*H ₂	-449.383	0.591737	0.761651	0.35426	0.407389
Cu	*H ₂	-427.223	0.294778	0.410038	0.280709	0.129328
	TS	-426.593	0.369225	0.438501	0.126612	0.311889
	*2H	-427.769	0.350252	0.37815	0.036417	0.341733
	*C ₅ H ₁₁ CHO+*H	-529.36	4.661801	5.155494	1.065767	4.089727
	TS	-527.71	4.618257	5.08528	0.955278	4.130002
	C ₅ H ₁₁ CH ₂ O+	-528.93	4.764797	5.170733	0.800561	4.370172
	*C ₅ H ₁₁ CH ₂ O+*H	-532.276	4.9056	5.351431	0.91401	4.43742
	TS	-532.106	4.87405	5.357289	0.99015	4.367139
	C ₅ H ₁₁ CH ₂ OH+	-533.395	5.117547	5.586337	0.96158	4.624758
	*H ₂ O	-1744.2	0.634097	0.773856	0.318171	0.455685
	TS	-1743.44	0.419979	0.506379	0.162394	0.343985
	*OH+*H	-1744.29	0.479353	0.58429	0.2266	0.35769
	*CO+*OH+*H	-1757.15	0.53155	0.71298	0.357041	0.35594
	TS	-1756.09	0.568742	0.728394	0.31864	0.409753
	*COOH+*H	-1759.48	0.718562	0.860856	0.292235	0.568621
	TS	-1758.98	0.627312	0.768796	0.289707	0.479089
	*CO ₂ +*H ₂	-1759.39	0.659887	0.813417	0.284827	0.52859
Cu@ZrO ₂	*H ₂	-1736.3	0.305213	0.373511	0.129036	0.244476
	TS	-1735.97	0.321603	0.392989	0.13511	0.25788
	*2H	-1737.26	0.337568	0.379741	0.064128	0.315613
	*C ₅ H ₁₁ CHO+*H	-1839.46	4.650485	5.047186	0.777926	4.26926
	TS	-1838.82	4.618898	5.04363	0.872195	4.171435
	C ₅ H ₁₁ CH ₂ O+	-1839.38	4.786809	5.199093	0.85693	4.342163
	*C ₅ H ₁₁ CH ₂ O+*H	-1842.54	4.970997	5.416286	0.913088	4.503198
	TS	-1842.08	4.931647	5.352619	0.848944	4.503675
	C ₅ H ₁₁ CH ₂ OH+	-1843.15	5.147832	5.560021	0.828798	4.731223

- 2) On line 38, the authors mention iso-butanol as a valuable product, but the proposed process does not seem to produce significant amounts of isobutanol since the *n/i* selectivity in the hydroformylation is not controlled.

Reply:

We agree with the Reviewer that the formation of *iso*-butanol is not high compared with that of *n*-butanol. Since this article is mainly aimed at C₄₊ alcohols, and the wide use of C₄, C₅₋₉ and C₁₀₋₂₀ alcohols is generally outlined in the Introduction. As the Reviewer may have also noticed, the *n/i* ratios of butanol result from the hydroformylation of propylene. In addition, our previous works indicated that the *n/i* ratios can be finely tuned by engineering the phosphine ligands of the polymer support. The Reviewer is kindly asked to refer to our previous works (J. Catal.,2017, 353, 123-132, Nat. Commun., 2024, 15, 6967)^{28 29}.

We have made revisions in the manuscript: “For instance, butanol, a key precursor in plastics (e.g., butyl acrylate and butyl acetate) and gasoline additive, The global markets for n-butanol are projected to grow at a compound annual growth rate (CAGR) of 6.3% during 2021-2032 reflecting their economic significance.”

- 3) It would be interesting to include a Table with the state-of-the-art in C₄₊ alcohol synthesis in the SI to support the statement of ca. 20% on line 73.

Reply:

We thank the Reviewer for the valuable comments. The method of synthesizing C₄₊ alcohols has been specifically described in Fig. S1, and the selectivity of C₂₊, C₄₊ and C₆₊ alcohols produced from syngas. Rh, Co, Cu, was summarized in Table S5. The total C₄₊ alcohol selectivity of Fe- and Mo-based catalysts is about 20%. We found that the C₄₊ alcohol selectivity of some catalysts is greater than 20%, but the products are mainly the mixture of alcohols and aldehydes.

These results were already mentioned in the previous manuscript:

“Despite advances in bi- or multi-component state-of-the-art catalysts (MoS₂, Cu-Co, Cu-Fe, Co-Fe, Co-Mn, Cs-Cu-Zn), selectivity for C₄₊ alcohols remains low (ca. 20%, Fig S1 and Table S5).”

Fig. S1. Different synthetic routes for the production of C_{4+} long-chain alcohols. Commercialized and envisaged processes were highlighted by the red and blue boxes, respectively. Selectivity to C_{4+} alcohols via different routes was shown in the green box.

Table S5. Catalytic performances of various catalysts for syngas conversion to oxygenates in the literatures.

Catalysts	T / °C	P / MPa	H_2/CO / mol mol ⁻¹	GHSV/ ml g ⁻¹ h ⁻¹	S_{C_2+OH} / %	S_{C_4+OH} / %	S_{C_6+OH} / %	refs.
CoMnCs/C Rh/3v-POPs- PPh ₃ CuZrO ₂ /SiO ₂	210/140 ^a	3.0	1.5	1600 ^b	80.0	73.0	35.7	this work
	210/140 ^a	8.0	1.5	1600 ^b	86.3	79.9	45.5	this work
	230/140 ^a	3.0	1.5	1600 ^b	50.3	43.4	26.6	this work
CoMnCs/MC Rh/POPs- BP&PPh ₃ CuZrO ₂ /SiO ₂	210/140	3.0	1.5	1600 ^b	72.7	68.5	50.1	this work
Cu ₄ Fe ₁ Mg ₄ -MMO	260	3.0	2.0	2400 ^b	18.6 ^c	~13 ^c	-	4
15Co5Fe/AC	220	3.0	2.0	2800	15.9 ^c	█	5.6 ^c	5
ER-MoS ₂ -K	240	5.0	2.0	3000 ^b	30.0 ^c	1.7 ^c	0.0	6
50Mo/50Co	250	2.0	2.0	15 ^c	18.0 ^c	█	-	7
Cs ₂ O-Cu/ZnO/Al ₂ O ₃	280	5.4	3.0	3750	15.0 ^{dc}	~5.0 ^c	0.0	8
Co/CuZnO	250	0.2	2.0	45 ^c	42.1 ^c	█	0.0	9
15Co/SiO ₂	220	3.0	2.0	5600	-	█	-	10
15Co/AC	220	3.0	2.0	2800	16.5 ^c	█	9.4 ^c	5
15Co6.3SiO ₂ /AC	220	3.0	2.0	500	18.6 ^c	█	10.9 ^c	10
15Co1.9Al ₂ O ₃ /AC	220	3.0	2.0	500	17.9 ^c	█	10.0 ^c	10
15Co/AC-H ₂ -HNO ₃	210	3.0	2.0	2000	9.2 ^c	█	3.8 ^c	11
15Co0.5Mn/AC	220	3.0	2.0	2000	17.4 ^c	█	6.1 ^c	12
15Co0.5Mn1La/AC	220	3.0	2.0	2000	22.9 ^c	█	5.5 ^c	13
15Co2Cr/AC	220	3.0	2.0	2000	0.0	█	0.0	14
15Co0.1Ca/AC	220	3.0	2.0	900	29.4 ^c	█	16.1 ^c	15
Co/MnO _x @quasi-MOF- 74	230	3.0	2.0	4500 ^b	36.0 ^c	~20 ^c	~14 ^c	16
Co ₁ Cu ₁ Mn ₁	240	6.0	2.0	2000	-	█	-	17
Co ₂ Cu ₁	240	4.0	1.5	2000	27.4 ^c	█	-	18
Co ₁ Cu ₂ Nb _{0.2}	200	6.0	1.5	40 ^c	-	█	-	19
Co ₄ Mn ₁ K _{0.1}	220	4.0	1.5	40 ^c	~52 ^c	~35 ^c	~22 ^c	20
Co ₁ Cu ₁ Mn ₁	240	6.0	2.0	2000	-	█	-	17
2.1Ru-CoMn	220	6.0	1.0	2000 ^b	26.7 ^c	█	-	21

1.1Rh-CoMn	220	6.0	1.0	2000 ^b	31.7 ^c	█	-	21
CuCoAl ZnO/ZrO ₂	250	5.0	2.0	4000 ^b	20.8 ^c	~8 ^c	-	22
CuZnAl-CoMn	220	6.0	2.0	2000 ^b	36.2 ^c	█	~3 ^c	23
CoMn CuZnAlZr	220	6.0	2.0	2000 ^b	47.5 ^c	~36 ^c	~33 ^{d,c}	24
NaPr-CoRu/AomM Co ₂ CO ₈ +PCy ₃	200	1.2	2.0	-	-	█	1.2 ^c	25
ZnCrAlO _x KNiMoS- MMO-5	250	5.0	1.0	3000 ^b	44.0 ^c	6.6 ^c	0.0	26
CuCoAl t-ZrO ₂	-	-	-	-	51.6 ^c	█	0.0	27

^a 210 °C (syngas-to-oxygenates/olefins), 140 °C (hydroformylation/hydrogenation).

^b h⁻¹.

^c sccm total flow.

^d estimated value

^e Alcohol and aldehyde.

4) The authors use subscripts to describe the catalysts (Co_{0.5}Mn_{0.1}Cs), but do not explain the subscripts. In the synthesis section, it is stated that all catalysts have 15 wt% Co loading. More details about the catalyst synthesis and composition (ICP?) could be included.

Thank you for pointing out the relevant comments. We have now measured the metal contents of the key catalysts by ICP and provided the results in Table S12. Relevant explanations are provided in the manuscript:

“The modified Co/C catalysts were prepared by incipient wetness impregnation method. Loadings of Co and the other metal additives was set as 15 wt.% and 0.03~3 wt.% (indicated by subscripts after the element), respectively.”

Table S12. Synthetic approach and compositional for catalysts.

Catalyst	Synthesis method	Elements of composition	Nominal Content / wt. %	Measured content / wt. %
Co/C	wetness impregnation	Co	15	15.2 ^a
Co _{0.5} Mn/C	wetness impregnation	Co	15	14.9 ^a
		Mn	0.5	0.5 ^a
Co _{0.1} Cs/C	wetness impregnation	Co	15	15.1 ^a
		Cs	0.1	0.1 ^a

Co _{0.5} Mn _{0.1} Cs/C	wetness	Co	15	14.9 ^a
	impregnation	Mn	0.5	0.5 ^a
		Cs	0.1	0.1 ^a
Rh/3v-POPs-PPh ₃	post-loading	Rh	0.125	0.1 ^a
Rh/POPs-BP&PPh ₃	post-loading	Rh	0.125	0.1 ^a
CuZrO ₂ /SiO ₂	ammonia	Cu	20	19.0 ^b
	evaporation	Zr	-	28.0 ^b
Cu/SiO ₂	ammonia	Cu	20	22.0 ^b
	evaporation			

^a Determined from ICP-OES

^b Determined from XRF

5) Evidence for interfacial sites on Co@Co₂C is not provided (line 147). Very little is known about the structure of the CoMnCs/C catalyst.

Reply:

We thank the Reviewer for the insightful instruction. We sincerely apologize for any confusion caused by our previous message. We should stress that here Co@Co₂C was used to indicate the formation of interphases between Co and Co₂C entities, but not core-shell structures.

In order to clarify the catalyst structure, we have supplemented additional experiments (in situ PXRD, STEM and HRTEM) for a more detailed description (Fig. S5 and Fig.1). The in situ PXRD tests of CoMnCs/C under different atmospheres revealed marked differences on the catalyst compositions. Specifically, only fcc-Co was formed subject to H₂ reduction even up to 430 °C, but both fcc-Co and Co₂C were generated after syngas treatment at 210 °C at prolonged time. The latter phenomenon evidenced the coexistence of both Co species, and indicated that Co₂C was likely in situ formed only under the syngas conversion conditions.

Fig. S5. In situ PXRD patterns of the CoMnCs/C catalysts under different conditions: (a) in H₂ atmosphere (430 °C, 0.1 MPa); (b) syngas ambience (210 °C, 0.1 MPa).

To further visualize the more precise catalyst structures, we performed additional electron microscopic analyses of the spent CoMnCs/C catalysts. First, we acquired STEM with elemental color mapping image of the spent CoMnCs/C, showing the apparent enrichment of carbon species around the Co nanoparticle (Fig. 1c). Then we also performed a line scanning of an individual particle as shown in Fig. 1d. One can see the relatively even distributions of Cs and Mn along the Co nanoparticle, possibly because of their lower contents. In contrast, the C signals were still quite strong beyond the Co nanoparticles, further substantiating the color-mapping result. As carburization gradually accumulates, Co particles and the carbon layer undergo further reaction under reaction conditions and are gradually and partially converted into Co₂C. To validate this hypothesis, we performed HRTEM test and analyzed the lattice fringes of an individual Co particle and the surroundings (Fig. 1e). We found that the Co(111) crystals and the periphery was decorated by smaller Co₂C ensembles. Notably, the Co particle was only partially decorated by several Co₂C ensembles, instead of fully covered by the latter. Therefore, we trust it is more appropriate to use interphases rather than core-shell structure to depict the key structure of the CoMnCs/C catalyst.

Fig. 1. Design of efficient syngas-to-oxygenates/olefins catalysts. (a) CO conversion vs. total selectivity of alcohols and olefins on carbon-supported Co catalysts with different promoters. (b) Comparison on the syngas-to-oxygenates/olefins performance on representative Co-based catalysts. Reaction conditions: 210 °C, 3.0 MPa, $H_2/CO = 2.0$, $F = 65 \text{ ml min}^{-1}$ (See Table S1 for details). (c-e) STEM and HRTEM images of $Co_{0.5}Mn_{0.1}Cs/C$ catalyst. (c) EDX mapping images. (d) Line scanning profiles (inset). (f) The reaction coordinate of $C_5H_{10}^*$ desorption, hydrogenation, and CO insertion pathways on Co(111), Co@Co₂C, and CoCs@Co₂C models. Bader charge analysis were shown in the inset.

We have now accommodated these new results and discussions into the revised manuscript.

“These observations strongly suggested the co-existence of both Co^0 and *in situ* formed Co_2C on the catalyst surface, which was further confirmed by detailed high-resolution transmission in situ PXRD and electron microscopy observations. The in situ PXRD experiments verified the formation of both fcc-Co and Co_2C on CoMnCs/C under syngas treatment conditions but only fcc-Co after H_2 reduction (Fig. S5), hinting the critical role of reaction atmospheres on the catalyst structure. Scanning transmission

electron microscopy (STEM) coupled with elemental color mapping image of the spent catalyst unambiguously pointed to the surface enriching of carbon species around Co nanoparticle of CoMnCs/C catalyst (Fig. 1c). This was further supported by line-scanning of an individual nanoparticle, showing relatively even distributions of Cs and Mn across the nanoparticles but much higher C signal than that of Co at the periphery (Fig. 1d). As carburization gradually accumulated, Co particles and the carbon layer underwent further reaction under reaction conditions and were gradually and partially converted into Co_2C . To validate this hypothesis, HRTEM image of the spent catalyst was acquired (Fig. 1e). Lattice fringe analysis revealed the structure of fcc-Co(111) in an individual particle decorated by small Co_2C ensembles (Fig. 1e). Noted that Co_2C was present as small patches on the Co surface, forming $\text{Co}@Co_2\text{C}$ interphases, but did not fully cover the latter.”

Reviewer #3:**Comment:**

The authors report an integration catalytic strategy for long-chain alcohol synthesis from syngas conversion using two separate reactors and a three-stage packed catalysts process, which realizes the tandem connection of syngas to long-chain oxygenates/olefins and the subsequent hydroformylation/hydrogenation to produce C₄₊ alcohols. The experiment achieves awfully excellent catalytic performance with 80% C₄₊ alcohols selectivity and 1% CO₂ selectivity that surpasses the previous reports of the related syngas conversion to alcohols. Generally, the present work is featured by process integration and the optimized combination of several different types of functional catalysts, which is not significantly innovative with respect to the design concept because similar tandem patterns have been commonly reported. Besides, the author's discussion about the cognition of the cooperative catalysis between different modules is somewhat general. The specific comments or queries are listed as follows.

Reply:

We sincerely appreciate the time and effort the Reviewer #3 have taken to review our manuscript. Your constructive feedback is valuable in helping us refine and improve our work. We also thank you for recognizing the merits of our work. Despite of the outstanding catalytic performance reached in this work, we kindly remind the Reviewer for the following three key innovative aspects:

- i) It is the first work targeting C₄₊ alcohols synthesis from syngas. A significant boost in C₄₊ alcohol selectivity (80%) at a reasonable CO conversion has been reached as compared with the previous works (Table S5).
- ii) Although tandem catalysis has been adopted in syngas conversions, to the best of our knowledge, it is the first report that combines three distinctive reactions (syngas-to-oxygenates/olefins + hydroformylation + hydrogenation) in two reactors for syngas conversion. This is a new catalytic process specifically designed for higher-alcohol synthesis with unprecedented selectivity. In particular, it is also the first that the CoCs@Co₂C catalyst was proposed for the effective targeted synthesis of oxygenates and olefins.

iii) We have also examined the complex compatibility of the three significantly different catalytic modules by combining broad screening of catalytic materials, probe reactions, and DFT simulations. This innovative approach can provide a rationale for the catalyst design with complicate feeds in other tandem reactions.

1) The authors presume that the CoMnCs/C catalysts used for syngas conversion to oxygenates/olefins possess the interfacial structure of CoCs@Co₂C, plus Co(111) and Co@Co₂C, which lacks characterization evidence. Does CoCs@Co₂C refer a core-shell structure? However, it was not obviously observed in the TEM image (Fig S4). In addition, this model also ignores the role of manganese promoters.

Reply:

We thank the Reviewer for the insightful instruction. We sincerely apologize for any confusion caused by our previous message. We should stress that here Co@Co₂C was used to indicate the formation of interphases between Co and Co₂C entities, but not core-shell structures.

In order to clarify the catalyst structure, we have supplemented additional experiments (in situ PXRD, STEM and HRTEM) for a more detailed description (Fig. S5 and Fig.1). The in situ PXRD tests of CoMnCs/C under different atmospheres revealed marked differences on the catalyst compositions. Specifically, only fcc-Co was formed subject to H₂ reduction even up to 430 °C, but both fcc-Co and Co₂C were generated after syngas treatment at 210 °C at prolonged time. The latter phenomenon evidenced the coexistence of both Co species, and indicated that Co₂C was likely in situ formed only under the syngas conversion conditions.

Fig. S5. In situ PXRD patterns of the CoMnCs/C catalysts under different conditions: (a) in H₂ atmosphere (430 °C, 0.1 MPa); (b) syngas ambience (210 °C, 0.1 MPa).

Fig. 1. Design of efficient syngas-to-oxygenates/olefins catalysts. (a) CO conversion vs. total selectivity of alcohols and olefins on carbon-supported Co catalysts with different promoters. (b) Comparison on the syngas-to-oxygenates/olefins performance on representative Co-based catalysts. Reaction conditions: 210 °C, 3.0 MPa, $H_2/CO = 2.0$, $F = 65 \text{ ml min}^{-1}$ (See Table S1 for details). (c-e) STEM and HRTEM images of $Co_{0.5}Mn_{0.1}Cs/C$ catalyst. (c) EDX mapping images. (d) Line scanning profiles (inset). (f) The reaction coordinate of $C_5H_{10}^*$ desorption, hydrogenation, and CO insertion pathways on Co(111), Co@Co₂C, and CoCs@Co₂C models. Bader charge analysis were shown in the inset.

To further visualize the more precise catalyst structures, we performed additional electron microscopic analyses of the spent CoMnCs/C catalysts. First, we acquired STEM with elemental color mapping image of the spent CoMnCs/C, showing the apparent enrichment of carbon species around the Co nanoparticle (Fig. 1c). Then we also performed a line scanning of an individual particle as shown in Fig. 1d. One can see the relatively even distributions of Cs and Mn along the Co nanoparticle, possibly because of their lower contents. In contrast, the C signals were still quite strong beyond the Co nanoparticles, further substantiating the color-mapping result. As carburization

gradually accumulates, Co particles and the carbon layer undergo further reaction under reaction conditions and are gradually and partially converted into Co_2C . To validate this hypothesis, we performed HRTEM test and analyzed the lattice fringes of an individual Co particle and the surroundings (Fig. 1e). We found that the Co(111) crystals and the periphery was decorated by smaller Co_2C ensembles. Notably, the Co particle was only partially decorated by several Co_2C ensembles, instead of fully covered by the latter. Therefore, we trust it is more appropriate to use interphases rather than core-shell structure to depict the key structure of the CoMnCs/C catalyst.

We have now accommodated these new results and discussions into the revised manuscript.

“These observations strongly suggested the co-existence of both Co^0 and *in situ* formed Co_2C on the catalyst surface, which was further confirmed by detailed high-resolution transmission in situ PXRD and electron microscopy observations. The in situ PXRD experiments verified the formation of both fcc-Co and Co_2C on CoMnCs/C under syngas treatment conditions but only fcc-Co after H_2 reduction (Fig. S5), hinting the critical role of reaction atmospheres on the catalyst structure. Scanning transmission electron microscopy (STEM) coupled with elemental color mapping image of the spent catalyst unambiguously pointed to the surface enriching of carbon species around Co nanoparticle of CoMnCs/C catalyst (Fig. 1c). This was further supported by line-scanning of an individual nanoparticle, showing relatively even distributions of Cs and Mn across the nanoparticles but much higher C signal than that of Co at the periphery (Fig. 1d). As carburization gradually accumulated, Co particles and the carbon layer underwent further reaction under reaction conditions and were gradually and partially converted into Co_2C . To validate this hypothesis, HRTEM image of the spent catalyst was acquired (Fig. 1e). Lattice fringe analysis revealed the structure of fcc-Co(111) in an individual particle decorated by small Co_2C ensembles (Fig. 1e). Noted that Co_2C was present as small patches on the Co surface, forming Co@ Co_2C interphases, but did not fully cover the latter. To understand the individual roles of Mn and Cs on the catalyst structure, the STEM with line-scanning images of Co/C, CoMn/C, and CoCs/C

catalysts, together with their corresponding HRTEM images, were acquired (Fig. S6). These results showed that the addition of Mn alone also promoted the clear carbon enrichment on Co particles and *in situ* formation of Co@Co₂C structure during FTS reaction, which were not observed on Co/C and CoCs/C. Considering that Cs was mainly enriched on Co nanoparticles for the best-performing CoMnCs/C catalyst (potentially existing as Cs⁺ species due to the nature of electronic promoter, also supported by DFT calculations in Table S2-3 and Fig. S9-10), the CoCs@Co₂C model was constructed for further simulation.”

Fig. S6. (a-d) STEM images, (e-g) Line scanning profiles and (i-l) HRTEM images of CoMnCs/C, CoMn/C, CoCs/C and Co/C

2) Personally, I think it is not of great significance to discuss the ecological benefits of the integrated catalytic strategy for syngas conversion in this work because the results are only obtained based on the small laboratory experiments without pilot or benchmark test.

Reply:

We fully agree with the Reviewer that the lab-scale experiments are far from enough to reach a solid conclusion on the ecological benefits. In fact, we have previously

conducted a life cycle analysis of this newly established process. But for the same reason, we decided to remove it and focus more on the comparative analysis of different syngas-to-C₄₊ alcohol technologies. We trust this information is still useful to give the readers a broader landscape of the technological developments.

Considering your kind advice, we have decided to lower our tune of this section and focus more on the comparison on the different process. The following two actions are taken in the revision:

We have replaced the section title “**Ecological benefits**” with “**Comparative analysis of syngas-to-C₄₊ alcohol synthesis pathways**”.

We have removed the sentence related to ecological implication:

~~“From an ecological perspective, the system provides remarkable benefits by minimizing CO₂ emissions. Fig. 4b compares CO₂ selectivity across the three routes.”~~

3) In the text (Line 128~132), the authors mentioned that the single Mn and Cs modified Co-based catalysts would increase the selectivity of ROH and Olefins, but would reduce CO conversion. For example, the CO conversion over Co_{0.5}Mn/C and Co_{0.1}Cs/C were 20% and 11%, respectively, but that was increased to 29% on Mn and Cs co-modified Co-based catalysts. Why?

Reply:

We thank the Reviewer for the valuable comments. Mn additives could form a relative “C-rich and H-lean” surface chemical environment, so that the Co@Co₂C interface could be in situ reconstructed during the reaction process, and the olefins and alcohols selectivity could be improved (ACS Catal., 2017, 8, 228-241)¹².

Furthermore, the co-addition of Cs and Mn promoted the dispersion of Co species as compared with the counterparts with individual promoters. Based on the TEM analysis of the catalysts after the reaction, it was observed that the particle size of CoMnCs/C catalyst was significantly reduced ($D = 6.9$ nm), while CoMn/C and CoCs/C catalysts showed much large mean particle sizes, 8.6 and 8.0 nm, respectively. The increased dispersion of the former may thereby increase the exposed Co nanoparticle active sites and improve the catalytic activity.

In the manuscript, we have provided corresponding supplements:

“In addition, the addition of Cs promoted the dispersion of Co species. Statistic particle counting from TEM images of the spent catalysts revealed that, the particle size of CoMnCs/C catalyst was significantly smaller than the counterparts with individual promoters ($D = 6.9$ vs. 8.0 , 8.6 nm, Fig. S7). The better dispersion could increase the exposed Co nanoparticle active sites, thereby improving the catalytic activity.”

Fig. S7. (a-c) STEM image of the CoMn/C, CoCs/C and CoMnCs/C catalysts at low magnification. (d-f) Particle distribution over the CoMn/C, CoCs/C and CoMnCs/C catalysts, respectively.

- 4) The space-time yield of C_{4+} alcohols was calculated based on the single CoMnCs/C catalyst. This calculation method might be not reasonable because the other two modular catalysts also synergistically promote the formation of C_{4+} alcohols products.

Reply:

We thank the reviewers for their insightful suggestions. In response to the Reviewer's suggestion, we have provided STY calculated based on all catalysts. The Reviewer is kindly asked to refer to Tables S6-S10.

Table S6. The catalytic performances of different combinations of catalysts in the tandem system for the syngas-to- C_{4+} alcohols conversion.

Catalysts	SC_{4+OH}	SC_{6+OH}	STY_{OH}	$STY_{C_{2+OH}}$	$STY_{C_{4+OH}}$	$STY_{C_{6+OH}}$
-----------	-------------	-------------	------------	------------------	------------------	------------------

	%	%	$\text{g kg}_{\text{cat}}^{-1} \text{h}^{-1}$	$\text{g kg}_{\text{cat}}^{-1} \text{h}^{-1}$	$\text{g kg}_{\text{cat}}^{-1} \text{h}^{-1}$	$\text{g kg}_{\text{cat}}^{-1} \text{h}^{-1}$
A	56.4	33.7	78.3	76.7	70.8	32.0
B	68.5	50.1	119.4	118.4	111.9	74.3
C	73.0	35.7	77.1	72.8	59.2	29.9

A: $\text{Co}_{0.5}\text{Mn}_{0.1}\text{Cs/C}|\text{Rh}/3\text{v-POPs-PPH}_3|5\text{Pt}4\text{Sn}/\text{TiO}_2$.

B: $\text{Co}_{0.5}\text{Mn}_{0.1}\text{Cs/MC}|\text{Rh}/3\text{v-POPs-BP\&PPH}_3|\text{CuZrO}_2/\text{SiO}_2$.

C: $\text{Co}_{0.5}\text{Mn}_{0.1}\text{Cs/C}|\text{Rh}/3\text{v-POPs-PPH}_3|\text{CuZrO}_2/\text{SiO}_2$.

Reaction conditions: $\text{H}_2/\text{CO} = 1.5$, $F = 130 \text{ ml min}^{-1}$, $T = 210 \text{ }^\circ\text{C}$ (syngas-to-oxygenates/olefins), $T = 140 \text{ }^\circ\text{C}$ (hydroformylation/hydrogenation), and $P = 3 \text{ MPa}$.

Table S7. Influence of H_2/CO ratios on the catalytic performances of $\text{Co}_{0.5}\text{Mn}_{0.1}\text{Cs/C}|\text{Rh}/3\text{v-POPs-PPH}_3|\text{CuZrO}_2/\text{SiO}_2$ tandem system for the syngas-to- C_{4+} alcohols conversion.

H_2/CO mol mol^{-1}	$S_{\text{C}_{4+\text{OH}}}$ %	$S_{\text{C}_{6+\text{OH}}}$ %	STY_{OH} $\text{g kg}_{\text{cat}}^{-1} \text{h}^{-1}$	$STY_{\text{C}_{2+\text{OH}}}$ $\text{g kg}_{\text{cat}}^{-1} \text{h}^{-1}$	$STY_{\text{C}_{4+\text{OH}}}$ $\text{g kg}_{\text{cat}}^{-1} \text{h}^{-1}$	$STY_{\text{C}_{6+\text{OH}}}$ $\text{g kg}_{\text{cat}}^{-1} \text{h}^{-1}$
1/2	75.0	46.8	31.4	30.2	28.7	14.3
1/1	67.9	43.0	41.9	39.3	34.0	12.7
3/2	73.0	35.7	78.3	76.7	70.8	32.0
2/1	69.5	29.2	68.1	67.8	59.2	23.0
3/1	69.2	39.1	58.5	55.9	52.2	24.2

Reaction conditions: $F = 130 \text{ ml min}^{-1}$, $210 \text{ }^\circ\text{C}$ (syngas-to-oxygenates/olefins), $140 \text{ }^\circ\text{C}$ (hydroformylation/hydrogenation), and 3 MPa .

Table S8. Influence of bed temperatures of the syngas-to-oxygenates/olefins reactor on the catalytic performances of $\text{Co}_{0.5}\text{Mn}_{0.1}\text{Cs/C}|\text{Rh}/3\text{v-POPs-PPH}_3|\text{CuZrO}_2/\text{SiO}_2$ tandem system for the syngas-to- C_{4+} alcohols conversion.

$T / ^\circ\text{C}$	$S_{\text{C}_{4+\text{OH}}}$ %	$S_{\text{C}_{6+\text{OH}}}$ %	STY_{OH} $\text{g kg}_{\text{cat}}^{-1} \text{h}^{-1}$	$STY_{\text{C}_{2+\text{OH}}}$ $\text{g kg}_{\text{cat}}^{-1} \text{h}^{-1}$	$STY_{\text{C}_{4+\text{OH}}}$ $\text{g kg}_{\text{cat}}^{-1} \text{h}^{-1}$	$STY_{\text{C}_{6+\text{OH}}}$ $\text{g kg}_{\text{cat}}^{-1} \text{h}^{-1}$
210	73.0	35.7	78.3	76.7	70.8	32.0
220	52.8	37.1	128.8	126.9	108.7	76.4
230	43.4	26.3	143.3	132.6	118.7	60.5
240	9.0	5.3	50.9	47.9	41.5	14.3

Reaction conditions: $\text{H}_2/\text{CO} = 1.5$, $F = 130 \text{ ml min}^{-1}$, $140 \text{ }^\circ\text{C}$ (hydroformylation/hydrogenation), and 3 MPa .

Table S9. Influence of GHSV (based on all catalysts) on the catalytic performances of $\text{Co}_{0.5}\text{Mn}_{0.1}\text{Cs/C}|\text{Rh}/3\text{v-POPs-PPH}_3|\text{CuZrO}_2/\text{SiO}_2$ tandem system for the syngas-to- C_{4+} alcohols conversion.

GHSV / h^{-1}	$S_{\text{C}_{4+\text{OH}}}$ %	$S_{\text{C}_{6+\text{OH}}}$ %	STY_{OH} $\text{g kg}_{\text{cat}}^{-1} \text{h}^{-1}$	$STY_{\text{C}_{2+\text{OH}}}$ $\text{g kg}_{\text{cat}}^{-1} \text{h}^{-1}$	$STY_{\text{C}_{4+\text{OH}}}$ $\text{g kg}_{\text{cat}}^{-1} \text{h}^{-1}$	$STY_{\text{C}_{6+\text{OH}}}$ $\text{g kg}_{\text{cat}}^{-1} \text{h}^{-1}$
800	67.8	42.3	70.2	63.7	60.6	27.4
1600	73.0	35.7	78.3	76.7	70.8	32.0

2400	75.8	31.5	75.7	74.5	68.5	15.7
3200	74.8	50.8	76.3	75.4	71.5	43.4
4000	60.5	43.3	67.2	62.8	55.0	30.2

Reaction conditions: $H_2/CO = 1.5$, 210 °C (syngas-to-oxygenates/olefins), 140 °C (hydroformylation/hydrogenation), and 3 MPa.

Table S10. Influence of pressures on the catalytic performances of $Co_{0.5}Mn_{0.1}Cs/C|Rh/3v-POPs-PPh_3|CuZrO_2/SiO_2$ tandem system for the syngas-to- C_{4+} alcohols conversion.

$P/$ MPa	$S_{C_{4+OH}}$ %	$S_{C_{6+OH}}$ %	STY_{OH} g kg _{cat} ⁻¹ h ⁻¹	$STY_{C_{2+OH}}$ g kg _{cat} ⁻¹ h ⁻¹	$STY_{C_{4+OH}}$ g kg _{cat} ⁻¹ h ⁻¹	$STY_{C_{6+OH}}$ g kg _{cat} ⁻¹ h ⁻¹
1	59.8	41.6	17.8	10.4	16.1	17.7
2	72.0	50.3	30.1	29.7	27.6	17.6
3	73.0	35.7	78.3	76.7	70.8	32.0
8	79.8	45.9	97.1	96.7	89.7	45.7

Reaction conditions: $H_2/CO = 1.5$, $F = 130$ ml min⁻¹, 210 °C (syngas-to-oxygenates/olefins), 140 °C (hydroformylation/hydrogenation).

5) The specific composition of the hydrogenation catalysts should be provided for deep understanding of the catalyst.

Reply:

Thank you for the relevant comments. We have supplemented the synthesis method and composition of Cu-based hydrogenation catalysts in the manuscript (Table S12).

Table S12. Synthetic approach and compositional for catalysts.

Catalyst	Synthesis method	Elements of composition	Nominal Content / wt.%	Measured content / wt.%
Co/C	wetness impregnation	Co	15	15.2 ^a
Co _{0.5} Mn/C	wetness impregnation	Co Mn	15 0.5	14.9 ^a 0.5 ^a
Co _{0.1} Cs/C	wetness impregnation	Co Cs	15 0.1	15.1 ^a 0.1 ^a
Co _{0.5} Mn _{0.1} Cs/C	wetness impregnation	Co Mn	15 0.5	14.9 ^a 0.5 ^a

		Cs	0.1	0.1 ^a
Rh/3v-POPs-PPh ₃	post-loading	Rh	0.125	0.1 ^a
Rh/POPs-BP&PPh ₃	post-loading	Rh	0.125	0.1 ^a
CuZrO ₂ /SiO ₂	ammonia	Cu	20	19.0 ^b
	evaporation	Zr	-	28.0 ^b
Cu/SiO ₂	ammonia	Cu	20	22.0 ^b
	evaporation			

^a Determined from ICP-OES.

^b Determined from XRF.

- 6) In the section of Catalyst evaluation, the pretreatment of the catalyst (e.g. reduction method) should be described in detail, because catalyst pretreatment has a great influence on the reaction performance. Moreover, in the characterization measurements, the authors should make clear how the samples were treated? What is the state of these samples (oxidic or reduced state, fresh or spent)?

Reply:

We thank the Reviewers for the valuable comments, which we have supplemented in the catalyst evaluation, as detailed below:

“The Co-based syngas-to-oxygenates/olefins catalysts were reduced in situ in a fixed-bed reactor for 10 h at 430 °C, 2000 h⁻¹, 0.1 MPa, under an H₂ atmosphere before reaction.

Rh-based heterogeneous hydroformylation catalysts had no reduction treatment process.

Cu-based hydrogenation catalysts were reduced in situ in fixed bed for 2 h at 260 °C, 2000 h⁻¹, 0.1 MPa under H₂ atmosphere before reaction.

The Pt- and Ru-based hydrogenation catalysts were reduced in situ in a fixed-bed for 4 h at 450 °C and 2000 h⁻¹ under H₂ atmosphere before reaction.”

In addition, the states of the catalyst samples for various characterizations have been annotated in the corresponding figures.

We have supplemented in the characterization measurements, as detailed below:

“Elemental content testing of samples by Inductively Coupled Plasma (ICP) technique

on the Perkin Elmer ICP-OES 7300DV equipment. The metal content in the Cu-based catalysts was determined by X-ray fluorescence spectroscopy (XRF).

The ac-HAADF-STEM and STEM images determination of catalyst samples was acquired on the JEM-ARM200F STEM/TEM (resolution of 0.08 nm) and JEM-F200 instrument of JEOL, respectively. Sample was dispersed in EtOH and placed onto Cu grids.

The CO pulse chemisorption experiment was carried out on the Zeton Altramira AMI-300 chemisorption instrument. About 100 mg samples were packed into a U-shaped tube using He as carrier gas at a flow rate of 30 mL min⁻¹. The catalyst sample was first heated to 120 °C at a heating rate of 10 °C min⁻¹ in a flow of He, purged for 1 h, and cooled to 50 °C. Then, 10% CO/He mixture was quantitatively pulsing injected until adsorption saturation, and the released CO was detected by a TCD detector.”

7) In Fig 2b, it demonstrated CO (not valeraldehyde) conversion. This might be written error because the image is related the probe test of valeraldehyde hydrogenation.

Reply:

We are sorry for the mistake that has now been corrected in the revision.

Fig. 2. Compatibility assessment between syngas-to-oxygenates/olefins and the downstream hydroformylation or hydrogenation processes. (a) Catalyst performances between syngas-to-oxygenates/olefins plus hydroformylation or hydrogenation processes (left) and syngas-to-oxygenates/olefins plus

hydrogenation (right). Reaction conditions: 210 °C (syngas-to-oxygenates/olefins) and 140 °C (hydroformylation/hydrogenation), $H_2/CO = 1.5$, 3.0 MPa, $F = 65 \text{ ml min}^{-1}$. (b) Catalytic performances of 5Pt/TiO₂ and 5Pt4Sn/TiO₂ for valeraldehyde hydrogenation in H₂ and syngas streams. Reaction conditions: 5Pt/TiO₂ or 5Pt4Sn/TiO₂, 1.5 g; $T = 140 \text{ °C}$; $P = 3.0 \text{ MPa}$; F (valeraldehyde) = 0.01 ml min⁻¹; F (H₂) or F (H₂/CO = 2) = 30 ml min⁻¹. (c) WGS performances and CO uptakes of CuZnAl, Cu/SiO₂, and CuZrO₂/SiO₂. Reaction conditions: catalysts 1.5 g; $T = 140 \text{ °C}$; $P = 3.0 \text{ MPa}$; F (H₂O) = 0.01 ml min⁻¹; F (CO) = 30 ml min⁻¹. (d) Hexanal hydrogenation and WGS reaction pathways on CuZrO₂ model system.

8) From Table S6, it can be seen that the CO₂ selectivity of Co_{0.5}Mn_{0.1}Cs/C|Rh₁/3v-POPs-PPh₃|CuZrO₂/SiO₂ should show an upward trend with the increase of reaction temperature. Why is the CO₂ selectivity at 230 °C lower than 220 °C?

Reply:

Thank you for your insightful question. We need to clarify that the temperature's impact on the performance of the tandem system was performed in a single test by ramping up the temperature from 210 to 240 °C. Besides, each of the original reported data were collected after stabilization at the aimed temperatures for 24 h.

One can see that when the temperature was elevated from 210 to 220 °C, the CO conversion significantly increased from 15% to 37%, accompanied with great fluctuations in the reaction system, which we suspect might result in higher CO₂ selectivity. In contrast, when the temperature was further increased from 220 to 230 °C, the CO conversion slightly increased from 36.5% to 42.5%, while the system remained relatively stable with milder impact on CO₂ selectivity.

To minimize the impact of the dramatic fluctuation caused by the temperature swing from 210 to 220 °C, we have repeated the set of experiments. In particular, we have performed three consecutive samplings at 220 °C for a longer duration (Please kindly refer to Table R3). One can see a higher initial CO conversion at 24 h. However, both the CO conversion and product selectivity stabilized at comparable levels during 48-72 h. Specifically, the CO₂ selectivity stabilizes at approximately 4% which is about the same as the data at 230 °C. We have now replaced the original 24 h data with the stabilized data at 72 h.

Table R3. Influence of bed temperatures of the syngas-to-oxygenates/olefins reactor on the catalytic performances of Co_{0.5}Mn_{0.1}Cs/C|Rh/3v-POPs-PPh₃|CuZrO₂/SiO₂ tandem system for the syngas-to-

C₄₊ alcohols conversion

T / °C	X _{CO} / %	S / %					
		CH ₄	CO ₂	C ₂₊	C ₂₊ ⁼	aldehydes	alcohols
220-24 h	36.5	8.6	7.3	21.3	0.6	7.1	55.1
220-48 h	33.2	4.8	4.2	20.5	1.3	5.4	63.8
220-72 h	33.8	5.9	4.3	18.6	1.8	6.9	62.5

Reaction conditions: H₂/CO = 1.5, *F* = 130 ml min⁻¹, 140 °C (hydroformylation/hydrogenation), and 3 MPa.

Supplementary references

- 1 Ziang Zhao, et al.. Integrated Fischer-Tropsch synthesis and heterogeneous hydroformylation technologies toward high-value commodities from syngas. *Chin. J. Catal.*, **73**, 16-28, (2025).
- 2 Jiang, M. et al. Effect of different synthetic routes on the performance of propylene hydroformylation over 3v-PPh₃ polymer supported Rh catalysts. *React. Kinet, Mech. Cat.*, **116**, 223-234, (2015).
- 3 Jiang, M. et al. Ultrastable 3v-PPh₃ polymers supported single Rh sites for fixed-bed hydroformylation of olefins. *J. Mol. Catal. A*, **404**, 211-217, (2015).
- 4 Li, Y. et al. Interfacial Fe₅C₂-Cu catalysts toward low-pressure syngas conversion to long-chain alcohols. *Nat. Commun.* **11**, 61, (2020).
- 5 Du, H. et al. Higher alcohols synthesis via CO hydrogenation on Fe-promoted Co/AC catalysts. *Catal. Today* **281**, 549-558, (2017).
- 6 Hu, J. et al. Edge-rich molybdenum disulfide tailors carbon-chain growth for selective hydrogenation of carbon monoxide to higher alcohols. *Nat. Commun.* **14**, 6808, (2023).
- 7 Asundi, A. S. et al. Enhanced alcohol production over binary Mo/Co carbide catalysts in syngas conversion. *J. Catal.* **391**, 446-458, (2020).
- 8 Sun, J. et al. Promotional effects of cesium promoter on higher alcohol synthesis from syngas over cesium-promoted Cu/ZnO/Al₂O₃ Catalysts. *ACS Catal.* **6**, 5771-5785, (2016).
- 9 Mo, X., Tsai, Y.-T., Gao, J., Mao, D. & Goodwin, J. G. Effect of component interaction on the activity of Co/CuZnO for CO hydrogenation. *J. Catal.* **285**, 208-215, (2012).
- 10 Zhao, Z., Li, Y., Zhu, H., Lyu, Y. & Ding, Y. A review of Co/Co₂C-based catalysts in Fischer – Tropsch synthesis: from fundamental understanding to industrial applications. *Chem. Commun.* **59**, 3827, (2023).
- 11 Li, Y. et al. Tuning surface oxygen group concentration of carbon supports to promote Fischer-Tropsch synthesis. *Appl. Catal. A: Gen.* **613**, 118017, 118017 (2021).
- 12 Zhao, Z. et al. Insight into the formation of Co@Co₂C catalysts for direct synthesis of higher alcohols and olefins from syngas. *ACS Catal.* **8**, 228-241, (2017).
- 13 Zhao, Z. et al. Tuning the Fischer-Tropsch reaction over Co_xMn_yLa/AC catalysts toward alcohols: Effects of La promotion. *J. Catal.* **361**, 156-167, (2018).
- 14 Zhao, Z. et al. Increasing the activity and selectivity of Co-based FTS catalysts supported by carbon materials for direct synthesis of clean fuels by the addition of chromium. *J. Catal.* **370**, 251-264, (2019).
- 15 Du, H. et al. Study on CaO-promoted Co/AC catalysts for synthesis of higher alcohols from syngas. *Fuel* **182**, 42-49, (2016).
- 16 Cui, W. et al. In situ encapsulated Co/MnO_x nanoparticles inside quasi-MOF-74 for the higher alcohols synthesis from syngas. *Appl. Catal. B: Environ.* **278**, 119262, (2020).
- 17 Xiang, Y., Barbosa, R. & Kruse, N. Higher alcohols through CO hydrogenation over CoCu catalysts: Influence of precursor activation. *ACS Catal.* **4**, 2792-2800, (2014).
- 18 Xiang, Y. et al. Long-chain terminal alcohols through catalytic CO hydrogenation. *J. Am. Chem. Soc.* **135**, 7114-7117, (2013).
- 19 Xiang, Y., Barbosa, R., Li, X. & Kruse, N. Ternary cobalt-copper-niobium catalysts for the selective CO hydrogenation to higher alcohols. *ACS Catal.* **5**, 2929-2934, (2015).
- 20 Xiang, Y. & Kruse, N. Tuning the catalytic CO hydrogenation to straight- and long-chain aldehydes/alcohols and olefins/paraffins. *Nat. Commun.* **7**, 13058, (2016).

- 21 Qin, T. *et al.* Tuning chemical environment and synergistic relay reaction to promote higher alcohols synthesis via syngas conversion. *Appl. Catal. B: Environ.* **285**, 119840, (2021).
- 22 Huang, C., Zhu, C., Zhang, M., Chen, J. & Fang, K. Design of efficient ZnO/ZrO₂ modified CuCoAl catalysts for boosting higher alcohol synthesis in syngas conversion. *Appl. Catal. B: Environ.* **300**, 120739 (2022).
- 23 Wang, J. *et al.* Dihydroxyacetone valorization with high atom efficiency via controlling radical oxidation pathways over natural mineral-inspired catalyst. *Nat. Commun.* **12**, 6840, (2021).
- 24 Lin, T. *et al.* Direct production of higher oxygenates by syngas conversion over a multifunctional catalyst. *Angew. Chem. Inter. Ed.* **58**, 4627-4631, (2019).
- 25 Jeske, K. *et al.* Direct conversion of syngas to higher alcohols via tandem integration of Fischer-Tropsch synthesis and reductive hydroformylation. *Angew. Chem. Inter. Ed.* **61**, e202201004, (2022).
- 26 Luan, X. *et al.* Selective conversion of syngas into higher alcohols via a reaction-coupling strategy on multifunctional relay catalysts. *ACS Catal.* **10**, 2419-2430, (2020).
- 27 Huang, C. *et al.* Direct conversion of syngas to higher alcohols over a CuCoAl|t-ZrO₂ multifunctional catalyst. *ChemCatchem* **13**, 3184-3197, (2021).
- 28 Li, C. *et al.* Xantphos doped Rh/POPs-PPh₃ catalyst for highly selective long-chain olefins hydroformylation: Chemical and DFT insights into Rh location and the roles of Xantphos and PPh₃. *J. Catal.* **353**, 123-132, (2017).
- 29 Fan, B. *et al.* Elucidation of hemilabile-coordination-induced tunable regioselectivity in single-site Rh-catalyzed heterogeneous hydroformylation. *Nat. Commun.* **15**, 6967, (2024).

Response to Reviewers

Comments in blue - Replies in black - Actions in bold - Citations in *italic*

Reviewer #2 (Remarks to the Author):

The authors have included extra information to support their results. The HRTEM images are particularly helpful.

We thank the Reviewer for recognizing our effort in this revision, and your further insightful comments that are invaluable for improving the quality of this work.

The DFT calculations remain problematic. The HRTEM image (Figure 1e) suggests Co₂C islands are formed on metallic Co. The DFT model is the inverse: Co island on Co₂C.

We fully accept the criticism on the catalyst model that deviates from the characterization results. In the initial modeling, the main objective was to establish the interface model between Co and Co₂C. We had two options, either building Co₂C on the top of Co nanoparticles (Co₂C-Co) or the inversed model (Co-Co₂C). Considering that the former will lead to a much larger model, and ultimately needs longer calculation time, the inversed model of Co on top of Co₂C was selected for the modeling in the previous manuscript. To keep in line with the characterization data, **we have now switched back to the model of Co₂C on top of Co nanoparticles and completely replaced in the revised manuscript (Figs. 1f, and S8-10, Table S3)**. It should be mentioned that, similar trends were observed for the two selected models, thus hinting the critical role of Co and Co₂C interface structures in the syngas-to-oxygenates/olefins.

Recent studies have conclusively shown that K is present as K₂O during FT, and the O atoms play a crucial role in the promotion. The model with a single Cs atom (not Cs⁺) and the active site next to it is not representative for the actual sites.

Thank you for the insightful comments that we fully agree. **We have taken this into account when binding the revised model for the CoMnCs/C catalyst.** Specifically, in the new model, Cs₂O species are placed at the interface between Co and Co₂C. **We have updated the simulations of the three key fundamental steps, including desorption, hydrogenation, and carbon monoxide insertion (Fig. 1f, Table S3 and S9-S10).**

Experimental and modeling studies have shown that alkylidyne species are the key intermediates, not olefins. If metallic Co sites are available, olefin hydrogenation will still dominate over CO insertion and olefin desorption, because of the much lower barrier.

The Reviewer is kindly reminded that it is C_nH_{2n}^{*}, but not alkylidyne, that C₅H₁₀^{*} was adopted for the modeling in our work. In fact, C_nH_{2n}^{*} has been widely used as the intermediate in syngas-to-alcohols synthesis in the previous studies (*Appl. Phys. Rev.*, 2025, 12, 011311). “*The metal cobalt first produces CH_x^{*} through the adsorption of CO by dissociation and then promotes the formation of C_nH_{2n}^{*} intermediates through a chain reaction. Due to the short distance between Co⁰ and Co₂C, C_nH_{2n}^{*} can quickly transfer from Co⁰ to Co₆Mo₆C. At the same time, Co₆Mo₆C assisted non-dissociative adsorption of CO to produce active CO^{*}, which was inserted into adjacent C_nH_{2n}^{*} species to form acyl intermediate C_nH_{2n}CO^{*}, followed by hydrogenation to form the final C_{n+1}H_{2n+1}OH alcohols.*”

Regarding the hydrogenation of olefins on metallic Co sites, this did not happen on **the CoMnCs/C catalyst** likely due to its unique structure. At first, the Co and Co₂C interface structure inhibits the

hydrogenation of $C_5H_{10}^*$ to form alkanes, as confirmed by the DFT calculations. The results in Fig. 1f showed that the energy barriers of this step on Co(111) is significantly lower with 0.36 eV compared to 1.62 eV on Co_2C-Co and 1.75 eV on Cs_2O-Co_2C-Co . Furthermore, Co nanoparticles are in situ encapsulated by Co_2C , as evidenced by the detailed HRTEM analysis (Fig. R1). This distinct structure might also reduce the H_2 adsorption capacity, thus inhibiting the further hydrogenation of olefins.

“The interfacial sites of Co_2C and Co nanoparticles, denoted hereafter as Co_2C-Co , are crucial for the preferential formation of alcohols^{33, 34}, and the addition of Cs can further enhance alcohol formation. Based on these observations and the catalyst characterization data, a new catalyst model of Cs_2O-Co_2C-Co , plus Co(111) and Co_2C-Co , was constructed to further elucidate their effects on the syngas-to-oxygenates/olefins activity and product distribution through DFT simulations (Fig. S8, Table S3). To explain the different product distributions, three key fundamental steps including the desorption, hydrogenation, and CO insertion of $C_5H_{10}^$ intermediates were simulated (Figs. 1f and S9-S10). For Co(111), the energy barrier of $C_5H_{10}^*$ hydrogenation is lower than the desorption energy (0.36 vs. 0.63 eV). In contrast, the $C_5H_{10}^*$ intermediates are more prone to directly desorb to $C_5H_{10}(g)$ or undergo CO insertion instead of hydrogenation on Co_2C-Co and Cs_2O-Co_2C-Co , suggesting that Co_2C species can generally enhance the formations of olefins and oxygenates. The energy barrier differences between $C_5H_{10}^*$ hydrogenation and CO insertion were calculated and used as a selectivity indicator. A larger difference was observed on Cs_2O-Co_2C-Co than on Co_2C-Co (0.34 vs. 0.24 eV), explaining the improved selectivity towards oxygenates/olefins on the former. To further understand the promotional effects of Cs addition on syngas-to-oxygenates/olefins activity, we calculated the projected density of states (PDOS) for the different models (Fig. S11). It was found that the 3d occupied and unoccupied electronic states of Co in Co(111) are significantly localized near the Fermi level. This indicated that the 3d electrons of Co are not easily engaged in strong interactions with the 2p orbital electrons of the adsorbed CO, due to a mismatch in their energy levels. In comparison to the Co_2C-Co model, the incorporation of Cs markedly broadens the distribution of Co's 3d orbitals, particularly promoting the extension of the 3d orbitals toward lower energy levels. This effectively enhances the interaction between the C 2p and Co 3d orbitals, facilitating CO activation as also supported by the Bader charge analysis (Fig. 1f).”*

Fig. 1(f). The reaction coordinate of $C_5H_{10}^*$ desorption, hydrogenation, and CO insertion pathways on Co(111), Co_2C-Co , and Cs_2O-Co_2C-Co models. Bader charge analysis results were shown in the insets.

Fig. S8. Illustrations of the top and side views of the surfaces of Co(111), Co₂C-Co, and Cs₂O-Co₂C-Co.

Fig. S9. Illustrations of different configurations of *CO insertion into *C₅H₁₀ species on the Co(111), Co₂C-Co, and Cs₂O-Co₂C-Co.

Fig. S10. Illustrations of different configurations of *C₅H₁₀ hydrogenation on the Co(111), Co₂C-Co, and Cs₂O-Co₂C-Co.

Table S3. The detailed E , ΔH , $T \times \Delta S$, ZPE, ΔG values for each IS, TS, and FS states of the reaction pathway studied in different calculation models

Model	Adsorbate	E / eV	ZPE / eV	ΔH / eV	$T \times \Delta S$ / eV	ΔG / eV
Co	*C ₅ H ₁₀ +*H	-476.401	3.79037	4.29604	1.01725	3.27879
	TS	-474.539	3.7998	4.32034	1.089303	3.23104
	C ₅ H ₁₁ +	-478.182	3.90762	4.423559	1.096589	3.32697
	*C ₅ H ₁₁ +*H	-481.493	4.09588	4.657957	1.214	3.443957

	TS	-481.256	4.17643	4.68669	1.107665	3.57902
	C ₅ H ₁₂ +	-482.153	4.243895	4.72761	1.05287	3.67474
	*C ₅ H ₁₀ +*CO	-491.074	3.814761	4.439282	1.310257	3.129025
	TS	-489.132	3.78738	4.3921	1.29069	3.1015
	*C ₅ H ₁₀ CO	-489.794	3.85222	4.44694	1.19525	3.25168
Co ₂ C-Co	*C ₅ H ₁₀ +*H	-973.421	3.770	4.281	1.028	3.253
	TS	-971.801	3.724	4.207	0.975	3.232
	C ₅ H ₁₁ +	-973.118	3.900	4.382	0.989	3.393
	*C ₅ H ₁₁ +*H	-977.017	4.068	4.617	1.159	3.458
	TS	-976.347	4.087	4.616	1.102	3.514
	C ₅ H ₁₂ +	-976.889	4.235	4.713	1.011	3.702
	*C ₅ H ₁₀ +*CO	-985.857	3.761	4.406	1.344	3.063
	TS	-984.377	3.795	4.358	1.163	3.196
	*C ₅ H ₁₀ CO	-984.695	3.877	4.481	1.287	3.194
		*C ₅ H ₁₀ +*H	-986.686	3.794	4.300	1.002
Cs ₂ O-Co ₂ C-Co	TS	-984.936	3.706	4.144	0.887	3.257
	C ₅ H ₁₁ +	-986.316	3.916	4.392	0.987	3.405
	*C ₅ H ₁₁ +*H	-990.443	4.089	4.629	1.129	3.500
	TS	-989.663	4.065	4.562	1.030	3.532
	C ₅ H ₁₂ +	-990.413	4.240	4.762	1.162	3.600
	*C ₅ H ₁₀ +*CO	-999.256	3.818	4.448	1.329	3.119
	TS	-997.746	3.803	4.375	1.171	3.204
	*C ₅ H ₁₀ CO	-998.680	3.888	4.476	1.194	3.282
Cu	*H ₂ O	-434.871	0.622854	0.744382	0.269046	0.475336
	TS	-433.461	0.41278	0.506652	0.19411	0.312538
	*OH+*H	-434.788	0.521215	0.624136	0.172818	0.451318
	*CO+*OH+*H	-450.507	0.698897	0.922948	0.420045	0.502903
	TS	-449.488	0.711526	0.900765	0.396387	0.504378
	*COOH+*H	-450.148	0.772586	0.968776	0.421367	0.547409
	TS	-449.075	0.587511	0.734066	0.294878	0.439187
	*CO ₂ +*H ₂	-449.383	0.591737	0.761651	0.35426	0.407389
	*H ₂	-427.223	0.294778	0.410038	0.280709	0.129328
	TS	-426.593	0.369225	0.438501	0.126612	0.311889
	*2H	-427.769	0.350252	0.37815	0.036417	0.341733
	*C ₅ H ₁₁ CHO+*H	-529.36	4.661801	5.155494	1.065767	4.089727
	TS	-527.71	4.618257	5.08528	0.955278	4.130002
	C ₅ H ₁₁ CH ₂ O+	-528.93	4.764797	5.170733	0.800561	4.370172
	C ₅ H ₁₁ CH ₂ O+ H	-532.276	4.9056	5.351431	0.91401	4.43742
	TS	-532.106	4.87405	5.357289	0.99015	4.367139
	*C ₅ H ₁₁ CH ₂ OH+ *	-533.395	5.117547	5.586337	0.96158	4.624758

Cu@ZrO ₂	*H ₂ O	-1744.2	0.634097	0.773856	0.318171	0.455685
	TS	-1743.44	0.419979	0.506379	0.162394	0.343985
	*OH+*H	-1744.29	0.479353	0.58429	0.2266	0.35769
	*CO+*OH+*H	-1757.15	0.53155	0.71298	0.357041	0.35594
	TS	-1756.09	0.568742	0.728394	0.31864	0.409753
	*COOH+*H	-1759.48	0.718562	0.860856	0.292235	0.568621
	TS	-1758.98	0.627312	0.768796	0.289707	0.479089
	*CO ₂ +*H ₂	-1759.39	0.659887	0.813417	0.284827	0.52859
	*H ₂	-1736.3	0.305213	0.373511	0.129036	0.244476
	TS	-1735.97	0.321603	0.392989	0.13511	0.25788
	*2H	-1737.26	0.337568	0.379741	0.064128	0.315613
	*C ₅ H ₁₁ CHO+*H	-1839.46	4.650485	5.047186	0.777926	4.26926
	TS	-1838.82	4.618898	5.04363	0.872195	4.171435
	C ₅ H ₁₁ CH ₂ O+	-1839.38	4.786809	5.199093	0.85693	4.342163
	C ₅ H ₁₁ CH ₂ O+ H	-1842.54	4.970997	5.416286	0.913088	4.503198
	TS	-1842.08	4.931647	5.352619	0.848944	4.503675
	*C ₅ H ₁₁ CH ₂ OH+ *	-1843.15	5.147832	5.560021	0.828798	4.731223

Fig. R1. (a, e) STEM and (c, f, i) HRTEM images of Co_{0.5}Mn_{0.1}Cs/C catalyst. (b) EDX mapping images. (d, g) Line and (e, h) point scanning profiles.

The selectivity for Co/C is unusual. Experimental studies have shown that olefins are the primary products of Co-FT, the high selectivity to methane and oxygenates, and the high alkane/olefin ratio

in Fig 1b should be compared to literature.

Thank you for the very relevant comments. We have supplemented an additional table (Table S2) to compare these important data with those from the previous works in the Supplementary Information. We have also added relevant comments in the revised piece: “In particular, $Co_{0.5}Mn_{0.1}Cs/C$ exhibited lower alkanes/olefins ratios (0.7-0.9) and the highest total selectivity of oxygenates/olefins, with the total selectivity of C_{4+} oxygenates/ C_{3+} olefins reaching 73% (Tables S2).”

Table S2. Catalytic performances of various catalysts for syngas conversion to oxygenates in the literatures.

Catalysts	GHSV / h ⁻¹	H ₂ /CO	T/ °C	X _{CO} /%	S / %				Alk/Ole /%	Refs.
					CO ₂	Alk	Ole	Oxy		
Co/C	2000	2.0	210	21.7	0.4	56.3	24.7	18.5	2.3	TW
Co _{0.1} Cs/C	2000	2.0	210	11.7	2.1	32.3	36.4	29.2	0.9	TW
Co _{0.5} Mn/C	2000	2.0	210	20.0	0.6	39.6	35.8	24.0	1.1	TW
Co ₁ Mn/C	2000	2.0	210	18.9	1.0	37.9	41.0	20.2	0.9	TW
Co _{0.5} Mn _{0.1} Cs/C	2000	2.0	210	29.2	2.7	27.3	31.5	38.6	0.9	TW
Co _{0.5} Mn _{0.1} Cs/C	2000	1.5	210	25.5	2.6	26.1	37.6	33.6	0.7	TW
Co _{0.5} Mn _{0.1} Cs/C	4000	1.5	210	12.4	0.6	22.2	30.4	46.4	0.7	TW
Co ₄ Cu ₁	7200 ^a	1.5	240	11.0	1.0	53.2	15.0	30.8	3.5	1
Co ₁ Cu ₁	2000	2	240	5.0	12.0	35.0	25.0	28.0	1.4	2
Co ₁ Cu ₂ Nb _{0.2}	40 ^b	1.5	200	6.0	1.3	28.3	18.4	52.0	1.5	3
Co ₁ Cu ₁ Mn ₁	2000	2	240	18.0	8.0	34.0	24.0	34.0	1.4	2
Co ₄ Mn ₁ K _{0.1}	40 ^b	1.5	220	18.0	16.0	24.0	20.0	38.0	1.2	4
0.3Rh-CoMn	2000 ^a	1	220	29.5	20.4	24.6	26.7	28.3	0.9	6
0.3Ru-CoMn	2000 ^a	1	220	32.6	23.3	33.7	20.0	22.9	1.7	6
Co/MnOx@quasi-MOF-74	4500 ^a	2	230	21.4	0.8	38.6	21.6	35.9	1.8	7

^a ml g_{cat}⁻¹ h⁻¹, ^b ml min⁻¹.

The grammar should be improved. Words are missing in some of the new sentences in the revised manuscript.

We are sorry for the mistakes and have carefully checked and corrected these.

“Syngas-to-oxygenates/olefins ~~are~~ **is** the key step to produce C_{4+} long-chain alcohols”

“This was further supported by line-scanning of an individual nanoparticle, showing relatively even distributions of Cs and Mn across the nanoparticles but much higher C signal ~~than that of Co~~ at the periphery (Fig. 1d).”

“Two porous organic polymers (POPs)-supported single-Rh-site ~~Rh~~ catalysts”

“Secondly, water-gas shift (WGS) reaction on different Cu catalysts ~~were~~ **was** studied.”

Reviewer #3 (Remarks to the Author):

In the revised manuscript, the authors well replied to the queries and made careful modification. So it may meet the publication requirements.

We are grateful for your support of the publication of our work.

Reviewer #4 (Remarks to the Author):

I have reviewed the revised manuscript submitted by Li et al., along with the authors’ responses to the comments raised in the original round of peer review. Since I have joined the peer review process

at this stage, I have taken into account both the content of the revised submission and the earlier reviewer remarks and authors' replies.

We warmly thanked the Reviewer for his/her careful reading and the insightful comments and guidance on our manuscript.

The authors present a dual-reactor, single-pass system for the conversion of syngas to higher alcohols. While the emphasis on C₄₊ alcohols rather than the more commonly targeted C₂₊ or C₃₊ alcohols is noted, I do not consider this shift in focus to represent a significant conceptual advance. Selectivity toward higher alcohols—except for ethanol, which cannot form via olefin hydroformylation—remains fundamentally constrained by the statistical nature of the Anderson–Schulz–Flory (ASF) product distribution. This is expected given that all alcohol products, either directly or indirectly, originate from FTS hydrocarbons formed on the surface of the first (Co-based) catalyst.

We sincerely thank the Reviewer for this important point. We fully agree that product selectivity in syngas-to-oxygenates/olefins is generally constrained by the ASF distribution. Nevertheless, we would like to emphasize high selectivity of C₄₊ alcohol. Ethanol and propanol are common chemicals, whereas C₄₊ alcohols are used in fine chemicals. The reason why we chose C₄₊ alcohol selectivity as indicator is that butanol can be directly used as the lowest carbon number of plasticizer alcohol, offering significant economic advantages.

In this integrated system, a large fraction of straight-chain α -alcohols is directly produced over the Cs₂O-Co₂C-Co active site, whereas the additional alcohols originate from the hydroformylation of by-product olefins. The overall n/i (normal and isomeric alcohols) ratio of the system is between 4 and 5 (Table S4). In the revised manuscript, we have incorporated the Anderson–Schulz–Flory (ASF) product distribution. In the integrated system, the C₁–C₃ products deviate from the ASF linear distribution, and the chain growth probability α increases from 0.65 to 0.68, indicating that the hydroformylation process effectively modulates product distribution.

We believe this represents a meaningful advance, even within the ASF constraint, as current reports typically achieve <45% C₄₊ oxygenates selectivity (Table S5).

Fig. S3. Products chain lengthening. Detailed selectivity patterns and Anderson-Schulz-Flory

(ASF) chain-lengthening characteristics of the various product classes over (a) Co_{0.1}Cs/C, (b) Co_{0.5}Mn/C, (c) Co_{0.5}Mn_{0.1}Cs/C, (d) integrated system.

In several ways, this study appears as an extension of the authors' earlier work (Li et al., ACS Catal. 2021, 11, 14791–14802; <https://doi.org/10.1021/acscatal.1c04442>), where a similar catalytic pairing, i.e., CoMn/AC for FTS and a Rh-based polymer catalyst for hydroformylation, was implemented within a single reactor using two packed-beds. In my opinion, the above article should have been cited in the current manuscript.

We thank the Reviewer for drawing attention to this point. The data of our previous work (ACS Catal. 2021, 11, 14791–14802) was already included in Fig. 3e for the comparison, and **has now been clearly cited in the revised manuscript** (Fig. 3e and Table S5). Comparisons and references were also made in the manuscript, “*Tandem catalysis approaches have achieved improved results, with notable results including CoMn|CuZnAlZr (46% oxygenates selectivity)²⁵, CuCoAl|t-ZrO₂ (33% C₂₊OH selectivity)²⁶, CuCoAl|ZnO/ZrO₂ (34% C₂₊OH selectivity)²⁷, ZnCrAlO_x|KNiMoS-MMO-5 (60% total alcohol selectivity)²⁸, CoMn/MAC(P)|Rh/3v-PPh₃@POPs (60.8% total oxygenates selectivity)²⁹ and NaPr-CoRu/AomM|Co₂CO₈+ PCy₃ (54% total alcohol)³⁰.*”

We also highlight the key differences: in the earlier work, the C₄₊ oxygenates selectivity plateaued at ~45% (Table S5), whereas in the present study, by rationally regulating the reaction network, precisely constructing the syngas-to-oxygenates/olefins catalyst (Cs₂O-Co₂C-Co) to direct control key the steps, and subsequently integrating the hydroformylation and hydrogenation systems to tailored specific pathways, an outstanding C₄⁺ alcohol selectivity of 80% was achieved. Thus, although our study builds upon prior work, the catalyst design and process segmentation here enable a significant breakthrough for syngas to C₄₊ alcohol with higher selectivity.

The primary distinctions in the present manuscript, relative to the earlier work are: (1) the addition of a third catalyst to hydrogenate aldehydes into alcohols, shifting the major oxygenate product class toward alcohols exclusively, and (2) the segmentation of the process into two reactors, allowing independent temperature control (210 °C for FTS and 140 °C for hydroformylation/hydrogenation), but clearly deviating from the concept of tandem operation of catalysts and reactions.

We appreciate the Reviewer's thoughtful assessment. Agree with the reviewer's opinion and make corresponding modification in advised manuscript. The following actions have been made in the revision:

(i) We have revised the manuscript text (please kindly refer to the Discussion section, or the answers to your next question) to more cautiously present these advances and to clarify that our innovation lies in the precise integration and adaptation of three different catalytic steps, rather than in redefining the concept of “tandem catalysis.”

(ii) We have replaced the term “tandem” with “integrated” throughout the revision.

The experimental work in the current submission is thorough, optimizes these different catalysts (although with promoters which have been well described for the actual role they are meant for here) and the alcohol selectivities achieved are indeed notable, primarily due to the inhibition of C-O cleavage at the stage of hydroformylation and oxygenate hydroformylation thanks to splitting these conversion steps in another reactor, operated at a milder temperature. Beyond the high selectivity, the claim of novelty appears somewhat overstated, especially with respect to the emphasis on C₄⁺

products. This emphasis, though presented as a point of differentiation, seems more arbitrary than chemically justified given the nature of the ASF product distribution. Thus, although the manuscript is compelling, and it deserves publication (following further revision), I do not perceive that the conceptual threshold typically required for a journal such as Nature Communications is reached.

We sincerely thank the Reviewer for the careful evaluation and for recognizing the thoroughness of our experimental work, the design of catalysts, and the notable alcohol selectivity achieved. The reviewer's judgment that the C-O cleavage at slight high reaction temperature (for example, lower than 200 °C in hydroformylation leads to the reduction of alcohols selectivity is not agreed. In fact, under the present reaction conditions, increasing temperature primarily enhances the hydrogenation of olefins rather than promoting C-O cleavage, which generally requires significantly harsher conditions.

We also acknowledge the Reviewer's concern that our emphasis on C₄₊ alcohols could appear overstated, since the ASF product distribution intrinsically governs chain growth probabilities. In the revised manuscript, **we have therefore moderated our claims and clarified that the novelty of our work does not solely lie in the focus on C₄₊ products.**

Syngas-to-long-chain alcohols involves a complex reaction network. In the work, by precisely constructing a Cs₂O-Co₂C-Co active-center structure, achieving the highly selective generation of oxygenates and olefins in one step. Subsequently, **by designing hydroformylation catalysts and anti-CO poisoning aldehydes hydrogenation catalysts under identical reaction conditions capable of effectively suppressing side reactions**—such as the water-gas shift and methanation reactions—we have constructed a directionality and synergistically coupled reaction channels, thereby facilitating the selective synthesis of C₄₊ alcohols. This study successfully resolves the catalyst compatibility issue and enables efficient cooperation among catalytic units within the multifunctional integrated system, representing a key advancement.

In the revised manuscript, **we have carefully rephrased our novelty claims in Abstract and Discussion section to more accurately reflect these advances and to avoid overstatement.** We greatly appreciate the Reviewer's constructive guidance, which has helped us present a more balanced and rigorous account of our work.

Abstract

The selective conversion of syngas to C₄₊ long-chain alcohols holds significant industrial and scientific interest, but challenges in product selectivity and process efficiency remain. Here, we report a precisely **integration integrated** catalytic strategy for C₄₊ alcohol synthesis with a record-breaking selectivity of 80% at 17% CO conversion. **The reaction channel involves: (i) the development of Cs₂O-Co₂C-Co** catalysts capable of catalyzing CO hydrogenation to long-chain oxygenates/olefins; and (ii) subsequent complete conversion to C₄₊ alcohols on the single-Rh-site and Cu-ZrO₂ interfaces by **tandem integrated cooperative** catalysis. Comprehensive catalyst design and compatibility assessment of each catalytic module ensure optimal combinations **for cooperative catalysis**, meanwhile effectively eliminates costly separation steps, and reduces CO₂ selectivity down to 1%. The developed process achieves ultra-high carbon-efficiency (>95%) and improves oxygen-efficiency, effectively overcoming the key limitations of current syngas conversion technologies and thus representing a competitive and sustainable solution for producing high-value long-chain alcohols with a minimal carbon footprint.

Discussion

The novel catalysis catalytic system presented in this study marks a significant breakthrough demonstrates a meaningful advance in the conversion of syngas to C₄₊ long-chain higher alcohols. The results show that careful catalyst design and process integration can substantially shift the product distribution toward C₄₊ long-chain alcohols by precise control of the reaction channels. The system achieves an unprecedented selectivity of 80% C₄₊ long-chain alcohols with a remarkably low CO₂ selectivity of 1% and 17% CO conversion. Under optimized conditions of (H₂/CO = 1.5, F = 130 ml min⁻¹, 210 °C for syngas-to-oxygenates/olefins, 140 °C for hydroformylation/hydrogenation), the system achieved up to 80% selectivity to C₄₊ alcohols with only 1% CO₂ selectivity, 17% CO conversion, and has ultra-high 97% overall carbon efficiency, and improved oxygen efficiency, effectively addressing a key limitation of the current syngas conversion technologies. The in situ reconstructed Cs₂O-Co₂C-Co active sites play a crucial role in directing the steps and intermediates toward oxygenates/olefins. Cs modification enhances Co dispersion favoring olefin desorption and CO insertion. In the subsequent stages by precisely regulating the reaction channels, single-site Rh catalysts enable efficient olefin-to-aldehyde conversion while suppressing overhydrogenation, and CuZrO₂/SiO₂ catalysts effectively hydrogenate aldehydes under syngas conditions, mitigating CO poisoning and minimizing side reactions. The stable CoCs@Co₂C active sites reconstructed in situ play a pivotal role in steering the reaction toward desired intermediates, particularly oxygenates/olefins. Cs modification of CoCs@Co₂C active sites reduce the olefin desorption barrier and facilitate CO insertion, promoting the formation of oxygenates/olefins. In the second stage, single-Rh site catalysts ensure the efficient olefin-to-aldehyde conversion while avoiding overhydrogenation to alkanes. Subsequently, CuZrO₂/SiO₂ catalysts effectively hydrogenate aldehydes under syngas conditions, preventing CO poisoning and minimizing side reactions, thereby achieving high alcohol selectivity with excellent stability. The strategic cooperative catalysis illustrates how rationally matching three distinct catalytic modules under temperature optimized conditions can achieve both high alcohol selectivity and operational stability. Rather than representing a complete departure from ASF-type behavior, this work provides a practical pathway to enhanced higher-alcohol selectivity through precise active-site engineering, staged reactor operation, and catalyst compatibility. We believe these findings offer valuable guidance for designing integrated catalytic systems for the sustainable and scalable production of long-chain alcohols from syngas, between different modules enables efficient and direct syngas to C₄₊ alcohol synthesis with greatly reduced reaction and separation units, thereby, it represents a transformative advance in syngas conversion, offering a competitive and impactful solution for sustainable, scalable, and economically viable production of value-added long-chain alcohols.

The system described extends earlier work while increasing process complexity through the serial arrangement of two separate reactors. Referring to this as a “tandem” process is, in my view, inaccurate: chaining reactors without intermediate workup is standard industrial practice, particularly when distinct reaction conditions (e.g., temperature) are needed for sequential transformations.

We appreciate the Reviewer's thoughtful assessment. Agree with the reviewer's opinion and make corresponding modification in advised manuscript.

We thank the Reviewer again for this clarification. We agree that, strictly speaking, connecting reactors in series is an industrial practice rather than “tandem catalysis” in the conventional sense.

To avoid confusion, we have avoided to use the term “tandem” when describing our approach. At the same time, we emphasize that the challenge addressed here lies not in the serial connection itself, but in achieving effective compatibility of three distinct catalysts and reaction conditions. In addition, the lower-temperature bed benefits from the residual heat of the upstream reactor, contributing to process efficiency. We have clarified these points in the revised text. We have replaced the term “tandem” with “integrated” throughout the revision.

Regarding the authors’ responses to previous reviewer comments, all revision points have been addressed, but not all have been satisfactorily resolved in my opinion.

- (1) In response to requests for greater clarity in the DFT section, the authors have now provided full energetics along the calculated reaction pathways in tabulated form, which is appreciated. However, they have not included the coordinate files for the optimized models, as requested by the reviewers. This is important, particularly because concerns have been raised about the physical realism of the modeled surfaces.

Thank you for the critical comments. We have adopted the reviewers' suggestions and included the coordination file of the optimized model in the revised manuscript. (Supplementary document for details).

For instance, Cs on low work function metal surfaces like Co would be expected to adsorb as a neutral species (e.g., oxide or hydroxide) under FT conditions. The authors model Cs as a cation (Cs^+), but no counterion is presented. I concur with other reviewers that this representation lacks chemical plausibility.

We agree with the Reviewer and have replaced Cs with Cs_2O in the catalyst model in the revision. The new calculation data have been provided in Figs. 1f and S8-10.

- (2) Also, direct comparisons of energy profiles for fundamentally different reactions, such as the water–gas shift (WGS) and CO insertion into C_xH_y species, are overinterpretative without accompanying microkinetic modeling.

We sincerely thank the Reviewer for this constructive comment. We fully agree that direct comparisons of energy barriers between fundamentally different reactions, such as the WGS and hydrogenation steps, are not rigorous without microkinetic modeling. In the revised manuscript, we have removed such fundamentally different reaction comparisons and now only present the energy barriers for valeraldehyde hydrogenation on Cu(111) and CuZrO₂ models (Fig. 2d). This revision avoids potential overinterpretation. We greatly appreciate this valuable suggestion and will pursue microkinetic modeling in our future work to enable more accurate comparisons.

Fig. 1(f). The reaction coordinate of $C_5H_{10}^*$ desorption, hydrogenation, and CO insertion pathways.

Fig. S8. Illustrations of the top and side views of the surface of Co(111), Co₂C-Co, and Cs₂O-Co₂C-Co.

Fig. S9. Illustrations of different configurations of *CO insertion into *C₅H₁₀ species on the Co(111), Co₂C-Co, and Cs₂O-Co₂C-Co.

Fig. S10. Illustrations of different configurations of *C₅H₁₀ hydrogenation on the Co(111), Co₂C-Co and Cs₂O-Co₂C-Co.

Table S3. The detailed E , ΔH , $T \times \Delta S$, ZPE, ΔG values for each IS, TS, and FS states of the reaction pathway studied in different calculation models

Model	Adsorbate	E / eV	ZPE / eV	ΔH / eV	$T \times \Delta S$ / eV	ΔG / eV
Co	*C ₅ H ₁₀ +*H	-476.401	3.79037	4.29604	1.01725	3.27879
	TS	-474.539	3.7998	4.32034	1.089303	3.23104
	C ₅ H ₁₁ +	-478.182	3.90762	4.423559	1.096589	3.32697
	*C ₅ H ₁₁ +*H	-481.493	4.09588	4.657957	1.214	3.443957
	TS	-481.256	4.17643	4.68669	1.107665	3.57902
	C ₅ H ₁₂ +	-482.153	4.243895	4.72761	1.05287	3.67474

	$*C_3H_{10}+*CO$	-491.074	3.814761	4.439282	1.310257	3.129025
	TS	-489.132	3.78738	4.3921	1.29069	3.1015
	$*C_5H_{10}CO$	-489.794	3.85222	4.44694	1.19525	3.25168
Co ₂ C-Co	$*C_5H_{10}+*H$	-973.421	3.770	4.281	1.028	3.253
	TS	-971.801	3.724	4.207	0.975	3.232
	$*C_5H_{11}+*$	-973.118	3.900	4.382	0.989	3.393
	$*C_5H_{11}+*H$	-977.017	4.068	4.617	1.159	3.458
	TS	-976.347	4.087	4.616	1.102	3.514
	$*C_5H_{12}+*$	-976.889	4.235	4.713	1.011	3.702
	$*C_5H_{10}+*CO$	-985.857	3.761	4.406	1.344	3.063
	TS	-984.377	3.795	4.358	1.163	3.196
Cs ₂ O-Co ₂ C-Co	$*C_5H_{10}+*H$	-986.686	3.794	4.300	1.002	3.297
	TS	-984.936	3.706	4.144	0.887	3.257
	$*C_5H_{11}+*$	-986.316	3.916	4.392	0.987	3.405
	$*C_5H_{11}+*H$	-990.443	4.089	4.629	1.129	3.500
	TS	-989.663	4.065	4.562	1.030	3.532
	$*C_5H_{12}+*$	-990.413	4.240	4.762	1.162	3.600
	$*C_5H_{10}+*CO$	-999.256	3.818	4.448	1.329	3.119
	TS	-997.746	3.803	4.375	1.171	3.204
Cu	$*H_2O$	-434.871	0.622854	0.744382	0.269046	0.475336
	TS	-433.461	0.41278	0.506652	0.19411	0.312538
	$*OH+*H$	-434.788	0.521215	0.624136	0.172818	0.451318
	$*CO+*OH+*H$	-450.507	0.698897	0.922948	0.420045	0.502903
	TS	-449.488	0.711526	0.900765	0.396387	0.504378
	$*COOH+*H$	-450.148	0.772586	0.968776	0.421367	0.547409
	TS	-449.075	0.587511	0.734066	0.294878	0.439187
	$*CO_2+*H_2$	-449.383	0.591737	0.761651	0.35426	0.407389
	$*H_2$	-427.223	0.294778	0.410038	0.280709	0.129328
	TS	-426.593	0.369225	0.438501	0.126612	0.311889
	$*2H$	-427.769	0.350252	0.37815	0.036417	0.341733
	$*C_5H_{11}CHO+*H$	-529.36	4.661801	5.155494	1.065767	4.089727
	TS	-527.71	4.618257	5.08528	0.955278	4.130002
	$*C_5H_{11}CH_2O+*$	-528.93	4.764797	5.170733	0.800561	4.370172
	$*C_5H_{11}CH_2O+*$ H	-532.276	4.9056	5.351431	0.91401	4.43742
	TS	-532.106	4.87405	5.357289	0.99015	4.367139
	$*C_5H_{11}CH_2OH+*$	-533.395	5.117547	5.586337	0.96158	4.624758
	Cu@ZrO ₂	$*H_2O$	-1744.2	0.634097	0.773856	0.318171
TS		-1743.44	0.419979	0.506379	0.162394	0.343985

*OH+*H	-1744.29	0.479353	0.58429	0.2266	0.35769
*CO+*OH+*H	-1757.15	0.53155	0.71298	0.357041	0.35594
TS	-1756.09	0.568742	0.728394	0.31864	0.409753
*COOH+*H	-1759.48	0.718562	0.860856	0.292235	0.568621
TS	-1758.98	0.627312	0.768796	0.289707	0.479089
*CO ₂ +*H ₂	-1759.39	0.659887	0.813417	0.284827	0.52859
*H ₂	-1736.3	0.305213	0.373511	0.129036	0.244476
TS	-1735.97	0.321603	0.392989	0.13511	0.25788
*2H	-1737.26	0.337568	0.379741	0.064128	0.315613
*C ₅ H ₁₁ CHO+*H	-1839.46	4.650485	5.047186	0.777926	4.26926
TS	-1838.82	4.618898	5.04363	0.872195	4.171435
C ₅ H ₁₁ CH ₂ O+	-1839.38	4.786809	5.199093	0.85693	4.342163
C ₅ H ₁₁ CH ₂ O+	-1842.54	4.970997	5.416286	0.913088	4.503198
H					
TS	-1842.08	4.931647	5.352619	0.848944	4.503675
*C ₅ H ₁₁ CH ₂ OH+					
*	-1843.15	5.147832	5.560021	0.828798	4.731223

Fig. 2(d). Valeraldehyde hydrogenation reaction on Cu(111) and CuZrO₂ model system.

The reported methane selectivity (1.9-5.1 C%) appears unrealistically low to me, certainly falling below the ASF-predicted value. This is surprising, as the ASF value would be expected to be the theoretically lowest, even if hydrogenation activity, and hence overall conversion, has been heavily inhibited on a Co FTS catalyst. The authors should carefully re-examine their product quantification, include linearized ASF plots for all major product families, and assess the extent of any deviations from the expected statistical distribution. Selectivity trends across product classes are critically dependent on this analysis.

We are extremely grateful for the insightful comments from the Reviewer. **We have supplemented the ASF distribution of syngas-to-oxygenates/olefins and integrated system (Fig. S3).** H₂ and CO chemisorption data showed that the distinct CO-rich and H₂-lean composition on the surface of CoMnCs/C catalyst indicates that its surface properties are unfavorable for hydrogenation reactions to produce methane and other alkanes (Table R1). This also explains the presence of significant olefin content in the product, which is slightly lower than the methane value predicted by ASF

distribution. The CH₄ selectivity of the integrated system at 210 °C was 6.6%, and the ASF distribution also conformed to expectations. In addition, low H₂/CO ratio is another reason for the low methane selectivity. In the manuscript, at a lower H₂/CO ratio of 0.5, the conversion rate is low and the methane selectivity was 1.9% (Table S7). At low reaction rates, the C₁-C₃ products will deviate from the linear ASF distribution, which is a common phenomenon in Fischer-Tropsch synthesis as reported in previous references (*Nat. Commun.* 2019, 10, 3953): “However, significant deviations from the linear ASF behavior are seen for C₁-C₃ on the low rate branch. These deviations seem to be general.”.

Table R1. H₂ and CO chemisorption characterization data for CoMnCs/C samples.

Samples	H ₂ uptake (μmol·g _{cat} ⁻¹ , STP)	CO uptake (μmol·g _{cat} ⁻¹ , STP)
CoMnCs/C	7.2	69.4

Table S7. Influence of H₂/CO ratios on the catalytic performances of Co_{0.5}Mn_{0.1}Cs/C|Rh/3v-POPs-PPh₃|CuZrO₂/SiO₂ tandem system for the syngas-to-C₄₊ alcohols conversion.

H ₂ /CO mol mol ⁻¹	X _{CO} / %	S / %					
		CH ₄	CO ₂	C ₂₊	C ₂₊ ⁼	aldehydes	alcohols
1/2	3.7	1.9	3.1	9.9	1.4	1.4	82.2

Fig. S3. Products chain lengthening. Detailed selectivity patterns and Anderson-Schulz-Flory (ASF) chain-lengthening characteristics of the various product classes over (a) Co_{0.1}Cs/C, (b) Co_{0.5}Mn/C, (c) Co_{0.5}Mn_{0.1}Cs/C, (d) integrated system.

Authors should note that the comparison established in Figure 4b is very much dependent particularly on the CO/syngas conversion level at which selectivities are considered.

We sincerely thank the Reviewer for this insightful comment. We fully acknowledge that product selectivity is strongly dependent on the CO/syngas conversion level. In Fig. 4b, our intention was to provide a comparative overview of the selectivity toward target products (C₄₊ higher alcohols, C₄₊ higher oxygenates, and C₃₊ olefins) obtained from syngas by different pathways reported in the literature. As these pathways employ distinct catalysts (Co-, Fe-, and Mo-based), and operate under

markedly different conditions (reaction temperature, pressure, H₂/CO ratio, etc.), with correspondingly different mechanisms and elementary steps, a direct comparison under identical conversion levels was not feasible (only three reference catalysts with comparable conversions were available in the literature). For completeness, **the detailed catalytic performance, including conversion and selectivity data from the cited references, has now been summarized in Table S5. Furthermore, we have revised Fig. 4b to improve the clarity.**

Fig. 4(b). Comparison on the C₄₊ alcohol and CO₂ selectivity from different technologies. (red, C₄₊ alcohols synthesis Route: C₄₊ alcohols selectivity, 1.CoMnCs/C|Rh₁/3v-POPs-PPh₃|CuZrO₂/SiO₂, 2. CoMnCs/MC|Rh₁/POPs-BP&PPh₃|CuZrO₂/SiO₂; black, Route 1: C₃₊ olefins selectivity, 3.Mn- γ -Fe₅C₂⁴⁴, 4.FeMn@Si⁴⁵, 5. γ -Fe₅C₂⁴⁴, 6.1.5Na-Fe₁Zn₁⁴⁶, 7.Fe-Al₂O₃ (SCS350)⁴⁷, 8.Na₂S-Fe-CNF⁴⁴, 9.Fe-Al₂O₃(SCP)⁴⁷, 10.Fe₃O₄/MAG⁴⁸; blue, Route 2: C₄₊ oxygenates selectivity, 11.Co/MnO_x@quasi-MOF-74⁴⁹, 12.CoMn|CuZnAlZr²⁵, 13.Co₄Mn₁K_{0.1}⁵⁰, 14.ZnCrAlO_x|KNiMoS-MMO-5²⁸, 15.Cu₄Fe₁Mg₄-MMO²⁰, 16.ER-MoS₂-K⁵¹. See Supplementary Table S5 for more details).

Other, minor, comments are:

The authors attribute paraffin formation to excessive hydrogenation of oxygenates. However, complete hydrogenation of oxygenates should yield alcohols, not paraffins. If paraffins are indeed forming, C-O bond cleavage must be occurring, implying additional chemistries that have not been considered in the current discussion.

Thank you for your insightful comments. To more clearly express our current viewpoint, **we have revised this part to more clearly elaborate on this point.** *"In addition to the poisoning effect of CO on hydrogenation catalysts, other side reactions could occur including generations of CH₄ and methanol, and water-gas shift reaction, and excessive hydrogenation of oxygen-containing compounds to alkanes."*

There is some inconsistency in the terminology used for catalyst preparation. The synthesis method is described at times as "incipient wetness impregnation" and at other times as "wetness impregnation." These are distinct techniques, and the usage should be clarified to avoid confusion. The mis correspondence between the composition and the notation for the catalysts has not been solved upon revision, which makes it very hard for a reader to actually understand what the composition of each material discussed in the text is.

Thank you for pointing out the inconsistency. **We have revised the expressions to "incipient**

wetness impregnation". "The modified Co/C catalysts were prepared by incipient wetness impregnation method."

Table S12. Synthetic approach and compositions of different catalysts in this work

Catalyst	Synthesis method	Elements of composition	Nominal Content / wt%	Measured content / wt%
Co/C	incipient wetness impregnation	Co	15	15.2
Co _{0.5} Mn/C	incipient wetness impregnation	Co Mn	15 0.5	14.9 0.5
Co _{0.1} Cs/C	incipient wetness impregnation	Co Cs	15 0.1	15.1 0.1
Co _{0.5} Mn _{0.1} Cs/C	incipient wetness impregnation	Co Mn Cs	15 0.5 0.1	14.9 0.5 0.1
Rh/3v-POPs-PPh ₃	post-loading	Rh	0.125	0.1
Rh/POPs-BP&PPh ₃	post-loading	Rh	0.125	0.1
CuZrO ₂ /SiO ₂	ammonia evaporation	Cu Zr	20 -	19.0 28.0
Cu/SiO ₂	ammonia evaporation	Cu	20	22.0

^a Determined from ICP-OES.

^b Determined from XRF.

In response to the description of the catalyst composition, we have added Table S12 for systematic presentation to facilitate readers' understanding. And to add further clarification "Loading of Co and the other metal additives was 15 wt.% and 0.03~3 wt.%, respectively (Co_xM_yN: M and N represent metal additives; x and y respectively represent the loading amounts of additives M and N Catalysts include: 15Co, 15Co_{0.03}Cs, 15Co_{0.1}Cs, 15Co_{0.3}Cs, 15Co_{0.5}Mn, 15Co₁Mn, 15Co_{0.5}Mn_{0.1}Cs, 15Co₁Mn_{0.1}Cs, 15Co₁Mn_{0.3}Cs, 15Co_{0.5}Mn₁Ni, 15Co_{0.5}Mn₂Ni, 15Co_{0.5}Mn₃Ni, 15Co_{0.5}Mn_{0.1}Ca, 15Co_{0.5}Mn_{0.3}Ca, 15Co_{0.5}Mn₂Cr, 15Co_{0.5}Mn₁Cr, 15Co₁Mn₁Cr, 15Co₁Mn₂Cr, 15Co₂Mn₂Cr, 15Co_{0.5}Mn_{0.1}Ba, 15Co_{0.5}Mn_{0.3}Ba, 15Co_{0.5}Mn_{0.5}Ba, 15Co_{0.5}Mn₁Ba, 15Co_{0.5}Mn₂Ba, 15Co_{0.5}Mn₃Ba).

Finally, the term "net CO₂ emissions" is typically associated with life-cycle analysis. Its use here to describe a scenario where minor CO₂ side-products are recycled upstream is, in my opinion, imprecise and should be revised.

We appreciate the insightful suggestions from the Reviewer and have made the corresponding revisions. "By coupling residual CO₂ with green hydrogen via the reverse water-gas shift reaction to recycle syngas, ~~the process might approach net-zero CO₂ emissions, significantly reducing greenhouse gas~~ emissions can be significantly reduced ~~outputs~~."

References

- 1 Xiang, Y., Barbosa, R. & Kruse, N. Higher alcohols through CO hydrogenation over CoCu catalysts: influence of precursor activation. *ACS Catal.* **4**, 2792-2800, (2014).
- 2 Xiang, Y. *et al.* Long-chain terminal alcohols through catalytic CO hydrogenation. *J. Am. Chem. Soc.* **135**, 7114-7117, (2013).
- 3 Xiang, Y., Barbosa, R., Li, X. & Kruse, N. Ternary cobalt-copper-niobium catalysts for the selective CO hydrogenation to higher alcohols. *ACS Catal.* **5**, 2929-2934, (2015).
- 4 Xiang, Y. & Kruse, N. Tuning the catalytic CO hydrogenation to straight- and long-chain aldehydes/alcohols and olefins/paraffins. *Nat. Commun.* **7**, 13058, (2016).
- 5 Lin, T. *et al.* Direct production of higher oxygenates by syngas conversion over a multifunctional catalyst. *Angew. Chem. Int. Ed.* **58**, 4627-4631, (2019).
- 6 Qin, T. *et al.* Tuning chemical environment and synergistic relay reaction to promote higher alcohols synthesis via syngas conversion. *Appl. Catal. B: Environ.* **285**, 119840 (2021).
- 7 Cui, W. G. *et al.* In situ encapsulated Co/MnOx nanoparticles inside quasi-MOF-74 for the higher alcohols synthesis from syngas. *Appl. Catal. B: Environ.* **278**, 119262 (2020).

Response to Reviewers

Comments in blue - Replies in black - Actions in bold - Citations in *italic*

Reviewer #2 (Remarks to the Author):

As suggested, the authors have improved the DFT model of the catalyst. Over the last decade, it has been established that alkylidyne species ($\text{CH}_3\text{CH}_2\text{C}$) are the key reaction intermediates that undergo hydrogenation, dehydrogenation or CO insertion. Direct olefin hydrogenation is not a relevant reaction in Fischer-Tropsch synthesis.

We gratefully acknowledge the reviewer for his/her insightful and constructive comments. Instead of using olefin hydrogenation as a model reaction, **we have now simulated the three key transformation routes for alkylidyne species ($\text{CH}_3\text{CH}_2\text{C}$) on different model catalysts (Figs. 1f, S8-11 and Table S3)**, as suggested by the reviewer.

The relative energies for the formation of three different products, including oxygenates, olefins, and alkanes, were compared. These results clearly show that the hydrogenation of alkylidyne species to the corresponding alkane is most favored on Co(111) with the lowest barrier of only 1.05 eV (1.09 and 1.34 eV, respectively, on $\text{Co}_2\text{C-Co}$ and $\text{Cs}_2\text{O-Co}_2\text{C-Co}$). Instead, the hydrogenation of alkylidyne species to the corresponding olefins on Co(111) is energy-demanding with a barrier of 1.83 eV, whereas it is only 1.03 and 0.90 eV, respectively, on $\text{Co}_2\text{C-Co}$ and $\text{Cs}_2\text{O-Co}_2\text{C-Co}$. For the formation of oxygenates, the CO insertion step is also accompanied with a relatively higher barrier of 1.37 eV on metallic Co, in comparison to only 0.86 eV on $\text{Co}_2\text{C-Co}$ and 0.82 eV on $\text{Cs}_2\text{O-Co}_2\text{C-Co}$. These newly supplemented DFT results are fully consistent with our experimental data, and thus provide a deeper molecular-level understanding of the divergent product distributions on the distinct active sites.

We have now replaced Fig. 1f with the newly derived DFT results in the revision, and rephrased the discussions accordingly.

“To explain the different product distributions, three key fundamental steps including the formation of alkanes, olefins, and oxygenates from alkylidyne species ($\text{C}_3\text{H}_7\text{CH}_2\text{C}^$), the presumed key reaction intermediates, were simulated (Figs. 1f, S8-S11 and Table S3). For Co(111), the energy barrier for the formation of alkanes from C_3H_9^* is much lower than those for olefins and oxygenates (1.05 vs. 1.83, 1.37 eV), explaining the high propensity of deep hydrogenation on pure metallic cobalt. In contrast, the C_5H_9^* intermediates are more prone to forming $\text{C}_3\text{H}_7\text{CHCH}_2^*$ or undergoing CO insertion instead of hydrogenation to alkanes on $\text{Co}_2\text{C-Co}$ and $\text{Cs}_2\text{O-Co}_2\text{C-Co}$, suggesting that $\text{Co}_2\text{C-Co}$ interphase can generally enhance the formations of olefins and oxygenates. To further reveal the promotional role of Cs, the energy barrier differences between C_5H_9^* hydrogenation to alkanes and olefins, CO insertion were calculated and used as a selectivity indicator. A larger difference was observed on $\text{Cs}_2\text{O-Co}_2\text{C-Co}$ than on $\text{Co}_2\text{C-Co}$ (0.44 vs. 0.06 eV), explaining the improved selectivity towards oxygenates/olefins on the former.”*

Fig. 1f. Design of efficient syngas-to-oxygenates/olefins catalysts. The reaction coordinate of $C_3H_7CH_2C^*$ desorption, hydrogenation, and CO insertion pathways on Co(111), Co_2C-Co , and Cs_2O-Co_2C-Co models. Bader charge analysis were shown in the inset.

Fig. S9. Illustrations of different configurations of $C_3H_7CH_2C^*$ desorption on the Co(111), Co_2C-Co and Cs_2O-Co_2C-Co .

Fig. S10. Illustrations of different configurations of $C_3H_7CH_2C^*$ hydrogenation on the Co(111), Co_2C-Co and Cs_2O-Co_2C-Co .

Fig. S11. Illustrations of different configurations of CO^* insertion into $C_3H_7CH_2C^*$ species on the Co(111), Co_2C-Co and Cs_2O-Co_2C-Co .

Table S3. The detailed E, H, S, ZPE, G values for each IS, TS, and FS states of the reaction pathway studied in different calculation models

Model	Adsorbate	E	ZPE	H	S	G
Co	*CCH ₂ C ₃ H ₇ +*H	-474.95	3.534	4.019	0.989	3.030
	TS	-474.27	3.489	3.960	0.956	3.004
	CHCH ₂ C ₃ H ₇ +	-474.278	3.698	4.219	1.061	3.158
	TS	-472.448	3.370	3.881	1.088	2.793
	*CHCHC ₃ H ₇ +*H	-474.094	3.462	3.967	1.060	2.907
	TS	-472.934	3.730	4.211	0.988	3.223
	CH ₂ CHC ₃ H ₇ +	-472.941	3.893	4.377	1.000	3.377
	*CHCH ₂ C ₃ H ₇ +*H	-478.162	3.698	4.219	1.061	3.158
	TS	-477.592	3.730	4.211	0.988	3.223
	CH ₂ CH ₂ C ₃ H ₇ +	-477.964	3.893	4.377	1.000	3.377
	*CH ₂ CH ₂ C ₃ H ₇ +*H	-481.88	4.083	4.621	1.127	3.495
	TS	-480.83	4.037	4.586	1.170	3.416
	CH ₃ CH ₂ C ₃ H ₇ +	-482.229	4.259	4.777	1.144	3.633
	*CHCH ₂ C ₃ H ₇ +*CO	-490.867	3.677	4.292	1.265	3.027
TS	-489.777	3.737	4.344	1.274	3.070	
COCHCH ₂ C ₃ H ₇ +	-490.13	3.798	4.411	1.287	3.123	
Co ₂ C-Co	*CCH ₂ C ₃ H ₇ +*H	-969.528	3.513	4.006	1.036	2.970
	TS	-968.492	3.470	3.962	1.013	2.949
	CHCH ₂ C ₃ H ₇ +	-968.527	3.581	4.076	1.028	3.047
	TS	-967.788	3.405	3.917	1.078	2.839
	*CHCHC ₃ H ₇ +*H	-968.934	3.458	3.944	0.964	2.980
	TS	-967.887	3.389	3.848	0.912	2.936
	CH ₂ CHC ₃ H ₇ +	-968.642	3.604	4.097	1.028	3.070
	*CHCH ₂ C ₃ H ₇ +*H	-972.649	3.754	4.271	1.046	3.225
	TS	-972.067	3.767	4.256	0.985	3.271
CH ₂ CH ₂ C ₃ H ₇ +	-972.188	3.891	4.398	1.054	3.345	

	*CH ₂ CH ₂ C ₃ H ₇ +*H	-976.258	4.072	4.613	1.126	3.487
	TS	-975.168	4.094	4.577	0.983	3.593
	CH ₃ CH ₂ C ₃ H ₇ +	-976.757	4.255	4.698	0.912	3.786
	*CHCH ₂ C ₃ H ₇ +*CO	-985.418	3.768	4.400	1.300	3.100
	TS	-984.556	3.728	4.349	1.271	3.079
	COCHCH ₂ C ₃ H ₇ +	-984.819	3.779	4.373	1.203	3.170
Cs ₂ O-Co ₂ C-Co	*CCH ₂ C ₃ H ₇ +*H	-983.122	3.528	4.017	0.995	3.023
	TS	-982.226	3.477	3.967	1.008	2.959
	CHCH ₂ C ₃ H ₇ +	-982.362	3.588	4.077	1.013	3.065
	TS	-981.784	3.400	3.867	0.945	2.922
	*CHCHC ₃ H ₇ +*H	-982.744	3.477	3.963	0.988	2.975
	TS	-982.001	3.463	3.927	0.934	2.993
	CH ₂ CHC ₃ H ₇ +	-982.265	3.591	4.078	0.995	3.084
	*CHCH ₂ C ₃ H ₇ +*H	-986.395	3.767	4.281	1.034	3.247
	TS	-985.455	3.762	4.259	1.022	3.237
	CH ₂ CH ₂ C ₃ H ₇ +	-985.83	3.874	4.380	1.054	3.326
	*CH ₂ CH ₂ C ₃ H ₇ +*H	-989.829	4.048	4.584	1.105	3.480
	TS	-988.487	4.056	4.559	1.075	3.483
	CH ₃ CH ₂ C ₃ H ₇ +	-990.342	4.266	4.786	1.142	3.644
	*CHCH ₂ C ₃ H ₇ +*CO	-999.288	3.752	4.377	1.280	3.097
	TS	-998.468	3.711	4.340	1.301	3.039
	COCHCH ₂ C ₃ H ₇ +	-998.548	3.751	4.353	1.199	3.154

It has also been established that olefins are the primary products of cobalt Fischer-Tropsch synthesis, but thermodynamically, hydrogenation to paraffins is of course favorable and will dominate at higher conversions.

We sincerely thank the reviewer for this insightful comment. We fully agree with the reviewer that alkane formation is thermodynamically more favored for cobalt-catalyzed Fischer-Tropsch synthesis, particularly at high conversions. We kindly remind the reviewer that, in most of our cases, the single-pass CO conversion was moderate (< 25%).

Moreover, both kinetic and thermodynamic factors play a decisive role in governing the reaction pathway and product distribution. As has been discussed in the replies to your previous comments, our DFT calculations across the three models reveal that the Cs₂O-Co₂C-Co active center significantly lowers the reaction energy barriers for the formation of both olefins and alcohols in comparison to alkanes. These pronounced kinetic effect induced by the proper design of active sites should play a decisive role in steering the products to olefins and oxygenates instead of paraffins.

Reviewer #4 (Remarks to the Author):

The authors have further revised their manuscript. Important flaws in the earlier version have been addressed, for example the construction of more relevant DFT models and the inclusion of Cs as a promoter in a more sensible speciation therein. Other technical aspects have also been improved, making the manuscript technically more sound.

We thank the reviewer for these encouraging remarks. We are glad that the manuscript was found technically more sound. Further improvements related to the novelty and ASF distribution are discussed below.

For publication in Nature Communications, I still consider that splitting a previously published process (by the same authors, ACS Catalysis 2021) into two reactors in series, to gain a more independent temperature control, does not constitute a sufficiently solid innovation.

We kindly remind the reviewer that this work does not simply spit the previous system into two reactors. Instead, we stress that catalyst innovation in FTS and compatibility of different catalytic modules in the current two-bed system are key to the exceptional C₄₊ alcohol formation.

First, the discovery of Cs₂O-modified Co₂C-Co interfacial sites for olefins- and oxygenates-oriented synthesis is novel and pivotal in this study, as it lies a solid foundation for the further transformations (hydroformylation, and hydrogenation) of the intermediates. To better explain this point, **we have performed an additional test, by accommodating the currently developed three catalysts into one reactor, and compared the results with those from our previously reported data in ACS Catal (see Table R1).**

The new results demonstrate that the current system not only enhances the catalytic performance (CO conversion 35.7% vs. 29.7%), and but also significantly improves C₄₊ alcohols selectivity (from 23.0% to 47.8%). Our DFT simulations (**Fig. 1f**) also demonstrate the much lower reaction energy barriers for the olefins/oxygenates formation on Cs₂O-Co₂C-Co, in comparison to those on the previously established Co₂C-Co and metallic Co sites. These results ambiguously demonstrated the superior performance of the newly developed Cs₂O-Co₂C-Co catalysts in steering the formation of high alcohols, thus providing valuable insights to guide catalyst design.

Second, the compatibility of three catalytic modules is also critical to the optimized C₄₊ alcohol selectivity. As have been stressed in the previous revision, complex interplays exist in the three cascade catalyst beds, primarily owing to the multiple effluents from the front beds. This will inevitably cause significant challenges for the design of robust catalysts from downstream transformations, for instance, constructions of monometallic Rh sites to enhance the hydroformylation activity toward multi-carbon mixed olefins instead of a single one, and of efficient hydrogenation ensembles tolerable of CO poisoning (**Fig. 2b**) and mitigating the water-gas shift side reaction (**Fig. 2c**). For this reason, we made significant efforts in the screening of different catalyst formulas to finally obtain the optimal combination of the three catalytic modules (**Fig. 2a**). Additionally, as pointed out by the reviewer, two-bed technology allows for more delicate tuning of the individual catalytic modules to reach the optimal performance. Building upon these results, we achieved a record-high C₄₊ selectivity of 80% at a single pass CO conversion of 17.2%.

Table R1. The catalytic performances of different combinations of catalysts in the tandem system for the syngas-to- C_{4+} alcohols conversion.

Catalysts	$X_{CO} / \%$	$S / \%$						$S_{C_{4+}OH} / \%$
		CH ₄	CO ₂	C ₂₊	C ₂₊ ⁼	aldehydes	alcohols	
A ^a	29.7	8.0	5.5	29.1	13.6	16.4	27.4	23
B ^a	35.7	5.5	6.8	23.9	3.3	1.1	59.4	48
C ^b	17.2	1.3	1.4	10.7	0.0	0.0	86.5	80

A: Our previous work reported in *ACS Catal.*

B and C: This work.

^a Reaction conditions: $H_2/CO = 2$, $F = 65 \text{ ml min}^{-1}$, $T = 210 \text{ }^\circ\text{C}$ (syngas-to-oxygenates/olefins and hydroformylation/hydrogenation), and $P = 3 \text{ MPa}$.

^b Reaction conditions: $H_2/CO = 1.5$, $F = 130 \text{ ml min}^{-1}$, $210 \text{ }^\circ\text{C}$ (syngas-to-oxygenates/olefins), $140 \text{ }^\circ\text{C}$ (hydroformylation/hydrogenation) and $P = 8 \text{ MPa}$.

Fig. 1f. Design of efficient syngas-to-oxygenates/olefins catalysts. The reaction coordinate of $C_3H_7CH_2C^*$ desorption, hydrogenation, and CO insertion pathways on Co(111), Co_2C-Co , and Cs_2O-Co_2C-Co models. Bader charge analysis were shown in the inset.

Fig. 2. Compatibility assessment between syngas-to-oxygenates/olefins and the downstream hydroformylation or hydrogenation processes. (a) Catalyst performances between syngas-to-oxygenates/olefins plus hydroformylation (left) and syngas-to-oxygenates/olefins plus hydrogenation (right). Reaction conditions: 210 °C (syngas-to-oxygenates/olefins) and 140 °C (hydroformylation/hydrogenation), H₂/CO = 1.5, 3.0 MPa, *F* = 65 ml min⁻¹. (b) Catalytic performances of 5Pt/TiO₂ and 5Pt4Sn/TiO₂ for valeraldehyde hydrogenation in H₂ and syngas streams. Reaction conditions: 5Pt/TiO₂ or 5Pt4Sn/TiO₂, 1.5 g; *T* = 140 °C; *P* = 3.0 MPa; *F* (valeraldehyde) = 0.01 ml min⁻¹; *F* (H₂) or *F* (H₂/CO = 2) = 30 ml min⁻¹. (c) WGS performances and CO uptakes of CuZnAl, Cu/SiO₂, and CuZrO₂/SiO₂. Reaction conditions: catalysts 1.5 g; *T* = 140 °C; *P* = 3.0 MPa; *F* (H₂O) = 0.01 ml min⁻¹; *F* (CO) = 30 ml min⁻¹. (d) Valeraldehyde hydrogenation reaction pathways on Cu(111) and CuZrO₂ model system.

In addition, several of the reported performance indicators remain unconvincing. As an example, the authors claim to achieve a selectivity to C₄+ alcohols above 80%, following an ASF distribution with a chain growth probability of approximately 0.67 (Figure S3). This is presented as a major achievement of this study. However, for said chain growth probability, the mathematically derived absolute maximum carbon selectivity to all products with four or more carbon atoms (C₄+) is approximately 60%. How, then, can the authors report selectivities to C₄+ alcohols as high as 80%? In my view, this strongly suggests issues in the product analysis and/or data treatment. It is similarly surprising that alcohols as short as C₁ and C₂ (methanol, ethanol) follow the same ASF distribution, despite the fact that neither of those could be produced by hydroformylation of olefins. All in all, I believe this study still requires substantial technical revision, including a critical assessment of product analysis and carbon balances, before it can be considered for publication.

We sincerely thank the reviewer for the careful evaluation. We are sorry for not clearly clarifying the relations between the α values from ASF distribution and the C₄+ alcohol selectivity in the previous revision. We kindly reminded the reviewer that the α values reported in previous Fig. S3 was associated only with FTS reaction, not for the integrated systems. As more complicated reactions can occur in the integrated systems, including hydroformylation, hydrogenation and etc.,

the product selectivity in the integrated system can be significantly deviated from the single-step FTS reaction.

To address this and to explain for the high C_{4+} alcohol selectivity of the developed integrated systems, we have now also plotted the product distribution of the integrated systems and compared the α values between single-step FTS and the integrated ones (Fig. S3e,f, Table R2). Several features are noteworthy,

i) For the FTS, all the catalysts follow the standard linear ASF distribution, the theoretical C_{4+} selectivity is roughly $\sim 56\text{-}60\%$ ($\alpha=0.65\text{-}0.67$) at 3 MPa, and *ca.* $\sim 69\%$ ($\alpha=0.72$) at elevated pressure of 8 MPa (indicating the favorable chain-growth probability at higher pressures);

ii) In our real integrated systems, mainly due to the chain elongation *via* hydroformylation (C_n olefins $\rightarrow C_{n+1}$ aldehydes, olefins selectivity of the FTS products: approximately 30%, Table S1), C_{4+} products still follow the linear relation but $C_1\text{-}C_3$ products show strong deviation. We then calculated the α values of the integrated systems starting from C_4 . By assuming that $C_1\text{-}C_3$ will follow the same trend, the theoretical C_{4+} selectivity of the integrated systems can be estimated, *i.e.*, $\sim 63\%$ ($\alpha=0.68$) and $\sim 69\%$ ($\alpha=0.72$), respectively, at 3 and 8 MPa. These theoretical values are still lower than our reported C_{4+} alcohol selectivity of 68% (3 MPa) and 80% (8 MPa), which can be reasonably explained by the strong deviation of $C_1\text{-}C_3$ products from the ASF distribution leading to largely underestimated values.

Based on the above reasoning, we trust our results on C_{4+} alcohol selectivity are reasonable.

We have now provided the ASF plots of both individual FTS and integrated systems in revised Fig. S3, and the results are briefly discussed in the revised piece.

Fig. S3 Products chain lengthening. Detailed selectivity patterns and Anderson-Schulz-Flory (ASF) chain-lengthening characteristics of the various product classes over (a) $Co_{0.1}Cs/C$ (3 MPa), (b) $Co_{0.5}Mn/C$ (3 MPa), (c) $Co_{0.5}Mn_{0.1}Cs/C$ (3 MPa), (d) $Co_{0.5}Mn_{0.1}Cs/C$ (8 MPa). (e) integrated system (3 MPa), (f) integrated system (8 MPa)

“Anderson-Schulz-Flory (ASF) plots have been constructed and are compiled in Fig. S3a-d. Accordingly, a linear ASF behavior with chain lengthening probability $\alpha = 0.65$ is obtained, and as the pressure increased to 8 MPa, α increased to 0.72. Control experiments showed that additions of

individual Mn or Cs could also improve the total selectivity of oxygenates/olefins and reduce CH₄ selectivity, with α values of 0.67 and 0.66, respectively, but both led to decreased activity (Table S1).”

“In the integrated system, the C₁-C₃ products **strongly** deviate from the ASF linear distribution, and the chain growth probability α increases from 0.65 to 0.68, indicating that the hydroformylation process effectively modulates product distribution (Fig. S3). **As the reaction pressure increased from 3.0 to 8.0 MPa, the deviation of C₁-C₃ products from the ASF linear distribution became increasingly pronounced (Fig. S3e,f), while the chain growth probability α rose from 0.68 ± 0.01 to 0.72 ± 0.01 , demonstrating that elevated pressure enhances carbon chain elongation and suppresses the formation of low-carbon products.**”

Furthermore, to resolve concerns regarding product quantification and carbon balance, **we have recalculated the total carbon input using precise measurements of feed gas (syngas) flow rate and composition (Tables R2, R3).** The total carbon output was determined by summing the carbon content across all product streams, including liquid phases (organic and aqueous), solid wax, and gaseous components in the tail gas. This comprehensive and quantitative approach ensured a carbon balance within 100±5%, robustly validating the accuracy and reliability of our product analysis.

Table R2. Carbon balance summary table ^a (3 MPa)

Input	Carbon molar flow rate in the feed gas / mmol	1671.4
	Total carbon input / mmol	1671.4
Output	Carbon molar flow rate in the off-gas / mmol	1457.3
	Carbon molar flow rate in the aqueous phase / mmol	20.0
	Carbon molar flow rate in the oil phase / mmol	114.9
	Carbon molar flow rate in the wax phase / mmol	56.5
	Total carbon output / mmol	1648.7
Carbon balance		98.6%

Table R3. Carbon balance summary table ^a (8 MPa)

Input	Carbon molar flow rate in the feed gas / mmol	1671.4
	Total carbon input / mmol	1671.4
Output	Carbon molar flow rate in the off-gas / mmol	1400.7
	Carbon molar flow rate in the	6.4

	aqueous phase / mmol	
	Carbon molar flow rate in the oil phase / mmol	127.6
	Carbon molar flow rate in the wax phase / mmol	115.3
	Total carbon output / mmol	1650.0
Carbon balance		98.7%

^aAfter stabilization for 24 hours, carbon balance data were obtained by sampling at 12 h intervals. Reaction conditions: H₂/CO = 1.5, *F* = 130 ml min⁻¹, 210 °C (syngas-to-oxygenates/olefins), 140 °C (hydroformylation/hydrogenation).

We have now described the carbon balance in the revision.

“A comprehensive analysis, including liquid phases (organic and aqueous), solid wax, and gaseous components, leads to excellent carbon balances (generally within 100 ± 5%).”